# Small heat shock proteins operate as molecular chaperones in the mitochondrial intermembrane space

Elias Adriaenssens[1,2], Bob Asselbergh [3,4], Pablo Rivera-Mejías[5], Sven Bervoets [6], Leen Vendredy[1,2], Vicky De Winter[1,2], Katrien Spaas[3,4], Riet de Rycke[7,8,9], Gert van Isterdael[8,9,10], Francis Impens [11,12,13], Thomas Langer [5,14] & Vincent Timmerman [1,2] ✉

Mitochondria are complex organelles with different compartments, each harbouring their own protein quality control factors. While chaperones of the mitochondrial matrix are well characterized, it is poorly understood which chaperones protect the mitochondrial intermembrane space. Here we show that cytosolic small heat shock proteins are imported under basal conditions into the mitochondrial intermembrane space, where they operate as molecular chaperones. Protein misfolding in the mitochondrial intermembrane space leads to increased recruitment of small heat shock proteins. Depletion of small heat shock proteins leads to mitochondrial swelling and reduced respiration, while aggregation of aggregation-prone substrates is countered in their presence. Charcot–Marie–Tooth disease-causing mutations disturb the mitochondrial function of HSPB1, potentially linking previously observed mitochondrial dysfunction in Charcot–Marie–Tooth type 2F to its role in the mitochondrial intermembrane space. Our results reveal that small heat shock proteins form a chaperone system that operates in the mitochondrial intermembrane space.

Mitochondria are composed of more than 1,100 proteins, of which the nuclear genome encodes more than 99% (refs. [1–3]). Most mitochondrial proteins are therefore imported from the cytosol, and this poses a considerable burden on mitochondrial surveillance systems[4,5]. The mitochondrial matrix is equipped with multiple chaperone systems, including mHSP60 and mHSP70 (refs. [1,6]). However, classical chaperones, like the Hsp70 and Hsp90 families, were not found in the mitochondrial intermembrane space (IMS)[7].

The only IMS chaperones known so far are the small translocases of the inner mitochondrial membrane (TIMMs), CHCHD4 (Mia40 in yeast) and ClpB (Skd3 in yeast). The small TIMMs and Mia40/CHCHD4 protect, respectively, transmembrane and disulphide-rich proteins during

[1]Peripheral Neuropathy Research Group, Department of Biomedical Sciences, University of Antwerp, Antwerp, Belgium. [2]Laboratory of Neuromuscular Pathology, Institute Born Bunge, Antwerp, Belgium. [3]Neuromics Support Facility, VIB Center for Molecular Neurology, University of Antwerp, Antwerp, Belgium. [4]Neuromics Support Facility, Department of Biomedical Sciences, University of Antwerp, Antwerp, Belgium. [5]Max Planck Institute for Biology of Ageing, Cologne, Germany. [6]VIB Center for Molecular Neurology, University of Antwerp, Antwerp, Belgium. [7]VIB Bioimaging Core, Ghent University, Ghent, Belgium. [8]Department of Biomedical Molecular Biology, Ghent University, Ghent, Belgium. [9]VIB Center for Inflammation Research, Ghent, Belgium. [10]VIB Flow Core, Ghent, Belgium. [11]VIB-UGent Center for Medical Biotechnology, VIB, Ghent, Belgium. [12]VIB Proteomics Core, VIB, Ghent, Belgium. [13]Department of Biomolecular Medicine, Ghent University, Ghent, Belgium. [14]Cologne Excellence Cluster on Cellular Stress Responses in Aging-Associated Diseases (CECAD), University of Cologne, Cologne, Germany. ✉e-mail: vincent.timmerman@uantwerpen.be

mitochondrial import[8–19]. Skd3/ClpB is the only IMS chaperone known to counter protein aggregation after substrate import[20]. However, most cellular compartments harbour multiple chaperones that cooperate to counter protein aggregation, and chaperones that complement Skd3/ClpB in the mitochondrial IMS have yet to be identified.

The periplasm of Gram-negative bacteria is considered the equivalent of the mitochondrial IMS and forms an excellent example of a multi-chaperone system. The periplasm is devoid of ATP and contains several ATP-independent chaperones such as Skp, Spy and SurA[21,22]. Like bacteria, humans also possess a class of ATP-independent chaperones known as small heat shock proteins (sHSPs). The monomeric low molecular mass (12–45 kDa) subunits are the building blocks for dynamic oligomeric ensembles up to 600 kDa and larger[23,24]. sHSPs maintain cellular proteostasis through rapidly recognizing unfolded or misfolded proteins and transferring them to larger ATP-dependent chaperones[24–26].

Mutations in the sHSPs HSPB1 (Hsp27) and HSPB8 (Hsp22) cause axonal Charcot–Marie–Tooth (CMT) disease and related disorders[27–33]. Mitochondrial defects were observed in motor neurons from in vitro and in vivo models of CMT, including patient-derived induced pluripotent stem cells expressing mutant HSPB1 and HSPB8 (refs. [34–37]). It is however unknown how these cytosolic chaperones can impair mitochondrial function[38–40].

Unlike the sHSPs of metazoans, which are predominantly found in the cytosol and nucleus, plant sHSPs are targeted to virtually every membrane-enclosed compartment, including mitochondria[41–46]. *Drosophila melanogaster* and *Toxoplasma gondii* are currently the only other eukaryotes known to have mitochondria-localized sHSPs[25,47,48]. However, human HSPB1 was found the most upregulated protein in retinas of *Yme1l* knockout mice, and human HSPB1 was also found to translocate to mitochondria in HAX1-deficient cells[49,50], raising the question whether a mitochondrial-protective function by sHSPs is preserved across metazoans.

In this Article, we report that mammalian cytosolic sHSPs undergo mitochondrial import and translocate into the mitochondrial IMS, where they form an effective IMS chaperone system.

## Results

### sHSPs reside in the mitochondrial IMS

To find out how cytosolic sHSPs cause mitochondrial dysfunction in CMT, we sought to understand the connection between sHSPs and mitochondria in mammals. We verified whether human sHSPs are present in mitochondria under basal or stress conditions. The stress condition consisted of an elevated temperature (hereafter referred to as heat shock) to induce protein misfolding and boost sHSPs activity. By collecting cells right after the heat shock, we primarily study the activation of pre-existing sHSPs, rather than de novo production of sHSPs, which occurs 4–6 h after heat shock.

To assess whether sHSPs are present in mitochondria, we isolated mitochondria from untreated and heat shock-treated HeLa cells to analyse the amount of sHSPs in the mitochondrial fraction. We detected low amounts of HSPB1 (Hsp27), HSPB5 (Hsp25) and HSPB8 (Hsp22)

in the mitochondrial fraction under basal conditions but observed a substantial enrichment after heat shock (Fig. 1a). As HeLa cells express only three sHSPs, we also isolated mitochondria from the mouse myoblast C2C12 cell line, which expresses more sHSPs. In C2C12 cells, we detected nearly all sHSPs in the mitochondrial fraction under basal conditions and observed enrichment of sHSPs after heat shock (Fig. 1a). We obtained comparable results with other cell lines, including primary human lymphoblasts (Extended Data Fig. 1a), confirming the presence of sHSPs in the mitochondrial fraction under basal conditions and their enrichment after heat shock. We calculated that approximately 1/275 molecules of HSPB1 (or 0.4%) resides in the mitochondrial fraction, which increases up to 3% after heat shock. The translocation of sHSPs from the cytosol to mitochondria after heat shock occurred within 20 min (Extended Data Fig. 1b) before any notable Heat Shock Factor 1 (HSF-1)-induced protein synthesis had occurred.

Next, we used a proteinase K protection assay (Extended Data Fig. 2a) to determine whether sHSPs are imported into mitochondria. We observed that, under basal conditions, most sHSPs were protected from proteinase K when mitochondria were intact. Only upon mitochondrial swelling and bursting of the mitochondrial outer membrane (OM) did proteinase K digest the sHSPs, suggesting that sHSPs reside in the IMS under basal conditions (Fig. 1b and Extended Data Figs. 2–4). After heat shock, the sHSPs accumulated mostly on the OM (Extended Data Fig. 2b). We decided to focus on the role of sHSPs in the IMS (under non-heat shock conditions) as, so far, only a single protein disaggregase has been identified in the mammalian IMS[20].

While most sHSPs reside in the IMS, some, such as HSPB3, are not imported into mitochondria (Fig. 1a and Extended Data Fig. 4). HSPB2 was also largely retained from mitochondrial import, which might be explained by the formation of an oligomeric complex between HSPB2 and HSPB3 (refs. [51,52]). Similarly, HSPB8 was also partially retained on the OM (Fig. 1b and Extended Data Fig. 4), and its stable binding partner BAG3 (refs. [53–55]) is also not imported into mitochondria (Extended Data Fig. 5a,b). The import retention of some sHSPs may therefore stem from their interaction with cytosolic binding partners. The observation that more HSPB8 is imported in cells expressing HSPB8-binding-deficient BAG3 (delta-IPV mutant) than wild-type BAG3 supports this notion (Extended Data Fig. 5c). Together, these biochemical fractionation experiments show that most sHSPs reside in the IMS, but interactions with binding partners may reduce their mitochondrial import.

To validate the IMS localization, we examined the mitochondrial location of HSPB1 using expansion microscopy (Extended Data Fig. 6a,b). This confirmed the localization of HSPB1 spots between TOM20 and HSP60, corresponding with the IMS (Fig. 1c and Extended Data Fig. 6c). Immuno-electron microscopy confirmed further that HSPB1 resides in the peripheral space of the IMS under basal conditions and is absent from the cristae and the matrix of mitochondria (Fig. 1d). Calculating local protein density revealed that HSPB1 was at least as densely populated, if not higher, in the mitochondrial IMS as in the cytosol (Fig. 1d). In conclusion, the microscopy data confirm the IMS localization of sHSPs.

**Fig. 1 | The mitochondrial IMS contains sHSPs. a,** HeLa and C2C12 cells were subjected to heat shock (42 °C) for 1 h. The mitochondrial fraction was separated from the cytosolic fraction, and the abundance of different sHSPs in each fraction was verified by immunoblotting. Tubulin was used as a marker for the cytosol and VDAC1 for mitochondria. Whole cell lysate (WCL) represents the NP-40 soluble fraction and was used as a reference. **b,** Submitochondrial localization of sHSPs was verified by subjecting intact mitochondria and mitoplasts (derived after osmotic swelling) to proteinase K (10 μg ml⁻¹) treatment. Matrix, mitochondrial matrix; *, lowest band is non-specific. **c,** Expansion microscopy of non-heat-shocked HeLa cells with immunofluorescence staining of endogenous HSPB1, HSP60 (matrix) and TOM20 (OM). Samples were fixed and stained before linkage in an expandable hydrogel, and imaged with confocal laser scanning microscopy. Zoomed area (3 μm × 3 μm) and line plot are represented. Indicated

distances are corrected for the expansion factor (4×). Scale bar, 2.0 μm and 0.25 μm (zoom). **d,** Representative electron micrograph of non-heat-shocked HeLa cells transduced with HSPB1-GFP and stained for anti-GFP. In three independent electron microscopy samples, 554–814 gold particles were scored on 14–16 images per sample and corrected for area. Local density estimations are presented (mean ± s.d.). Scale bar, 200 nm. **e,f,** Mitochondria were isolated from HeLa cells and treated with 1 M NaCl (e) or Na₂CO₃ at pH 11.5 (f). T, total; P, pellet; S, supernatant. **g,** Mitochondria, cytosolic fraction and WCL from HeLa cells were boiled in sample buffer with or without DTT to destroy or maintain disulphide bonds, respectively. The ratio of monomers to dimers was calculated by densitometry and corrected for loading with VDAC1. Results are representative of two (**c**) or three replicates (**a, b, d, e, f** and **g**). Source numerical data and unprocessed blots are available in source data.

We then performed a sodium chloride assay to explore the physical interactions of sHSPs in the IMS. We found that HSPB1, HSPB5 and HSPB8 were retained in the membrane-rich pellet fraction (Fig. 1e), indicating that sHSPs predominantly form hydrophobic interactions.

As this result suggests membrane association, we next performed a sodium carbonate assay to separate integral membrane proteins from membrane-associated and soluble proteins. As expected, HSPB1, HSPB5 and HSPB8 dissociated from the membrane at pH 11.5 and higher

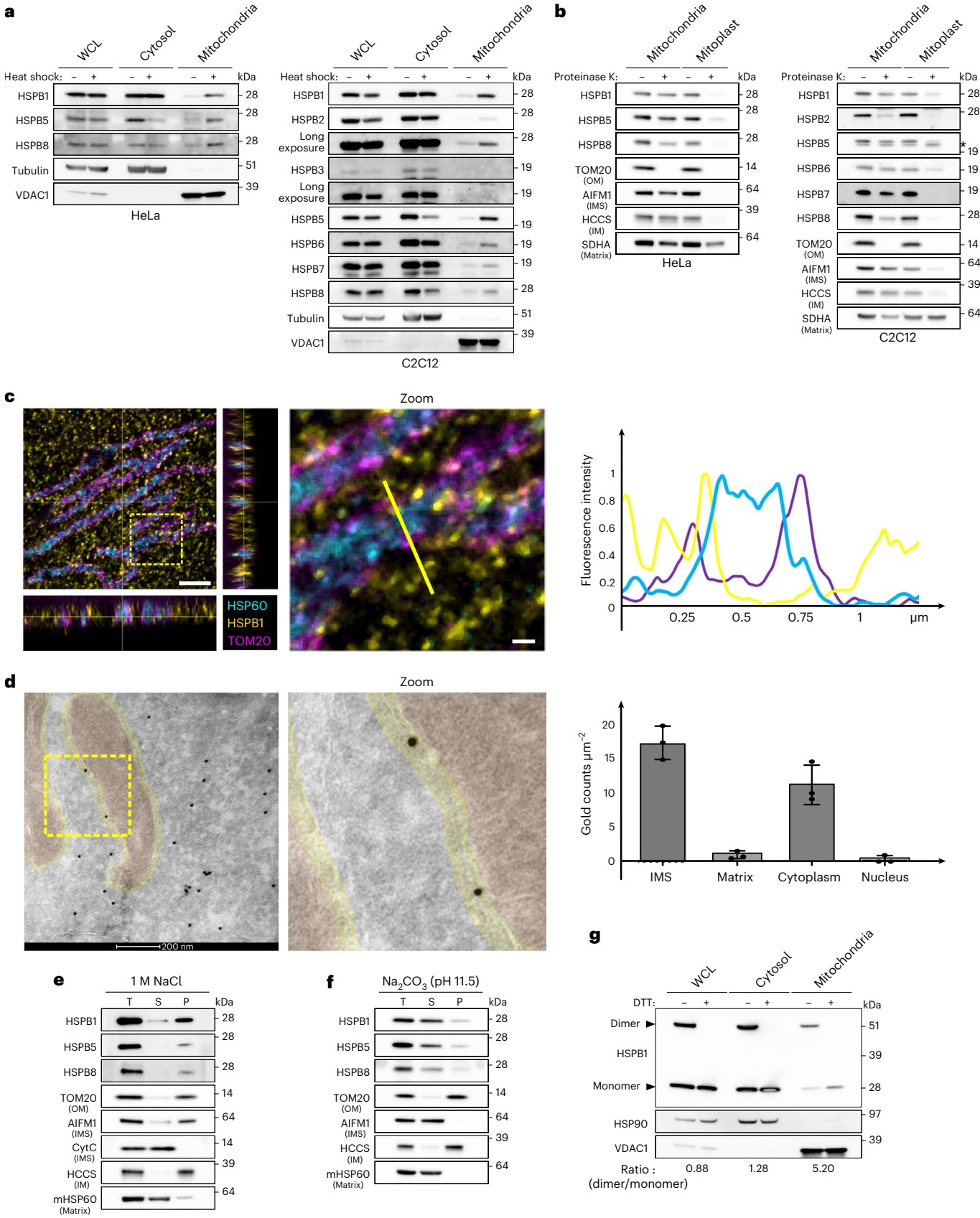

(Fig. 1f and Extended Data Fig. 7). This confirms that sHSPs could have some form of membrane association in the IMS but are not integral membrane proteins.

As HSPB1 forms dimers through a redox-sensitive disulphide bond, we verified its redox state by assessing the monomer/dimer ratio in the mitochondrial fraction (Fig. 1g). Whereas the cytosolic fraction has a near-equal ratio of monomers and dimers in HeLa cells, the mitochondrial fraction has a two- to fivefold enrichment of dimers over monomers, further indicating that mitochondrial HSPB1 resides in a compartment different from the cytosol.

Together, the results above demonstrate that sHSPs are present in the mitochondrial IMS, suggesting that these nuclear-encoded proteins translocate from the cytosol into mitochondria.

## Reconstitution of the mitochondrial import of sHSPs

To study this translocation in a reconstituted system, we performed in vitro import assays using rabbit reticulocyte lysates. The mitochondrial protein LIAS served as a positive control. A fraction of HSPB1 became resistant to proteinase K digestion, confirming the mitochondrial import of HSPB1 (Fig. 2a). The import was independent of the mitochondrial membrane potential, which is often observed for IMS-localized proteins[7]. The import assay also revealed a lower band for HSPB1, suggesting that HSPB1 might undergo processing inside mitochondria. Mass spectrometric analysis of this lower band revealed that mitochondrial HSPB1 had undergone an N-terminal modification (acetylation after removal of the N-terminal methionine), possibly explaining the size shift on gel (Fig. 2b). These data show, together with mitochondrial subfractionation and localization experiments, that sHSPs are imported into mitochondria, providing further evidence for a role of sHSPs in the IMS.

## Mitochondrial import of sHSPs is not dependent on phosphorylation

As it is well established that chaperone activity and oligomeric size of sHSPs are tightly linked to their phosphorylation status[56–64], we studied the relationship between sHSP phosphorylation and mitochondrial import.

We first evaluated whether mitochondrial HSPB1 is phosphorylated, and we used the heat shock condition as a positive control for elevated HSPB1 phosphorylation. We detected small amounts of phosphorylated HSPB1 in the mitochondrial fraction under basal conditions, which increased after heat shock (Fig. 2c). Note that, under heat shock conditions, not only the amount of phosphorylated HSPB1 increases in the mitochondrial fraction, but also the total amount of HSPB1 due to increased recruitment of sHSPs to the mitochondrial outer membrane. We then assessed whether phosphorylation drives import using phosphomimetic variants. Replacing all three serines (S15, S78 and

S82) with alanines prevented HSPB1 from becoming phosphorylated (HSPB1[3A]), while substitution by aspartates mimicked phosphorylated HSPB1 (HSPB1[3D]). We verified import of HSPB1[3A] and HSPB1[3D] with proteinase K assays and found that both HSPB1[3A] and HSPB1[3D] were still imported (Fig. 2d), and translocate to the mitochondrial IMS in equal amounts as wild-type HSPB1 (Fig. 2e). We next confirmed these observations for endogenous HSPB1 by pharmacologically blocking the mitogen-activated protein kinase (MAPK) pathway, known to target HSPB1. Despite the decrease in HSPB1-phosphorylation upon inhibition of the MAPK pathway, the total amount of mitochondrial HSPB1 remained unchanged (Fig. 2f), confirming that cytosolic phosphorylation of HSPB1 is not a prerequisite for mitochondrial import.

Together, this demonstrates that phosphorylation of HSPB1 is neither necessary nor promoting mitochondrial import.

## Depletion of sHSPs perturbs mitochondrial morphology and function

To assess the impact of the loss of sHSPs on mitochondria, we depleted sHSPs in HeLa cells with clustered regularly interspaced short palindromic repeats (CRISPR)/Cas9 (knockout) or short hairpin RNAs (shRNA) (knockdown). Depletion of individual sHSPs (HSPB1, HSPB5 and HSPB8) did not change mitochondrial morphology (Extended Data Fig. 8a). However, to exclude functional compensation between sHSPs, we tested whether depletion of all sHSPs would perturb mitochondrial function. To this end, we generated two cell lines with expression of each sHSP below 15% (referred to as HeLa[sHSPs−/−] lines) (Fig. 3a). We observed drastic morphological changes in both lines, as demonstrated by mitochondrial fragmentation and swellings (Fig. 3b and Extended Data Fig. 8a). In contrast, we found no obvious morphological changes for other organelles such as the endoplasmic reticulum and the Golgi network (Extended Data Fig. 8b,c), suggesting that mitochondria might be particularly vulnerable to the depletion sHSPs.

We examined the mitochondrial cristae structure in HeLa[sHSPs−/−] lines using transmission electron microscopy and found exclusively severely affected mitochondria with swelling of the IMS and missing or disorganized cristae (Fig. 3c and Extended Data Fig. 9). Morphology of other organelles appeared normal on electron microscopy, further indicating that this may be an organelle-specific effect.

To verify if morphological changes were accompanied by mitochondrial membrane potential decrease, we loaded HeLa[WT] and HeLa[sHSPs−/−] lines with tetramethylrhodamine ethyl ester (TMRE) and found that the membrane potential of the HeLa[sHSPs−/−] lines was unaltered (Fig. 3d). We then verified mitochondrial function by measuring mitochondrial respiration with a Seahorse assay, providing insight into the activity of the electron transport chain. Both HeLa[sHSPs−/−] lines showed severely compromised respiration, leading to a reduction in ATP production (Fig. 3e–h).

---

**Fig. 2 | Mitochondrial import of sHSPs is not dependent on phosphorylation or mitochondrial membrane potential. a**, Recombinant [35S]-labelled LIAS and HSPB1 were produced with rabbit reticulocyte lysate and incubated with mitochondria purified from Hela cells for the indicated times (min) in the presence or absence of a membrane potential (+CCCP) across the IM. Treatment with proteinase K (PK) led to the degradation of non-imported proteins, while treatment with Triton X-100 (TX) allowed the digestion of all soluble proteins, controlling for protease resistance due to protein aggregation. Samples were analysed by SDS–PAGE followed by autoradiography or immunoblotting. TOM20 is a marker for the OM, CHCHD4 for the IMS and SLP2 for the IM. 10% represents the input fraction of recombinantly produced proteins. **b**, HSPB1 was immuno-precipitated from purified mitochondria using the V5-tag, after treatment with proteinase K to digest all accessible (non-imported) HSPB1. Bands were cut from a Coomassie-stained gel and analysed by LC–MS/MS, which revealed the removal of the N-terminal methionine and the addition of acetyl to the N-terminal peptide. **c**, HeLa cells were subjected to heat shock for 1 h. Phosphorylation of HSPB1 was determined with phospho-specific antibodies

directed towards Ser15, Ser78 and Ser82. **d**, Submitochondrial localization of sHSPs was verified by subjecting intact mitochondria and mitoplasts (derived after osmotic swelling) to proteinase K (10 μg ml⁻¹) treatment. **e**, Phosphorylation variants were transfected in HeLa cells and analysed for their mitochondrial import. Substitution of all three phosphorylation residues with alanine (HSPB1[3A]) or aspartate (HSPB1[3D]) represents non-phosphorylated and phosphorylated HSPB1, respectively. Proteinase K (10 μg ml⁻¹) was added to verify if the proteins were imported. Densitometric analysis of the bands is represented (mean ± s.d.) (n = 3 biologically independent experiments). One-way ANOVA with Dunnett's multiple comparison test was performed. NS, non-significant. **f**, Inhibition of the MAPK pathway with SB203580 (10 μM) in HeLa cells. Phosphorylation of HSPB1 was determined with phospho-specific antibodies. Densitometric analysis was performed for the mitochondrial fraction and presented in the graph (mean ± s.d.) (n = 3 biologically independent experiments). Two-tailed unpaired Student's t-test was performed. Results are representative of two (**a**) or three replicates (**c**, **d**, **e** and **f**). Source numerical data, including exact P values, and unprocessed blots are available in source data.

Together, these experiments show that depletion of sHSPs leads to drastic changes in mitochondrial morphology and function while retaining the membrane potential. Other organelles appeared unchanged, suggesting that mitochondria are particularly affected by the depletion of sHSPs.

## sHSPs counter protein aggregation in the mitochondrial IMS

Now that we established that sHSPs are imported into mitochondria and that depletion of sHSPs causes mitochondrial dysfunction, we sought to identify the role of sHSPs in the IMS. In the cytosol, sHSPs rapidly recognize and prevent misfolding proteins from further misfolding,

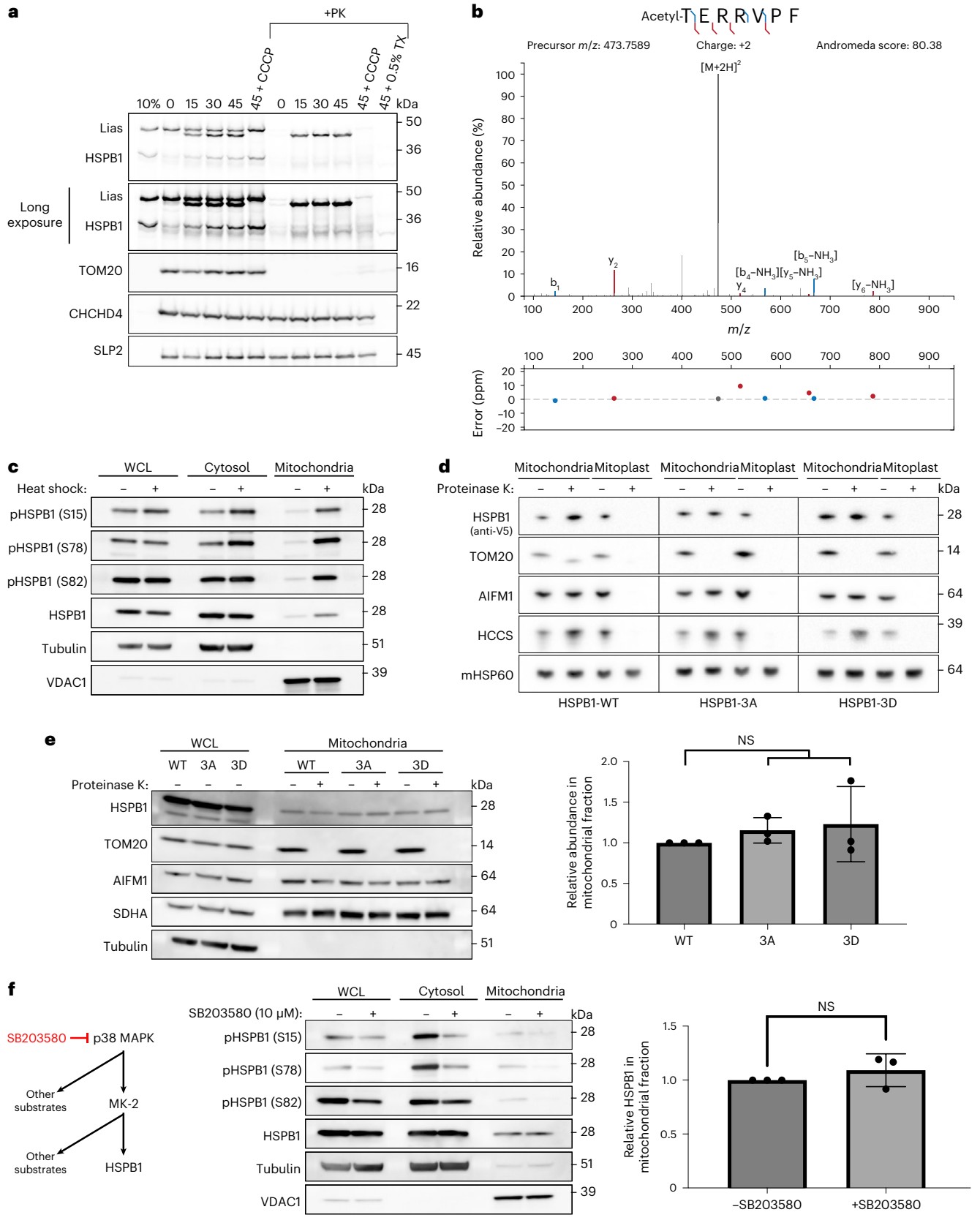

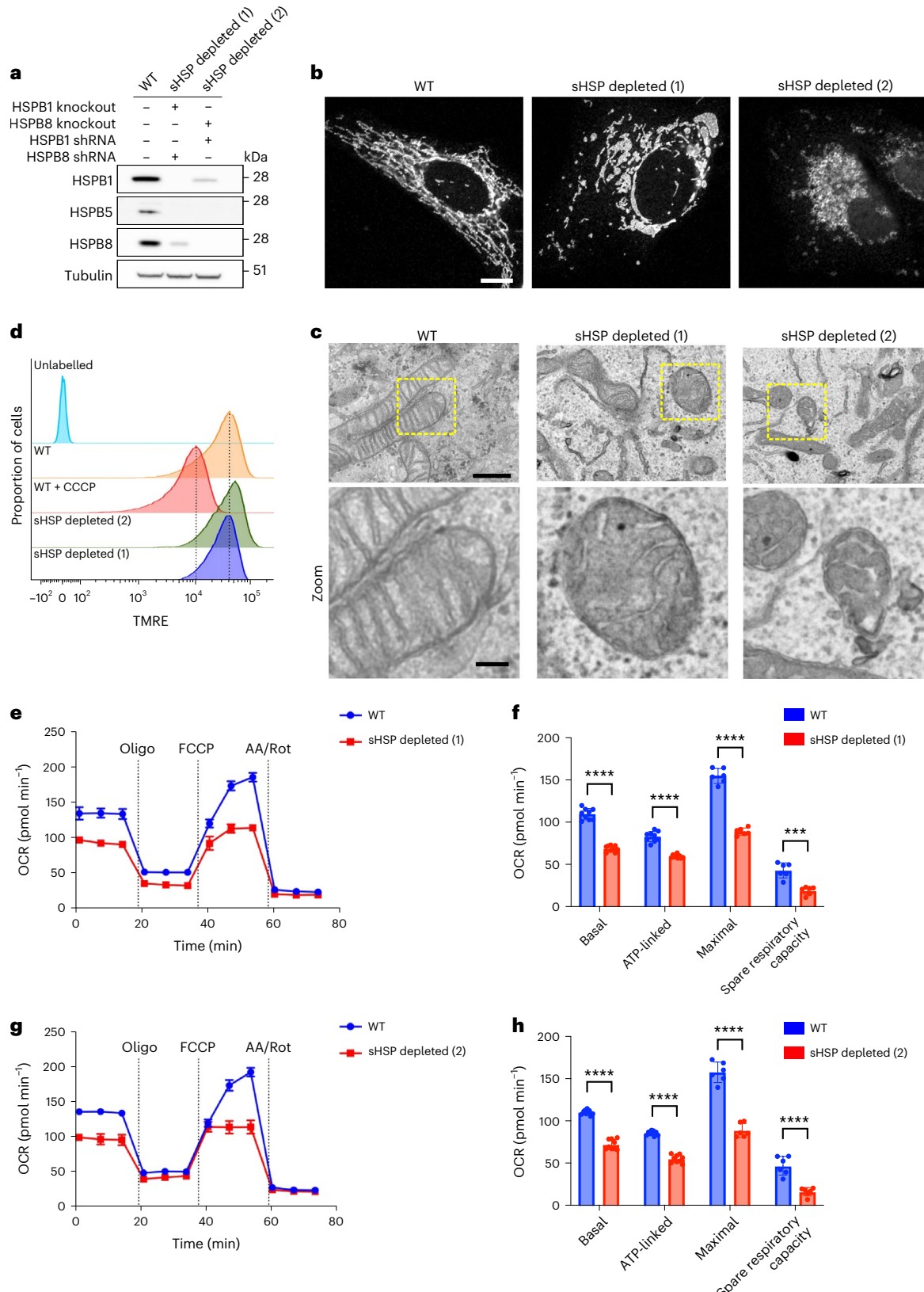

so we hypothesized that sHSPs also counter protein aggregation in the mitochondrial IMS. We transfected HeLa cells with mutant forms of superoxide dismutase 1 (SOD1) and endonuclease G (EndoG), two proteins that are known to translocate to the IMS (Extended Data Fig. 10) and of which aggregating variants have been described[65–71]. Overexpression of SOD1-G93A or EndoG-N174A resulted in increased HSPB1

levels in the mitochondrial fraction (Fig. 4a,b). Note that the reduced signal for SOD1-G93A is due to its tendency to aggregate, and only the soluble fraction is displayed here. To assess whether the HSPB1 response was specific to protein aggregation in the IMS, we transfected mutant ornithine transcarbamylase (ΔOTC), which aggregates in the mitochondrial matrix[72]. HSPB1 did not accumulate in the mitochondrial

**Fig. 3 | Depletion of sHSPs disrupts mitochondrial morphology and function. a**, HeLa cells were depleted from sHSPs with a combination of CRISPR/Cas9 knockout and shRNA. Proteins were extracted and subjected to western blot. Immunoblotting shows the level of depletion for each of the respective sHSPs and tubulin is used as loading control. Note that we did not have to use HSPB5 shRNAs, as the HSPB5 expression decreased spontaneously in the absence of HSPB1 and HSPB8. **b**, The mitochondrial network of control and sHSP-depleted HeLa cells visualized by anti-TOM20 immunostaining and confocal microscopy. Representative cells are shown from more than three replicates. Scale bar, 10 µm. **c**, Mitochondrial ultrastructure of control and HSP-depleted HeLa cells visualized by transmission electron microscopy. Scale bar, 1.0 µm and 0.25 µm (zoom). **d**, Control or sHSP-depleted HeLa cells were incubated with TMRE for 30 min at 37 °C. After washing, the cells were collected and analysed by flow cytometry.

Histograms representing TMRE signals for each of the respective cell lines are displayed. As a negative control, the membrane potential was dissipated by CCCP treatment. **e–h**, Mitochondrial respiration measurement with Seahorse. Oxygen consumption rates (OCR) were continuously measured in control and sHSP-depleted HeLa cells (**e** and **g**). Oligomycin (1 µM), FCCP (2 µM) and Antimycin **A** (AA, 0.5 µM)/Rotenone (Rot, 0.5 µM) were added at the indicated timepoints. Data are presented as mean ± s.d. (*n* = 3 biologically independent experiments). The parameters indicated in **f** and **h** were calculated from the measurement in **e** and **g**. Basal respiration, ATP-linked respiration, maximal OCR and spare respiratory capacity are presented as mean ± s.d. (*n* = 3 biologically independent experiments). Two-tailed unpaired Student's *t*-tests were performed. ****P* < 0.0001. Source numerical data, including exact *P* values, and unprocessed blots are available in source data.

fraction following protein aggregation in the mitochondrial matrix (Fig. 4c), suggesting that sHSPs respond specifically to protein aggregation in the IMS.

To assess whether sHSPs can prevent protein aggregation, we overexpressed HSPB1, HSPB5, and HSPB8 in cells expressing wild-type or mutant SOD1. While wild-type SOD1 was mostly soluble, mutant SOD1-G93A accumulated in the insoluble fraction, which was countered by overexpression of sHSPs (Fig. 4d). To verify whether sHSPs are also required to counter protein aggregation in the IMS, we repeated this experiment in HeLa^WT and HeLa^sHSPs−/− cells. This revealed that depletion of sHSPs leads to more insoluble SOD1-G93A (Fig. 4e), suggesting further that sHSPs counter protein aggregation in the IMS.

While these results indicate that sHSPs recognize overexpressed mutant IMS proteins, we wanted to verify whether sHSPs would also respond to more physiological substrates. Inspired by the observation that HSPB1 is the most upregulated protein upon YME1L deletion in mouse retinas[50], we evaluated sHSP recruitment to mitochondria upon depletion of YME1L and PARL. Indeed, depletion of these mitochondrial proteases, which process mitochondrial IMS and inner membrane (IM) substrates, resulted in increased recruitment of HSPB1 and HSPB8 (Fig. 4f). Additional depletion of OMA1 did not lead to the further enrichment of HSPB1. Depletion of ClpB, an IMS-localized chaperone, also led to higher levels of HSPB1 in the mitochondrial fraction (Fig. 4g).

Together, these results support the notion that impaired mitochondrial proteostasis in the IMS leads to the activation of sHSPs.

## sHSPs chaperone mitochondrial transmembrane proteins

As we found that sHSPs prevent aggregation of model clients, we sought to identify the endogenous substrates of sHSPs in the mitochondrial IMS. Due to the transient nature of chaperone–client complexes, it has proven challenging to identify molecular clients handled by sHSPs.

However, we showed previously that some CMT-neuropathy-associated mutations in HSPB1 increase the affinity for substrates and lead to stable chaperone–client complexes[73,74]. We therefore took advantage of these molecular trap mutants to identify substrates handled by HSPB1. We overexpressed mutant HSPB1^S135F in HeLa cells and determined its interactome (Fig. 5a). The co-immunoprecipitation was performed on whole cell lysates (WCLs), and the obtained interactome showed enrichment of mitochondrial proteins over other cellular organelles. Validation of these interactions revealed that only a small fraction of mitochondrial proteins was bound to HSPB1 (Fig. 5b). This suggests that HSPB1 is not a constitutive interactor but deals with a small subset of the proteins, hinting at a chaperone–client interaction.

Among the mitochondrial interactors, we found many mitochondrial solute carriers. As solute carriers undergo post-translational import and their transmembrane domains need shielding in the cytosol, we asked whether HSPB1 bound the solute carriers in the cytosol or inside mitochondria. We repeated the co-immunoprecipitation experiments on mitochondria pre-treated with proteinase K to retain only interactions within mitochondria. The HSPB1–client interactions remained, indicating that the interactions occur inside mitochondria (Fig. 5c). We ruled out that interactions were mutant specific by demonstrating that wild-type HSPB1 also binds to SLC25A12 (Fig. 6d).

To identify where HSPB1 binds the solute carriers, we generated deletion variants of SLC25A12. This revealed that HSPB1 binds to the hydrophobic transmembrane domain (Fig. 5e), which would not be accessible when SLC25A12 is appropriately folded and incorporated into the membrane, suggesting that HSPB1 binds the solute carrier in an unfolded state.

As our experiments above revealed that sHSPs respond to protein misfolding in the mitochondrial IMS, we tested whether destabilising mitochondrial solute carriers also increased mitochondrial

**Fig. 4 | sHSPs respond to the aggregation of client proteins in the mitochondrial IMS. a–c**, HeLa cells were transiently transfected with SOD1 wild-type (WT) or G93A (**a**), EndoG wild-type (WT) or N174A (**b**), OTC wild-type (WT) or delta-OTC (deleted 30–114 aa) (**c**). Samples were analysed by SDS–PAGE followed by immunoblotting using anti-HSPB1, anti-GFP (SOD1 or EndoG), anti-Tubulin (cytosolic marker) or anti-VDAC1 (mitochondrial marker) antibodies. Note that only the soluble fractions are displayed, and the reduced signal for SOD1-G93A is due to its insolubility. **d**, Cells were transfected as in **a** in wild-type or HSPB1/HSPB5/HSPB8-overexpressing HeLa cells. Isolated mitochondria were lysed in NP-40-containing lysis buffer, and the soluble fraction was separated from the non-soluble fraction by centrifugation. Samples were analysed by SDS–PAGE followed by immunoblotting using anti-V5 (for HSPB1/HSPB5/HSPB8) and anti-GFP (SOD1) antibodies. **e**, Protein aggregation assay in mitochondria isolated from transiently transfected HeLa cells with wild-type (WT) or G93A-mutant SOD1. Isolated mitochondria were lysed in NP-40-containing lysis buffer, and the soluble fraction was separated from the non-soluble fraction by centrifugation. Samples were analysed by SDS–PAGE followed by immunoblotting using anti-

GFP (SOD1), anti-HSPB1, anti-HSPB5, anti-HSPB8 and anti-mHSP60 (marker for soluble mitochondrial fraction) antibodies. The amount of insoluble SOD1 was quantified using the anti-GFP signal from the non-soluble fraction, and data are shown as mean ± s.d. from three independent experiments. One-way ANOVA with Dunnett's multiple comparisons test was performed. **P* < 0.05, ***P* < 0.005. **f**, HeLa cells were depleted from YME1L, PARL and OMA1 with shRNA. Mitochondria were isolated and compared with the cytosolic or the NP-40-soluble WCL fraction. Samples were analysed by SDS–PAGE followed by immunoblotting using anti-HSPB1, anti-HSPB8, anti-YME1L, anti-PARL, anti-Tubulin (cytosolic marker) or anti-VDAC1 (mitochondrial marker) antibodies. **g**, HeLa cells were depleted from ClpB with two different shRNA. Mitochondria were isolated and compared with the cytosolic or the NP-40-soluble WCL fraction. Samples were analysed by SDS–PAGE followed by immunoblotting using anti-HSPB1, anti-ClpB, anti-Tubulin (cytosolic marker) or anti-VDAC1 (mitochondrial marker) antibodies. Results are representative of two (**d**, **f** and **g**) or three replicates (**a**, **b**, **c** and **e**). Source numerical data, including exact *P* values, and unprocessed blots are available in source data.

sHSP levels. We used HEK293T Flp-In cells lacking the phospholipid transacylase Tafazzin, which catalyses a crucial step in cardiolipin biosynthesis, a lipid required to stabilize the structure of mitochondrial solute carriers[75–79]. This resulted in more mitochondrial HSPB1 in both Tafazzin-knockout clones (Fig. 5f), even in a clone with very low HSPB1 expression.

Together, this shows that mitochondrial substrates for sHSPs are enriched by transmembrane proteins, whose folding state is under the surveillance of sHSPs.

## CMT mutations impair the mitochondrial function of sHSPs

Mutations in HSPB1 and HSPB8 cause CMT neuropathy, and mitochondrial defects have been observed in CMT disease models[27,29,35,36]. We hypothesized that these defects could in part result from dysfunctional sHSPs inside mitochondria, and studied two mutations in the conserved alpha-crystallin domain (HSPB1[R127W] and HSPB1[S135F]) and the C-terminal mutant (HSPB1[P182L]), which causes the most severe phenotype reported so far[80]. Note that the P182L mutation has reduced solubility in the cytosol, explaining reduced levels observed by western blot.

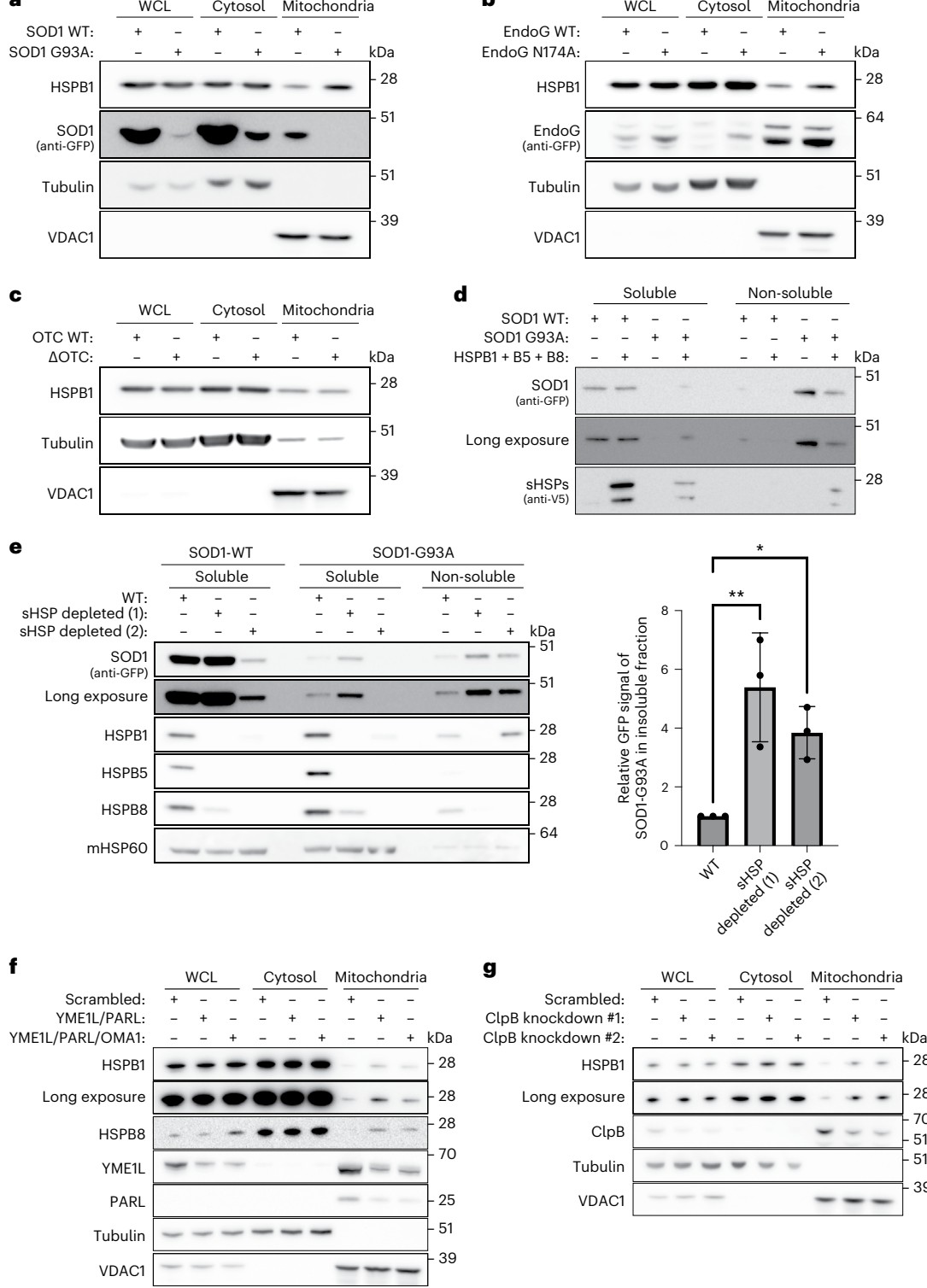

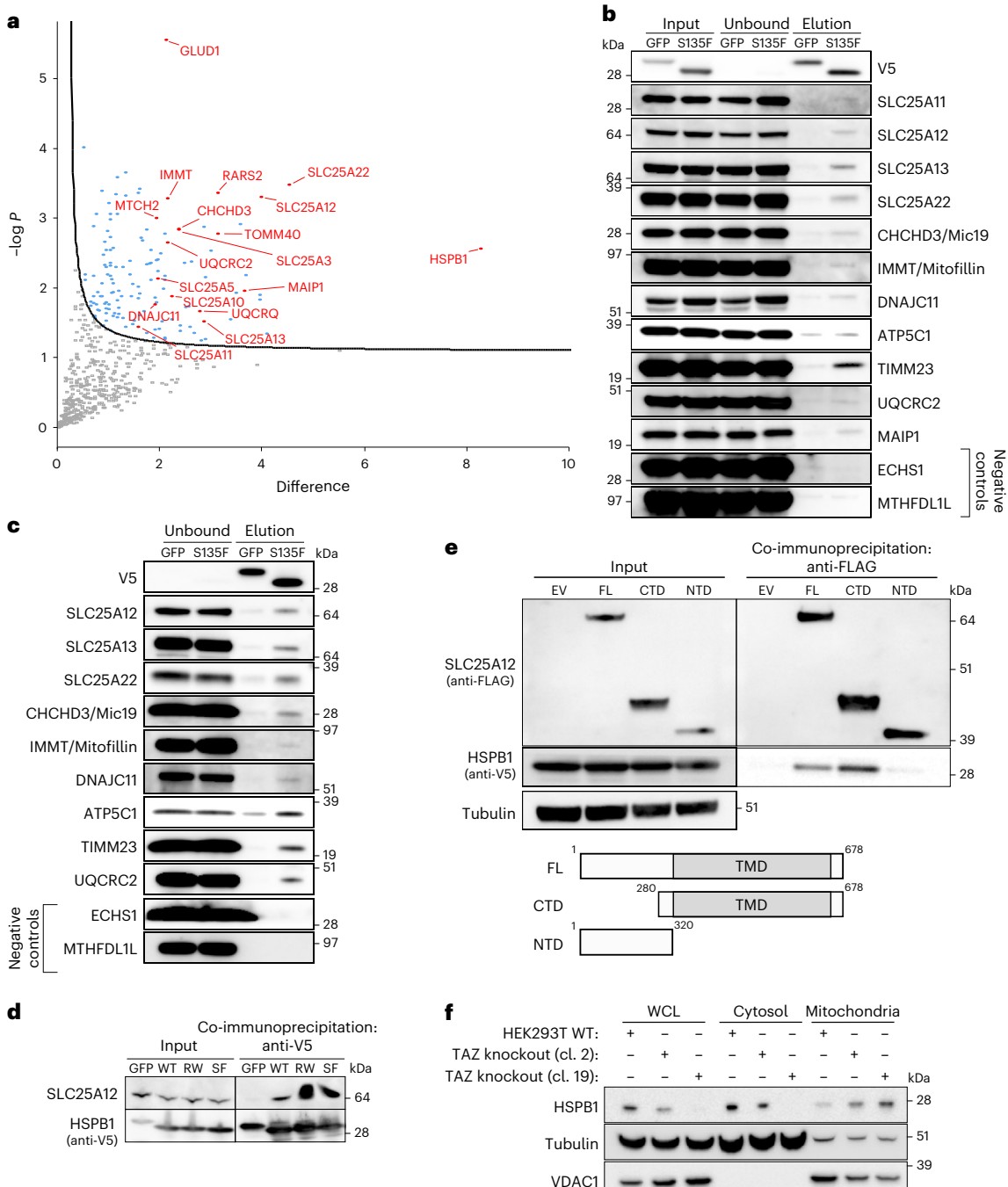

**Fig. 5 | The mitochondrial interactome of HSPB1 is enriched for transmembrane proteins of the inner mitochondrial membrane. a**, Volcano plot of the interactome of HSPB1$^{S135F}$ versus eGFP obtained from WCL analysis. Mitochondrial proteins are highlighted in red. A one-sided *t*-test was performed for pairwise comparison of both conditions. Significantly enriched proteins are indicated by curved line, set by an FDR value of 0.05 and $S_0$ value of 1. **b**, Co-immunoprecipitation with anti-V5 beads to validate the mitochondrial interactors in WCL. **c**, Co-immunoprecipitation with anti-V5 beads to validate the mitochondrial interactors in mitochondria that were pre-treated with proteinase K (10 μg ml$^{-1}$) to remove all non-imported proteins. **d**, Co-immunoprecipitation with anti-V5 beads to validate the interaction between V5-tagged HSPB1 wild-type (WT), HSPB1 mutants R127W (RW), S135F (SF) or P182L (PL) and SLC25A12. GFP-V5 was used as negative control. **e**, Determination of the binding sites of HSPB1

on SLC25A12. Immunoprecipitation of SLC25A12 using anti-FLAG beads to pull down SLC25A12-3xFLAG (full-length (FL), N-terminal domain (NTD) or C-terminal domain (CTD)) and assessed for the co-immunoprecipitation of HSPB1$^{S135F}$. Samples were analysed by SDS–PAGE followed by immunoblotting using anti-FLAG (SLC25A12), anti-V5 (HSPB1$^{S135F}$) and anti-tubulin (loading control) antibodies. Schematic representation of SLC25A12 with FL, NTD or CTD. The transmembrane domain is depicted in grey. **f**, HEK293 Flp-In cells were depleted from Tafazzin (TAZ) as previously described[75]. cl., clone. Mitochondria were isolated and compared with the cytosolic or the NP-40-soluble WCL fraction. Samples were analysed by SDS–PAGE followed by immunoblotting using anti-HSPB1, anti-Tubulin (cytosolic marker) or anti-VDAC1 (mitochondrial marker) antibodies. Results are representative of two (**d**) or three replicates (**b**, **c**, **e** and **f**). Source numerical data and unprocessed blots are available in source data.

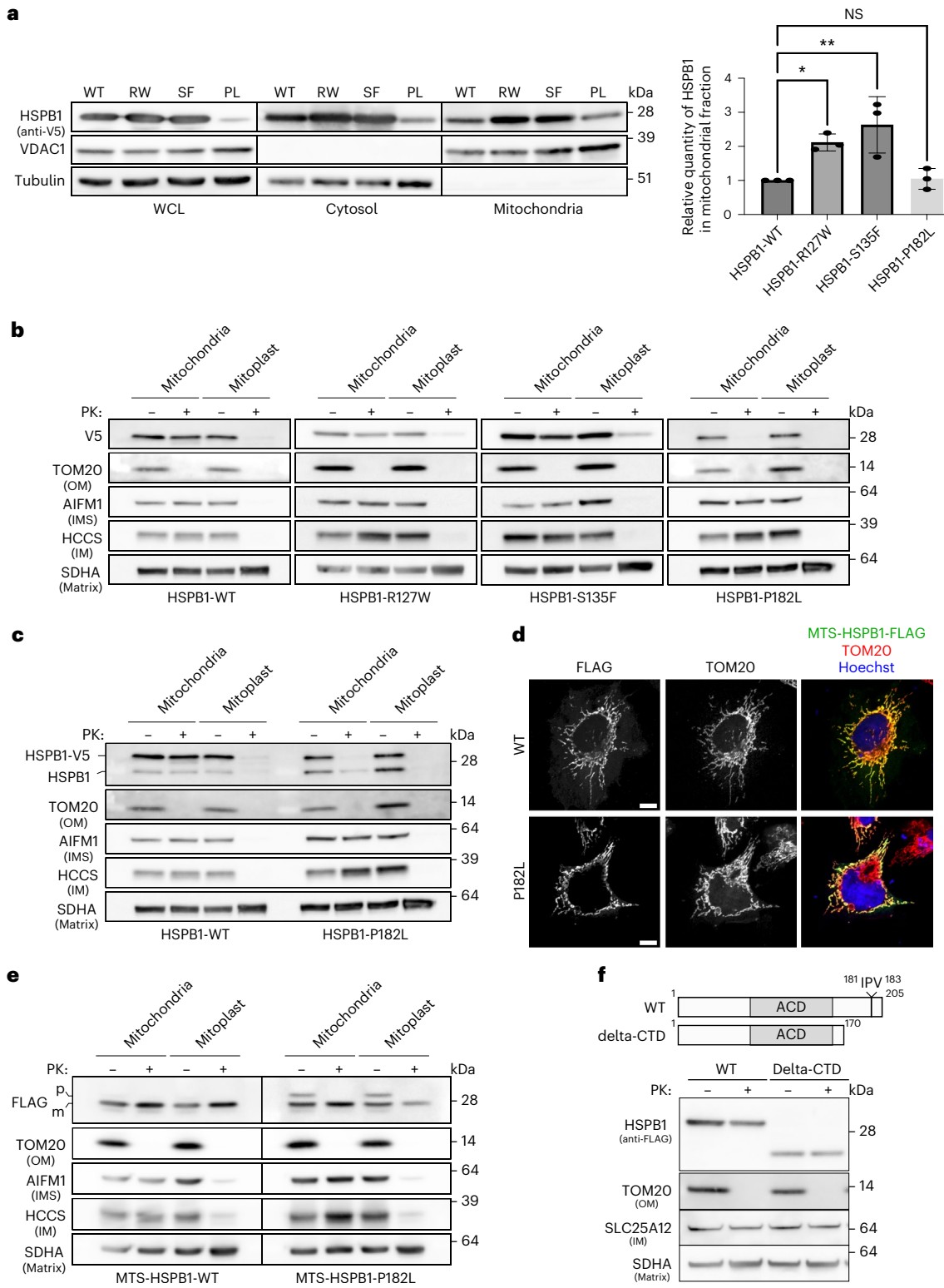

The amount of HSPB1 in the complete mitochondrial fraction was increased for the two alpha-crystallin mutants, HSPB1[R127W] and HSPB1[S135F], but unaltered for the C-terminal mutant (Fig. 6a). Protein-ase K protection assays to determine the submitochondrial localization demonstrated that the alpha-crystallin HSPB1[R127W] and HSPB1[S135F] mutants still resided in the mitochondrial IMS (Fig. 6b). However, the HSPB1[P182L] accumulated at the OM (Fig. 6b). We hypothesized that this import defect resulted from the extremely large oligomers formed by this mutant[81], precluding mitochondrial import. If so, then endogenous

wild-type HSPB1, incorporated into these oligomers[81], would also show import retention. Indeed, import of endogenous HSPB1 was reduced in the presence of HSPB1[P182L] (Fig. 6c). The large oligomeric structures formed by the P182L mutant, in which endogenous HSPB1[WT] becomes incorporated, may fail to release monomeric subunits required for mitochondrial import.

To test this hypothesis, we fused HSPB1 to the mitochondrial targeting sequence (MTS) from Cytochrome C oxidase subunit 8A (COX8A) (MTS_COX8A-HSPB1, Fig. 6d). The MTS_COX8A is sufficient to import

**Fig. 6 | CMT disease-causing mutations in HSPB1 disturb its mitochondrial function. a**, Mitochondria were isolated from HeLa cells overexpressing HSPB1 wild-type (WT), HSPB1 mutants R127W (RW), S135F (SF) or P182L (PL) and compared to the cytosol and the NP-40 soluble WCL. Note that the reduced expression of P182L in the WCL is due to reduced solubility in the NP-40-containing buffer. Samples were analysed by SDS–PAGE followed by immunoblotting using anti-V5 (HSPB1), anti-tubulin (cytosolic marker) and anti-VDAC1 (mitochondrial marker) antibodies. The graph represents the densitometric analysis of the mitochondrial fraction after correction for loading using VDAC1 (mean ± s.d.) (*n* = 3 biologically independent experiments). One-way ANOVA with Dunnett's multiple comparison test was performed. *$P < 0.05$, **$P < 0.01$. NS, non-significant. **b**, Submitochondrial localization of HSPB1-V5 wild-type (WT) and HSPB1 mutants R127W, S135F and P182L was verified by subjecting intact mitochondria and mitoplasts (derived after osmotic swelling) to proteinase K (PK, 10 μg ml⁻¹) treatment. Samples were analysed by SDS–PAGE followed by immunoblotting using anti-V5 (HSPB1), anti-TOM20 (OM), AIFM1 (IMS), HCCS (IM) and SDHA (matrix) antibodies. **c**, The same samples as in **b**

were analysed with an anti-HSPB1 antibody to detect both endogenous and exogenous HSPB1. **d**, A fusion protein between HSPB1 and the MTS of COX8A was expressed in HeLa cells. Immunostaining with anti-FLAG (MTS_COX8a-HSPB1) and anti-TOM20 (mitochondrial marker). Scale bar, 10 μm. **e**, Submitochondrial localization of wild-type or P182L-mutant MTS_COX8a-HSPB1 was verified as in **b**. Samples were analysed by SDS–PAGE followed by immunoblotting using anti-FLAG (MTS_COX8a-HSPB1), anti-TOM20 (OM), anti-AIFM1 (IMS), anti-HCCS (IM) and anti-SDHA (matrix) antibodies. Precursor (p) and mature (m) proteins were separated by size as the N-terminal MTS was cleaved off after mitochondrial import. **f**, Proteinase K protection assay of mitochondria isolated from HeLa cells expressing full-length HSPB1 or C-terminally truncated HSPB1 (delta-CTD). Samples were analysed by SDS–PAGE followed by immunoblotting using anti-FLAG (HSPB1), anti-TOM20 (OM), SLC25A12 (IM) and SDHA (matrix) antibodies. The IPV motif is a conserved C-terminal motif, disrupted by the P182L mutation, and further described in ref. [81]. Results are representative of three replicates (**a**–**f**). Source numerical data, including exact *P* values, and unprocessed blots are available in source data.

the fusion protein constitutively into the mitochondrial matrix. Upon import, the targeting sequence is cleaved, and the mature form can be distinguished from the precursor by size. A proteinase K assay confirmed that MTS_COX8A-HSPB1^WT was imported into the mitochondrial matrix (Fig. 6e). However, the MTS_COX8A-HSPB1^P182L variant displayed remnants of the pre-imported fusion protein, suggesting that HSPB1^P182L has acquired biophysical properties, like increased oligomer size, that limit its mitochondrial import. We ruled out that these results stemmed from the inactivation of an import motif by the P182L mutation, with a C-terminal deletion variant being imported at equal rates as the wild-type protein (Fig. 6f).

Together, these data show that mutations in HSPB1 can alter mitochondrial import, suggesting that the mitochondrial phenotypes observed in disease models for CMT2F could stem, at least partially, from a disturbed mitochondrial function.

## Discussion

Mitochondria are complex organelles composed of two lipid membranes and two aqueous compartments, each harbouring their own proteostasis machinery. Here we reveal that mammalian sHSPs can translocate to the mitochondrial IMS, where they form an effective chaperone system.

The recruitment and mitochondrial import of sHSPs seems an evolutionarily conserved feature. Various types of plants but also *D. melanogaster*, *Caenorhabditis elegans* and *T. gondii* express mitochondria-specific sHSPs[25,47,48,82–85]. In *Arabidopsis thaliana*, depletion of all three mitochondrial sHSPs leads to defects in growth and development[86]. These phenotypes were observed only after simultaneous loss of all three sHSPs, revealing molecular compensation by the sHSPs, similar to what we observed in human cells.

For many IMS proteins, MTSs have not yet been identified[87]. Others rely on the Mia40 import pathway, which requires a conserved Cx9C motif[15–19]. However, sHSPs contain only up to one cysteine, except for HSPB10, and their mitochondrial import may therefore occur independent from Mia40. In plants and *D. melanogaster*, mitochondrial import of sHSPs relies on an N-terminal MTS[47,85]. However, these specific sHSPs are exclusively mitochondrial and reside often in the matrix, whereas mammalian sHSPs are subject to a dual localization. Elucidating whether mammalian sHSPs harbour an MTS is complicated by the fact that all three protein domains are involved in the formation of secondary, tertiary and quaternary structures, in ways that are incompletely understood. With limited information on how mutations or deletion variants would affect the dynamic ensemble of sHSP oligomers, it will be a challenging task to verify if and where mammalian sHSPs have an MTS.

Our results support a model in which sHSPs localize to the IMS to contribute to mitochondrial proteostasis, and disruption of the IMS proteostasis leads to increased recruitment of sHSPs. A limitation of our study is that, while all these data are consistent with a chaperone role for sHSPs in the IMS, these results were obtained by expressing mutant forms of model substrates or interfering with mitochondrial proteases and lipids. For the mitochondrial substrates identified with the molecular trap variant, we found no evidence for their accumulation in the absence of sHSPs, which might be due to their rapid turnover by potent mitochondrial proteases. Moreover, future work would be required to decipher if and how sHSPs cooperate with Skd3/ClpB in the IMS.

Our work also provides insights into CMT disease, the most common inherited neurodegenerative disorder of the peripheral nervous system. While it remained enigmatic how cytosolic chaperones HSPB1 and HSPB8 caused mitochondrial deficits in CMT, our work now provides a plausible explanation. The most severe disease-causing HSPB1-P182L mutation leads to drastically reduced mitochondrial import. Other mutants, with milder phenotypes, are still imported but may interfere with the chaperone function through a dominant negative gain-of-function mechanism, as we previously identified for HSPB1 in the cytosol[73,74]. Our work therefore suggests that mutations in HSPB1 and HSPB8 may result in protein quality control defects in both the cytosol and mitochondria.

In summary, our study shows that mammalian sHSPs are imported into the mitochondrial IMS, where they assist in countering protein aggregation and thereby form part of a wider surveillance network that protects the IMS against protein aggregation. This function is impaired by CMT mutations, potentially explaining the previously observed mitochondrial dysfunction.

## Online content

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

## Methods

### Ethical approval

Animal procedures complied with all relevant ethical regulations and were carried out in accordance with European, national and institutional guidelines and with approval by the Medical Ethics Committee (20/36/461) and Ethical Committee for Animal Experiments (2022-13) of the University of Antwerp, Belgium. Primary human-derived lines (lymphoblasts and fibroblasts) were obtained from healthy volunteers, with their informed consent, according to protocols approved by the local Medical Ethics Committee (University of Antwerp, Belgium).

### Plasmid origin and vector construction

For each of the vectors, we cloned complementary DNA sequences into the respective plasmid backbones: p3xFLAG-CMV-14 (Sigma-Aldrich), pSP64-Poly(A) (P1241, Promega), pDONR221 and pLenti6/V5 (Life Technologies). Untagged sHSPs HSPB1 to HSPB10 (Frt-HSPBx, Addgene plasmids #63092–#63101, Addgene) were a kind gift from Harm Kampinga (University of Groningen). Full-length OTC and ΔOTC (lacking amino acids 30–114) were a kind gift from Nicholas Hoogenraad (plasmids #71877 and #71878, Addgene), MTS from Cox8a was subcloned from pmEmerald-Mito-7 (plasmid #54160, Addgene), SOD1 wild-type vector was previously described[33] and used to generate the SOD1-G93A mutant, EndoG wild-type and N174A mutant were a kind gift from Doris Germain (Mount Sinai School of Medicine, New York). BAG3 wild-type and IPV-mutant (I96V/V98G and I208G/V210G) were a kind gift from Josée N. Lavoie (University Laval, Quebec). shRNAs of the Sigma Mission pLKO.1-library were obtained from the BCCM/LMBP Plasmid collection (https://bccm.belspo.be). We used shRNAs against HSPB1 (TRCN0000011466), HSPB5/CRYAB (TRCN0000010822), HSPB8 (home-made 3′ untranslated region), ClpB (TRCN0000043564 and TRCN0000043566), YME1L (TRCN0000073867), PARL (TRCN0000048618) and OMA1 (TRCN0000062121).

Open reading frames were subcloned using restriction enzymes (Fastdigest, Thermo Fisher Scientific), Gateway cloning (Thermo Fisher Scientific) or in vivo assembly cloning[88]. All polymerase chain reaction (PCR) amplification reactions were performed with the Kapa HiFi HotStart PCR kit (07958897001, Roche Diagnostics Belgium).

For restriction enzyme cloning, we PCR amplified our gene of interest with restriction enzyme recognition sites appended to the forward and reverse primers. The PCR amplicons were purified using the QIAQuick PCR purification kit (28104, Qiagen) before the purified amplicon and vector backbone were digested by restriction enzymes (Fastdigest, Thermo Fisher Scientific) in separate tubes. FastAP Thermosensitive Alkaline Phosphatase was added to the vector reaction to prevent re-annealing (EF0654, Thermo Fisher Scientific). Vector and insert were purified with the QIAQuick PCR purification kit (28104, Qiagen) before being pooled in a 1:3 molar ratio and ligated with T7 ligase (EZ L6020L, Enzymatics) for 1 h at room temperature. Residual non-ligated DNA was digested with PlasmidSafe exonuclease (E3101K, Epicentre) for 30 min at 37 °C and inactivated for 30 min at 70 °C. Constructs were transformed into DG-1 chemo-competent bacteria (GE-DG1C-96, Eurogentec), plated on agar plates and sequence verified with Sanger sequencing.

For Gateway cloning, we followed the companies' protocol using BP- and LR-reaction enzymes (11789020 and 11791020, Thermo Fisher Scientific, Waltham) and pDONR221 and pLENTI6/V5 vectors (Thermo Fisher Scientific).

For in vivo assembly cloning[88], we PCR-amplified both fragments with 20 nucleotide overhang primers (homologous to the other primer pair) for 18 amplification cycles and with 1 ng template DNA. After DpnI digestion (FD1704, Thermo Fisher Scientific), samples were co-transformed into DG-1 chemo-competent bacteria (GE-DG1C-96, Eurogentec) after which the recombinant plasmids were sequence verified with Sanger sequencing.

For in vitro mutagenesis, we performed PCR amplification of the plasmid using primers that contained 20 nucleotides overhang on either side of the point mutation. In case of multiple mutations, we counted 20 nucleotides from the most distal point mutations. After PCR amplification, template DNA was digested with DpnI (FD1704, Thermo Fisher Scientific). PCR products were transformed into DG-1 chemo-competent bacteria (GE-DG1C-96, Eurogentec) after which the mutant fragments were sequence verified with Sanger sequencing.

### Cell lines and cell culture

All cell lines (HeLa, HEK293T, NSC34, Neuro-2a, SH-SY5Y, COS1, A498 and C2C12) were acquired from the American Type Culture Collection. Primary human-derived lines (lymphoblasts and fibroblasts) were obtained from healthy volunteers, with their informed consent, according to protocols approved by the local Medical Ethics Committee (University of Antwerp, Belgium). HeLa cells were grown in MEM medium supplemented with 10% (v/v) foetal bovine serum, 1% (v/v) L-glutamine, and 1% (v/v) penicillin–streptomycin (Life Technologies). Neuro-2A neuronal crest derived and SH-SY5Y neuroblastoma cells were grown in MEM medium supplemented with 10% (v/v) foetal bovine serum, 1% (v/v) L-glutamine, 1% (v/v) non-essential amino acids and 1% (v/v) penicillin–streptomycin (Life Technologies). A498 renal carcinoma cells were grown in MEM medium supplemented with 10% (v/v) foetal bovine serum, 1% (v/v) L-glutamine, 1% (v/v) non-essential amino acids, 1% (v/v) sodium pyruvate and 1% (v/v) penicillin–streptomycin (Life Technologies). Fibroblasts, HEK293T and NSC-34 motor neuron-like cells were grown in Dulbecco's modified Eagle medium (DMEM) supplemented with 10% (v/v) foetal bovine serum, 1% (v/v) L-glutamine and 1% (v/v) penicillin–streptomycin (Life Technologies). COS1 cells were grown in DMEM:HAM-F12 medium supplemented with 10% (v/v) foetal bovine serum, 1% (v/v) L-glutamine and 1% (v/v) penicillin–streptomycin (Life Technologies). C2C12 cells were grown in DMEM medium supplemented with 10% (v/v) foetal bovine serum and 1% (v/v) penicillin–streptomycin (Life Technologies). HEK293 Flp-In Tafazzin knockout cells were grown in DMEM medium supplemented with 10% (v/v) foetal bovine serum, 1% (v/v) L-glutamine and 1% (v/v) penicillin–streptomycin (Life Technologies) and 100 μg ml$^{-1}$ zeocin (Invitrogen). Lymphoblasts were grown in RPMI medium supplemented with 15% (v/v) foetal bovine serum, 1% (v/v) L-glutamine and 1% (v/v) penicillin–streptomycin (Life Technologies). Cells were cultured at 37 °C and 5% $CO_2$, except for lymphoblasts, which were cultured at 37 °C and 6% $CO_2$. For heat shock experiments, cells were incubated at 42 °C for 1 h. Note that some of the heat shock experiments were performed in a different incubator and that required the cells to be heated to 45 °C to see mitochondrial recruitment.

### Generation of CRISPR/Cas9 knockout cells

Knockout cells were generated using the CRISPR/Cas9 technology. Our protocol is based on that described by Ran et al.[89]. In brief, we designed sgRNAs with crispr.mit.edu or the sgRNA Designer from Broad Institute (https://portals.broadinstitute.org/gpp/public/analysis-tools/sgrna-design) and we selected at least two different sgRNAs per gene of interest, sgRNAs that cut only after 30 amino acids (to avoid producing micro open reading frames) and those that had the lowest amount of predicted off-target sites in coding regions. We ordered the sgRNAs as phosphorylated primers from Integrated DNA Technologies. Both top and bottom primers contained BbsI overhang sites and were annealed at room temperature for 1 h after they were denatured at 95 °C for 5 min. The annealed oligonucleotides were diluted 1:200 and then cloned into the pSpCas9(BB)-2A-PURO (PX459) V2.0 (plasmid #62988, Addgene) following a single-step digestion-and-ligation reaction, for which we mixed BbsI restriction enzymes (FD1014, Thermo Fisher Scientific) with T7 DNA ligase (EZ L6020L, Enzymatics) along with ATP (10 mM), 1,4-dithiothreitol (DTT, 10 mM) and Tango buffer. The reaction was incubated at 37 °C for 5 min and then at 21 °C for 5 min,

which was repeated six times (total ligation reaction of 1 h). Residual linearized DNA was digested by PlasmidSafe exonuclease (E3101K, Epicentre) for 30 min at 37 °C and inactivated for 30 min at 70 °C. Constructs were transformed into DG-1 chemo-competent bacteria (GE-DG1C-96, Eurogentec), plated on agar plates and sequence verified with Sanger sequencing. Constructs co-expressing the sgRNA of interest and Cas9 were transiently transfected in the mammalian destination cells with Lipofectamine LTX with Plus Reagent (15338500, Thermo Fisher Scientific). Twenty-four hours after the transfection, cells were refreshed with medium containing puromycin (1 μg ml$^{-1}$). Cells were subjected to this puromycin-containing medium for 72 h in total, after which we verified that the non-transfected well had completely died. Surviving cells were allowed to recover for 2 days and serially diluted into a 96-well plate with an estimated 0.5 cells per well. The wells were visually inspected the following days, and wells that contained more than one clone were discarded. We verified the gene inactivation in multiple clones using DNA sequencing and western blot (using a polyclonal antibody against the full-length protein). For DNA sequencing, we first PCR amplified the region of interest and cloned these amplicons in pUC19 vectors. Constructs were transformed into DG-1 chemo-competent bacteria (GE-DG1C-96, Eurogentec), plated on agar plates and individual colonies were picked to sequence the insert. This strategy allowed us to sequence verify the different alleles as some in vitro cell lines contained multiple copies of our genes of interest. To ensure we would cover all alleles, we sequenced twice as many clones as the predicted copy number of our genes of interest. Clones that contained sequence-verified premature stop codons, confirmed absence of the protein product, and a normal morphology were selected for experiments.

### Generation of stable cells through lentiviral transduction

Stable cell lines were generated by lentiviral transduction. Lentiviral vectors that contained the open reading frames of HSPB1 (NM_001540), HSPB5/CRYAB (NM_001289807), HSPB8 (NM_014365), BAG3 (NM_004281) or shRNAs against the indicated targets were used to perform the lentiviral transduction. To start, HEK293T cells were transiently transfected with packaging (pCMV dR8.91), envelope (pMD2-VSV) and pLenti6/V5 (Life Technologies) plasmids using linear polyethylenimime 25K (PEI) (23966-1, PolySciences Europe) or PEI MAX 40K (24765-1, PolySciences Europe). After 48 h, the virus containing supernatant was collected from the HEK293T cells, filtered through a 0.45 μm filter (SLHV033RB, Millipore) and used to infect the destination cells. Virus medium was removed 24 h after transduction, and we selected for infected cells by adding blasticidin S (5 μg ml$^{-1}$; ant-bl-1, InvivoGen Europe) or puromycin (1 μg ml$^{-1}$; ant-pr-1, InvivoGen Europe). Cells were cultured at 37 °C and 5% CO$_2$.

### Mouse breeding and tissue collection

Mice were maintained at the animal house facility of the University of Antwerp. Animals were inspected daily, and those that showed health concerns were culled. All experimental mice were free of pathogens and were on a 12 h light/12 h dark cycle, with unlimited access to food and water. The temperature and humidity at the facility were maintained (respectively 20–21 °C and 40–60%). Samples for mitochondrial isolation from the heart were obtained from 6-month-old female C57BL/6J mice.

### Cellular fractionation and mitochondrial isolation

Cells were collected by trypsin or with cell scrapers and washed in PBS. A fraction of the PBS-washed cell pellet was transferred to a new tube and was used to obtain the cell lysate (WCL) fraction. To this end, the cell pellet was resuspended in RIPA lysis buffer (137 mM NaCl, 2.7 mM KCl, 10 mM sodium fluoride, 1 mM sodium pyrophosphate and 0.5% Nonidet-P-40, supplemented by complete protease inhibitor and Phospho-STOP inhibitor mix (4906837001, Roche Applied Science)) and incubated for 20 min on ice, cleared by centrifugation for 10 min at

20,000$g$ and supernatant was collected as NP-40 soluble fraction and used as the WCL fraction for loading on western blot. The remaining PBS-washed cells were processed further for mitochondrial isolation. In this case, the cell pellet was resuspended in mitochondrial isolation buffer (250 mM mannitol, 0.5 mM EGTA and 5 mM HEPES–KOH pH 7.4). The remaining of the sample was processed further using a 26.5 G needle (303800, Becton Dickinson), cells were lysed with ten strokes after which the homogenate was subjected to differential centrifugation. The homogenate was centrifuged twice at 600$g$ for 10 min at 4 °C to remove cell debris and intact nuclei. The resulting supernatant was centrifuged twice at 7,000$g$ for 10 min at 4 °C. The mitochondrial pellet was resuspended in mitochondrial isolation buffer, while the supernatant was kept as cytosolic lysate. The resuspended pellets were centrifuged twice more at 10,000$g$ for 10 min at 4 °C. Finally, the purified mitochondrial pellet was lysed in RIPA buffer. All steps were performed at 4 °C.

Mitochondrial isolation from mouse hearts was achieved by isolating the heart from a C57/BL6J mouse. The tissue was incubated with trypsin (15090-046, Life Technologies) for 20 min before bovine serum albumin (BSA, A7906, Sigma-Aldrich) was added to inactivate trypsin. Digested tissue was then homogenized with a douncer (ten strokes) in mitochondrial isolation buffer (220 mM mannitol, 70 mM sucrose, 2 mM EGTA, 0.1% (w/v) BSA and 20 mM HEPES–KOH pH 7.4) before the homogenate was subjected to differential centrifugation. The differential centrifugation was performed as described above for cells and contained centrifugation steps of 600$g$, 7,000$g$ and 10,000$g$. All steps were performed at 4 °C.

### Proteinase K protection assay

Mitochondria were isolated as described above and subjected to a proteinase K protection assay (as explained in Extended Data Fig. 2). Isolated mitochondria were divided over four different tubes. Two tubes contained regular mitochondrial isolation buffer (250 mM mannitol, 0.5 mM EGTA and 5 mM HEPES–KOH pH 7.4) while two other tubes contained osmotic swelling buffer (0.5 mM EGTA and 5 mM HEPES–KOH pH 7.4) due to the absence of mannitol. We added 10 μg ml$^{-1}$ proteinase K (EO0492, Life Technologies) to one tube of each condition (that is, one tube of regular isolation buffer and one tube of osmotic swelling buffer). Mitochondria were subjected to proteinase K for 20 min at 4 °C on a head-over-head rotor, and where indicated, the proteinase K treatment was supplemented with 0.5% (v/v) Triton X-100. Proteinase K digestion was blocked with 1 mM phenylmethylsulfonyl fluoride (PMSF, sc-482875, Santa Cruz Biotechnology) on ice for 10 min. Mitochondria were isolated by centrifugation at 10,000$g$ for 10 min at 4 °C. Mitochondrial pellets were resuspended in NuPAGE LDS sample buffer (NP0007, Life Technologies) and analysed by SDS–PAGE and immunoblotting.

### Sodium carbonate assay

For sodium carbonate extraction, isolated mitochondria were pelleted at 10,000$g$ and resuspended in freshly prepared 100 mM Na$_2$CO$_3$ (pH 10.5, 11.0, 11.5, 12.0 and 12.5). Samples were rigorously pipetted up and down and incubated on ice for 30 min. Homogenates were centrifuged at 20,000$g$ for 30 min at 4 °C to separate the soluble from pellet fraction. The supernatant and pellet fractions were mixed with or resuspended in NuPAGE LDS sample buffer (NP0007, Life Technologies) and analysed by SDS–PAGE and immunoblotting.

### Sodium chloride assay

For sodium chloride extraction, isolated mitochondria were pelleted at 10,000$g$ and resuspended in 1 M NaCl. Samples were rigorously pipetted up and down and incubated on ice for 30 min. Homogenates were centrifuged at 20,000$g$ for 30 min at 4 °C to separate the soluble from pellet fraction. The supernatant and pellet fractions were mixed or resuspended, respectively, in NuPAGE LDS sample buffer (NP0007, Life Technologies) and analysed by SDS–PAGE and immunoblotting.

## Mitochondrial import assay

For in vitro import assays in isolated mitochondria, pSP64-Poly(A) vectors (P1241, Promega) encoding HSPB1 or LIAS were used for in vitro coupled transcription/translation using the TNT SP6 Quick Coupled Transcription/Translation kit (L2080, Promega). Precursor proteins were synthesized for 60 min at 30 °C using 1 μg of cDNA in rabbit reticulocyte lysate supplemented with 20 μM $^{35}$S-methionine shaking slowly at 600 rpm the day before the experiment and stored at −20 °C. Mitochondria were isolated in mitochondrial isolation buffer (220 mM mannitol, 70 mM sucrose, 5 mM HEPES–KOH pH 7.4 and 1 mM EGTA–KOH without protease inhibitor). In vitro synthesized precursor proteins were mixed with 50 μg of isolated mitochondria in mitochondrial isolation buffer for different timepoints (0 min, 30 min and 60 min) at 37 °C, shaking 600 rpm. To dissipate the mitochondrial membrane potential, mitochondria were supplemented with 20 μM carbonyl cyanide-3-chlorophenylhydrazone (CCCP; C2759, Sigma-Aldrich) 5 min before incubation with translation products. After import, samples were returned to ice. After centrifugation at 10,000$g$ at 4 °C to collect mitochondria, mitochondrial pellets were resuspended in 400 μl of 20 μg ml$^{-1}$ proteinase K (Carl Roth, 7528) isolation buffer and left on ice for 20 min. The reactions were stopped by adding 2 μl of 200 mM PMSF (10837091001, Roche) and spun at 10,000$g$ at 4 °C. The pellet was resuspended in 25 μl 2× sample buffer (NuPage, NP007, Invitrogen) containing fresh β-Mercapto (M6250, Sigma) and boiled at 95 °C for 5 min and analysed by SDS–PAGE.

## Western blot

Cells were collected by centrifugation and lysed in RIPA lysis buffer (137 mM NaCl, 2.7 mM KCl, 10 mM sodium fluoride, 1 mM sodium pyrophosphate and 0.5% Nonidet-P-40) supplemented by complete protease inhibitor and Phospho-STOP inhibitor mix (4906837001, Roche Applied Science) for 20 min on ice and cleared by centrifugation for 10 min at 20,000$g$. The protein supernatant was collected as soluble fraction; the pellet contained the non-soluble fraction. To determine the protein concentration, we used a Pierce BCA assay (23225, Thermo Fisher Scientific) or Bradford assay (B6916, Sigma-Aldrich) in case protein lysis was achieved with a 26.5 G needle (303800, Becton Dickinson). Equal amounts of protein lysate were mixed with reducing NuPAGE LDS Sample buffer (NP0007, Life Technologies) supplemented with 100 mM DTT (20290, Thermo Fisher Scientific). Samples were boiled at 95 °C for 5 min before loaded on 4–12% or 12% SDS–PAGE gels (NP0322BOX or NP0342BOX, Life Technologies) with SeeBlue Plus2 Pre-stained Protein Standard (LC5925, Life Technologies). After separation, proteins were transferred to nitrocellulose membranes (RPN132D, GE Healthcare Life Sciences) for 1 h at 4 °C. Membranes were blocked with 5% milk powder diluted in PBS (14190250, Thermo Fisher Scientific) with 0.1% Tween 20 (28352, Thermo Fisher Scientific). Afterward, membranes were incubated with primary and secondary antibodies for 1 h at room temperature or overnight at 4 °C. Primary antibodies were obtained for: HSPB1 (1:1,000, SPA-800, Enzo Life Sciences), HSPB2 (1:1,000, SAB4501458, Sigma-Aldrich), HSPB3 (1:500, ab213591, Abcam), HSPB4 (1:1,000, PA5-72139, Thermo Fisher Scientific), HSPB5 (1:500, MAB4849, R&D Systems), HSPB6 (1:2,500, MAB4200, R&D Systems), HSPB7 (1:200, sc-393739, Santa Cruz Biotechnology), HSPB8 (1:1,000, #3059, Cell Signaling Technology), HSPB9 (1:1,000, PA5-49139, Thermo Fisher Scientific), Hsp90 (1:200, sc-13119, Santa Cruz Biotechnology), VDAC1 (1:1,000, ab14734, Abcam), Tubulin (1:10,000, ab7291, Abcam, Cambridge, UK), TOM20 (1:1,000, ab56783, Abcam), AIFM1 (1:1,000, GTX102399, Genetex) or (1:1,000, #4642, Cell Signaling Technology), HCCS (1:2,000, 15118-1-AP, Proteintech Group), SDHA (1:3,000, GTX632636, Genetex), mHSP60 (1:200, sc-13115, Santa Cruz Biotechnology), CytC (1:1,000, #4272, Cell Signaling Technology), V5 (1:1,000, R96025, Invitrogen), GFP (1:1,000, ab290, Abcam) or (1:1,000, 1:5,000, GTX113617, Genetex), FLAG (1:1,000, F7424, Sigma-Aldrich), Mic19/CHCHD3 (1:1,000, GTX119821, Genetex), MFN2 (1:1,000, M6319, Sigma-Aldrich), BAG3 (1:1,000, A302-806A, Bethyl Laboratories),

TIM23 (1:200, sc-514463, Santa Cruz Biotechnology), Mic19/CHCHD3 (1:1,000, GTX119821, Genetex), MFN2 (1:1,000, M6319, Sigma-Aldrich), TIM23 (1:200, sc-514463, Santa Cruz Biotechnology), SLC25A11 (1:200, sc-515593, Santa Cruz Biotechnology), SLC25A12 (1:1,000, ab200201, Abcam), SLC25A13 (1:200, sc-393303, Santa Cruz Biotechnology), SLC25A22 (1:1,000, SAB2702048, Sigma-Aldrich), IMMT/Mitofillin (1:500, GTX115523, Genetex), DNAJC11 (1:1,000, 17331-1-AP, Proteintech), ATP5C1 (1:1,000, 60284-1-AP, Proteintech), UQCRC2 (1:200, sc-390378, Santa Cruz Biotechnology), MAIP1 (1:500, 24930-1-AP, Proteintech), ECHS1 (1:200, sc-515270, Santa Cruz Biotechnology), MTHFDL1L (1:200, sc-376722, Santa Cruz Biotechnology), SLP2 (1:1,000, 10348-1-AP, Proteintech) and CHCHD4 (1:1,000, 21090-1-AP, Proteintech), YMEL1 (1:1,000, 11510-1-AP, Proteintech), PARL (1:1,000, 26679-1-AP, Proteintech) and ClpB (1:1,000, A9130, ABclonal). Secondary antibodies were horseradish peroxidase conjugated (Jackson ImmunoResearch Europe). Blots were developed with either enhanced Chemiluminiscence ECL Plus (32132, Life Technologies) or SuperSignal West Femto Maximum Sensitivity Substrate (34096, Thermo Fisher Scientific) and imaged with an Amersham Imager 600 (GE Healthcare). The instrument was operated with ImageQuant software (GE Healthcare) and allowed us to use the semi-automatic exposure function. As stated in the operating instructions, the semi-automated exposure includes a short pre-exposure to determine the signal intensity. The system uses that information to calculate which exposure time will give the highest possible signal below saturation to enable accurate quantification of the bands, resulting in unsaturated signals within the linear range. The images were then saved in 16-bit image files. Band intensities were calculated by quantifying the mean pixel grey values using ImageJ software[90]. Mean pixel grey values were calculated for rectangular regions of interest.

## Monomer/dimer assay for HSPB1

To determine the monomer/dimer ratio of HSPB1, we first separated mitochondria from cytosol using the standard mitochondrial isolation protocol as described above. Samples were kept on ice all the time, and equal amounts of protein lysate were supplemented with NuPAGE LDS Sample buffer (NP0007, Life Technologies) either with or without 100 mM DTT (20290, Thermo Fisher Scientific). The non-reducing condition retained the disulphide bond between HSPB1 dimers and was visualized using the standard SDS–PAGE protocol using anti-HSPB1 (1:1,000, SPA-800, Enzo Life Sciences), Tubulin (1:10,000, ab7291, Abcam) and VDAC1 (1:1,000, ab14734, Abcam).

## Phosphorylation of HSPB1

To assess the phosphorylation of HSPB1, we extracted proteins from untreated cells, cells treated with 10 μM SB203580 (S8307, Sigma-Aldrich), or purified mitochondria with RIPA lysis buffer (137 mM NaCl, 2.7 mM KCl, 10 mM sodium fluoride, 1 mM sodium pyrophosphate and 0.5% Nonidet-P-40) supplemented by complete protease inhibitor and Phospho-STOP inhibitor mix (4906837001, Roche Applied Science) for 20 min on ice and cleared by centrifugation for 10 min at 20,000$g$. Samples were kept on ice all the time, and equal amounts of protein lysate were analysed by SDS–PAGE followed by immunoblotting as described above. Primary antibodies were obtained from: HSPB1 (1:1,000, SPA-800, Enzo Life Sciences), pHSPB1 phospho-ser15 (1:500, A00343, Genscript Corporation), pHSPB1 phospho-ser78 (1:500, A00528, Genscript Corporation), pHSPB1 phospho-ser82 (1:500, A00530, Genscript Corporation), pHSPB1 phospho-ser15 (1:1,000, ab5581, Abcam), pHSPB1 phospho-ser78 (1:1,000, ab32501, Abcam), pHSPB1 phospho-ser82 (1:1,000, ab155987, Abcam), Tubulin (1:10,000, ab7291, Abcam), VDAC1 (1:1,000, ab14734, Abcam) and FLAG (1:1,000, F7424, Sigma-Aldrich).

## Mitochondrial oxygen consumption rates

Mitochondrial respiration was measured with a Seahorse XFp instrument (Agilent Technologies) and the Cell Mito Stress Test assay

(103010-100, Agilent Technologies). HeLa cells were cultured in Seahorse XFp miniplates (103025-100, Agilent Technologies) for one night. Before the respiration measurements, cells were rinsed in PBS and changed to Seahorse XF DMEM medium pH 7.4 enriched with glucose, pyruvate and glutamine as described in the Mito Stress Test protocol. After the medium change, cells were cultured for 1 h at 37 °C in a $CO_2$-free incubator. The Seahorse injection ports on the sensor cartridge were filled with oligomycin (1 μM), FCCP (2 μM) and Antimycin A/Rotenone (0.5 μM). The sensor cartridges were placed on top of the Seahorse miniplates containing the cells and placed in the Seahorse XFp instrument for real-time analysis. Data were analysed with Wave software (Agilent Technologies).

## Immunofluorescence staining and confocal microscopy

For immunofluorescence microscopy, cells were seeded on glass coverslips (12 mm #1.5) and fixed in 4% paraformaldehyde (28906, Thermo Fisher Scientific) for 20 min at room temperature. After washing with PBS, cells were permeabilized with 0.1% Triton X-100 (9002-93-1, Sigma-Aldrich) in PBS for 2 min. Blocking was performed with 5% BSA (9048-46-8, Sigma-Aldrich) diluted in PBS for 1 h at room temperature. Primary and secondary antibodies were diluted in 5% BSA and incubated for 1 h at room temperature with three PBS washing steps in between. For visualizing the mitochondrial network, we used anti-TOM20 (1:100, ab186734, Abcam) followed by AlexaFluor secondary antibodies (Thermo Fisher Scientific). For visualizing the endoplasmic reticulum, we used anti-KDEL (1:100, ADI-SPA-827, Enzo Life Sciences). For visualizing the Golgi network, we used anti-Giantin (1:100, sc-46993, Santa Cruz Biotechnology). For visualizing FLAG-tagged MTS-HSPB1, we used anti-FLAG (1:100, F7424, Sigma-Aldrich). For detection of primary antibodies, we used AlexaFluor secondary antibodies (Thermo Fisher Scientific). Nuclei were stained at the end with Hoechst33342 (H3570, Life Technologies). Cells were mounted on microscopy slides in DAKO fluorescence mounting medium (S3023, Dako-Agilent) and stored at 4 °C until use. Confocal microscopy was performed with a Zeiss LSM700 laser scanning confocal microscopy with Plan-Apochromat 63×/1.40 objective.

## Expansion microscopy

For expansion microscopy[91], we followed the protocol by Chozinski et al.[92]. Cells were grown on standard cover glasses (12 mm #1.5), fixed for 20 min at room temperature in 3.2% paraformaldehyde (28906, Thermo Fisher Scientific) and 0.1% glutaraldehyde (111-30-8, Sigma-Aldrich). After reduction in sodium borohydride (16940-66-2, Sigma-Aldrich) for 5 min, we proceeded with a standard immunofluorescence staining protocol (as described above). We used anti-TOM20 (1:100, ab186734, Abcam), anti-HSPB1 (1:500, SPA-800, Enzo Life Sciences) and anti-HSP60 (1:50, MA3-012, Thermo Fisher Scientific) for the immunostaining. The standard immunofluorescence staining protocol was followed by crosslinking with 0.25% glutaraldehyde (111-30-8, Sigma-Aldrich) in PBS for 10 min. The samples were subjected to a gelation step using a mixture of: 2 M NaCl (7647-14-5, Sigma-Aldrich), 2.5% (w/w) acrylamide (79-06-1, Sigma-Aldrich), 0.15% (w/w) $N,N'$-methylenebisacrylamide (110-26-9, Sigma-Aldrich) and 8.625% (w/w) sodium acrylate (7446-81-3, Sigma-Aldrich) in PBS. Polymerization of this reaction was activated by addition of TEMED (110-18-9, Sigma-Aldrich) and APS (17874, Thermo Fisher Scientific) after which the solution was quickly added on top of the glass coverslip. After the gels had polymerized at room temperature, the gels were incubated for 30 min at 37 °C in a digestion buffer containing 8 U ml$^{-1}$ proteinase K (25530049, Thermo Fisher Scientific). Cover glasses were removed from underneath the digested gels. The gels were then incubated in high volumes (>30 ml) of distilled water, exchanged at least five times to obtain the maximal expansion of the gels. The expanded gels (4.3× expansion) were trimmed and positioned in a 50-mm-diameter glass-bottom dish (GWST-5040, WillCo Wells) and immobilized using 2% UltraPure LMP agarose (16520-050, Invitrogen). Images were acquired on a Zeiss LSM700 with Plan-Apochromat 63×/1.40 objective or, when indicated, on a Nikon Ti2 N-SIM S with a 100× CFI SR HP Plan Apochromat Lambda (1.35 numerical aperture) silicone immersion objective.

## Transmission electron microscopy

For transmission electron microscopy, we seeded HeLa cells on glass coverslips. Cells were immersed in fixative solution consisting of 4% paraformaldehyde (28906, Thermo Fisher Scientific) and 2.5% glutaraldehyde (111-30-8, Sigma-Aldrich) in 0.1 M sodium cacodylate buffer pH 7.2 for 4 h at room temperature followed by fixation overnight at 4 °C. After washing, cells were post-fixed in 1% osmium tetraoxide with potassium ferricyanide in 0.1 M sodium cacodylate buffer pH 7.2 at room temperature for 1 h. Afterwards, samples were washed with ddH$_2$O and subsequently dehydrated through a graded series of ethanol, including a bulk staining with 1% end concentration of uranyl acetate at the 50% ethanol step followed by embedding in Spurr's resin. Ultrathin sections of a gold interference colour were cut using an ultramicrotome (Leica EM UC6) followed by a post-staining in a Leica EM AC20 for 40 min in uranyl acetate and for 10 min in lead stain at 20 °C. Sections were collected on formvar-coated copper slot grids and images were acquired with a JEM 1400plus transmission electron microscope (JEOL) operating at 80 kV. Image analysis was performed with ImageJ[90].

## Immunogold electron microscopy

For immunogold labelling of cryosections according to the Tokuyasu method[93], we produced lentivirally transduced HeLa cells overexpressing wild-type HSPB1-eGFP and non-transduced cells were used as negative control. Cells were fixed in 4% paraformaldehyde (28906, Thermo Fisher Scientific) and 0.3% glutaraldehyde (111-30-8, Sigma-Aldrich) dissolved in 0.1 M phosphate buffer pH 7.4 for 1 h at room temperature. Cells were pelleted, embedded in 12% gelatin and cryoprotected in 2.3 M sucrose at 37 °C, mounted on aluminium cryopins and frozen in liquid nitrogen. Ultrathin frozen sections were cut in a Leica UC7/FC7 at −120 °C, collected with either 2% methylcellulose:2.3 M sucrose (1:1 ratio) or 2.3 M sucrose alone and mounted on formvar-carbon-coated nickel grids. Sections were incubated with anti-GFP (1:25, ab6556, Abcam, Cambridge, UK) for 60 min, followed by washing five times for 5 min (0.1% BSA in PBS). The grids were then incubated with PAG$_{10nm}$ (Cell Biology Utrecht University) and washed twice for 5 min, each time with 0.1% BSA in PBS, PBS and ddH$_2$O. The control consisted of treating sections with PAG$_{10nm}$ alone or, in addition, repeating the entire procedure of primary and secondary antibodies on non-transduced (GFP-negative) HeLa cells. Grids were viewed with a JEM 1400plus transmission electron microscope (JEOL) operating at 80 kV or with a Tecnai G2 Spirit Bio Twin Microscope (FEI) at 100 kV. To measure the density of HSPB1 in different subcellular compartments (IMS, matrix, cytoplasm and nucleus; Fig. 1d), images were acquired (43,000× magnification, 2,048 × 2,048 pixels, 1.13 nm per pixel), immunogold particles were counted and subcellular regions were manually segmented using ImageJ[90]. The nucleus, the cytoplasm, and the mitochondrial matrix were delineated with the ImageJ polygon selection tool, and the IMS was delineated with the segmented line tool using a line thickness of 20 pixels corresponding to a width of 22.5 nm. In total, over 2,133 immunogold particles were counted on 48 images originating from three sample grids. Image analysis was performed with ImageJ[90].

## Flow cytometric TMRE analysis

HeLa cells were loaded with 100 nM TMRE (ENZ-52309, Enzo Life Sciences) for 30 min at 37 °C. Cells were washed in PBS and trypsinized, and single cells were resuspended in PBS with 2% (v/v) foetal bovine serum before analysis. Flow cytometry was performed on a LSR Fortessa cytometer (BD Biosciences). Data analysis was performed with FlowJo10 software (Tree Star).

## LC–MS/MS analysis

For the affinity-enriched mass spectrometry, after HSPB1 affinity isolation, washed beads were resuspended in 150 µl trypsin digestion buffer (T6567, Sigma-Aldrich) and incubated with 1 µg trypsin (V511A, Promega) for 4 h at 37 °C. Beads were removed by centrifugation, and another 1 µg of trypsin was added to the supernatants to complete digestion overnight at 37 °C. Peptides were purified on Omix C18 tips (A57003100, Agilent), dried and redissolved in 20 µl loading buffer (0.1% trifluoroacetic acid in water:acetonitrile (98:2, v/v)) of which 2 µl was injected for liquid chromatography with tandem mass spectrometry (LC–MS/MS) analysis on an Ultimate 3000 RSLC nano LC (Thermo Fisher Scientific) in-line connected to a Q Exactive mass spectrometer (Thermo Fisher Scientific). The peptides were first loaded on a trapping column (in-house, 100 µm internal diameter (ID) × 20 mm, 5 µm beads C18 Reprosil-HD, Dr. Maisch). After flushing the trapping column, peptides were loaded in solvent A (0.1% formic acid in water) on a reverse-phase column (in-house, 75 µm ID × 250 mm, 3 µm beads C18 Reprosil-Pur, Dr. Maisch), packed in the needle and eluted by an increase in solvent B (0.1% formic acid in acetonitrile) in a linear gradient from 2% solvent B to 55% solvent B in 120 min, followed by a washing step with 99% solvent B, all at a constant flow rate of 300 nl min$^{-1}$. The mass spectrometer was operated in data-dependent, positive ionization mode, automatically switching between MS and MS/MS acquisition for the ten most abundant peaks in a given MS spectrum. The source voltage was set at 3.4 kV, and the capillary temperature at 275 °C. One MS1 scan ($m/z$ 400–2,000, AGC target 3 × 10$^6$ ions, maximum ion injection time 80 ms), acquired at a resolution of 70,000 (at 200 $m/z$), was followed by up to ten tandem MS scans (resolution 17,500 at 200 $m/z$) of the most intense ions fulfilling pre-defined selection criteria (AGC target 5 × 10$^4$ ions, maximum ion injection time 60 ms, isolation window 2 Da, fixed first mass 140 $m/z$, spectrum data type: centroid, underfill ratio 2%, intensity threshold 1.7 × 10$^4$, exclusion of unassigned, 1, 5–8 and >8 positively charged precursors, peptide match preferred, exclude isotopes on, and dynamic exclusion time 20 s). The higher-energy C-trap dissociation collision energy was set to 25% normalized collision energy, and the poly-dimethyl-cyclosiloxane background ion at 445.120025 Da was used for internal calibration (lock mass).

For the identification of the mitochondrial HSPB1 modification, after HSPB1 affinity isolation from purified mitochondria that were pre-treated with proteinase K for 20 min to remove all non-mitochondrial proteins. Samples were separated by SDS–PAGE and stained with Coomassie from which the HSPB1 bands were cut out. The Coomassie-stained gel bands were destained with a mixture of acetonitrile (Chromasolv, Sigma-Aldrich) and 50 mM ammonium bicarbonate (Sigma-Aldrich). The proteins were reduced using 10 mM DTT (Roche) and alkylated with 50 mM iodoacetamide. Chymotrypsin (Roche) digestion was carried out at 25 °C for 5 h in 50 mM ammonium bicarbonate. To stop the digestion, 10% formic acid was used, and peptides were extracted twice with 5% formic acid for 10 min in a cooled ultrasonic bath. Extracted peptides were pooled and desalted using C18 Stagetips[94]. Peptides were analysed on an UltiMate 3000 HPLC RSLC nanosystem (Thermo Fisher Scientific) coupled to an Exploris 480 orbitrap (Thermo Fisher Scientific), equipped with a nano-spray ion source using coated emitter tips (PepSep, MSWil). Samples were loaded on a trap column (Thermo Fisher Scientific, PepMap C18, 5 mm × 300 µm ID, 5 µm particles, 100 Å pore size) at a flow rate of 25 µl min$^{-1}$ using 0.1% Trifluoroacetic acid as mobile phase. After 10 min, the trap column was switched in line with the analytical C18 column (Thermo Fisher Scientific, PepMap C18, 500 mm × 75 µm ID, 2 µm, 100 Å) and peptides were eluted applying a segmented linear gradient from 2% to 80% solvent B (80% acetonitrile and 0.1% formic acid; solvent A 0.1% formic acid) at a flow rate of 230 nl min$^{-1}$ over 60 min. The mass spectrometer was operated in data-dependent mode, and survey scans were obtained in a mass range of 340–1,500 $m/z$ with lock mass activated, at a resolution of 60,000 at 200 $m/z$ and a normalized AGC target of 100%. In a 2 s cycle, the most intense ions were selected with an isolation width of 1.2 Thomson for a maximum of 100 ms, fragmented in the higher-energy C-trap dissociation cell at a normalized collision energy of 28%. The spectra were recorded at a normalized AGC target of 100% and a resolution of 15,000. Peptides with a charge of +1 or >+7 were excluded from fragmentation, the peptide match feature was set to preferred, the exclude isotope feature was enabled and selected precursors were dynamically excluded from repeated sampling for 20 s within a mass tolerance of 10 ppm.

## Protein identification and statistical analysis

Data analysis of the affinity-enriched immunoprecipitation samples was performed with MaxQuant (version 2.0.1.0) using the Andromeda search engine with default search settings including a false discovery rate (FDR) set at 1% at both the peptide and protein levels. Spectra were searched against the human proteins in the Uniprot/Swiss-Prot database (database release version of April 2015 containing 20,193 human protein sequences (www.uniprot.org) expanded with the eGFP sequence). Mass tolerances for precursor and fragment ions were set to 4.5 ppm and 20 ppm, respectively, during the main search. Enzyme specificity was set as C-terminal to arginine and lysine, also allowing cleavage at proline bonds with a maximum of two missed cleavages. Variable modifications were set to oxidation of methionine residues and acetylation of protein N-termini, phosphorylation of serine, threonine and tyrosine residues, and biotinylation of lysine residues. Proteins were quantified by the MaxLFQ algorithm integrated in the MaxQuant software. A minimum ratio count of two unique or razor peptides was required for quantification. Further data analysis was performed with the Perseus software (version 1.5.2.6) after loading the protein groups file from MaxQuant. Proteins only identified by site, contaminants and reverse database hits were removed, and replicate samples were grouped. Proteins with fewer than three valid values in at least one group were removed, and missing values were imputed from a normal distribution around the detection limit. For each quantified protein, a $t$-test was performed for pairwise comparison of both conditions. The results of this $t$-test are shown in the volcano plot in Fig. 5a. For each protein, the fold change value is indicated on the $X$ axis, while the statistical significance (−log $P$ value) is indicated on the $Y$ axis. Proteins outside the curved lines, set by an FDR value of 0.05 and an $S_0$ value of 1 in the Perseus software are considered to be significantly enriched. To identify mitochondrial proteins, we identified all significantly enriched proteins that are listed in MitoCarta3.0 database and compared the number of mitochondrial proteins to non-mitochondrial proteins.

Data analysis of the Coomassie excised bands to identify the modifications of mitochondrial HSPB1 was done using the MaxQuant software package[95] (version 2.1.0.0) to identify peptides and proteins from raw data. Spectra were searched against human reference proteome (Uniprot, version 2022.01) and a database containing common contaminants. The search was performed with chymotrypsin specificity and a maximum of four missed cleavages at a protein and peptide spectrum match FDR of 1%. Carbamidomethylation of cysteine residues was set as fixed, oxidation of methionine, and N-terminal acetylation as variable modifications. All other parameters were left at default. The annotated MS/MS spectrum displayed in Fig. 2b as generated using PeptideAnnotator (http://www.interactivepeptidespectralannotator. com/PeptideAnnotator.html).

## Statistics and reproducibility

Sample sizes were chosen according to common standards. Experiments were repeated at least three times unless otherwise stated in the figure legends. The $N$ number for all cell experiments represents independent experimental cultures. Statistical analyses have been performed using the statistical tests as stated in each figure legend. Data are presented as mean ± standard deviation (s.d.) if not indicated otherwise. The

statistical significance was assessed using Student's *t*-tests or one-way analysis of variance (ANOVA) with Dunnett's multiple comparisons tests when more than two groups were analysed. All *P* values below 0.05 were considered significant with *$P \leq 0.05$, **$P \leq 0.01$, ***$P \leq 0.001$ and ****$P \leq 0.0001$. Exact *P* values can be found in source numerical data. The statistical analysis was performed using PRISM software (GraphPad Software). We have not excluded any samples. No statistical methods were used to pre-determine the sample size. Images were processed with ImageJ National Institutes of Health. For the immunogold labelling, the staining appeared as very specific on the basis of the near-complete absence of immunogold labels on the negative-control samples, and multiple grids were viewed, ensuring consistency.

### Reporting summary
Further information on research design is available in the Nature Portfolio Reporting Summary linked to this article.

### Data availability
The Uniprot/Swiss-Prot database (database release version of April 2015) was accessed for proteomics data analysis. Proteomics from HSPB1-S135F affinity-enriched mass spectrometry has been deposited to the ProteomeXchange Consortium via the PRIDE partner repository with the dataset identifier PXD038275. Source data are provided with this paper. All other data supporting the findings of this study are available from the corresponding author on reasonable request. Further information on the research design is available in the Nature Research Reporting Summary linked to this article.

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

### Acknowledgements

We thank J. Herrmann (University of Kaiserslautern, Germany) and J. Buchner (Technical University Munich, Germany) for their critical input, M. Krols and T. Geuens for helpful discussions and assisting in establishing some of the assays, and the Peripheral Neuropathy Group for helpful discussions. We are grateful to M. de Bruyne and F. Baeke from the VIB BioImaging Core – Ugent Expertise Centre for their assistance with the Transmission Electron Microscopy, J. Van Duyse from the VIB Flow Core – Ugent for her assistance with the flow cytometer, M. Strazisar, L. Mateiu, T. De Pooter, S. Sluijs and G. Joris from the VIB-UAntwerp Neuromics Support Facility of the VIB Genomics core for their assistance with analysing genomic and transcriptomic samples, N. Corthout and A. Escamilla Ayala of the Leuven VIB BioImaging Core for assistance with structured illumination microscopy, and M. Hartl from the Max Perutz Labs Mass Spectrometry Facility for his help with identifying the HSPB1 modification. We thank S. M. Claypool (The Johns Hopkins University School of Medicine, USA) for sharing the Tafazzin-knockout cells. This work benefitted further from the help of MSc-thesis students K. Claes, R. Boons and T. Logghe as well as the help from short-rotation students in the lab. This research was supported in part by the Research Foundation - Flanders (FWO-Vlaanderen; grant number G041416N to V.T.), an FWO junior postdoc scholarship (grant number 1228021N to E.A.), an FWO-travel grant for participation in a sHSP conference abroad (grant number K1A3118N to E.A.), the Medical Foundation Queen Elisabeth (GSKE; to V.T.), the American Muscular Dystrophy Association (MDA; grant number 577497 to V.T.), the Association Belge contre les Maladies Neuromusculaires (ABMM; to E.A. and V.T.), the Rotary 'Hope in Head' programme (to E.A. and V.T.) and the H2020 Solve-RD programme 'Solving the unsolved rare diseases' under grant agreement 2017-779 257 (to V.T.). The Seahorse XF HS Mini Analyzer was supported by the University of Antwerp Basic Research Infrastructure grant number 41438. Tecnai G2 Spirit BioTwin Electron Microscope was supported by FWO-HERCULES large infrastructure grant Nr 25340. V.T. is a member of the μNeuro Center for Excellence at the University of Antwerp.

### Author contributions

E.A., B.A., P.R.-M., T.L. and V.T. conceived the project and designed the experiments. E.A., B.A., P.R.-M. and S.B. performed the experiments with assistance from the other co-authors. Microscopy was performed by E.A. and B.A, supervised by B.A. and analysed by B.A. Samples for expansion microscopy were prepared with the help from K.S. Samples for electron microscopy were prepared by R.D.R. and imaged by E.A., B.A. and R.D.R. Samples for affinity-enriched mass spectrometry were prepared by E.A. and analysed by F.I. Samples for mass spectrometry analysis of mitochondrial HSPB1 size shift were prepared by E.A. and analysed by the Max Perutz Labs Mass Spectrometry Facility. Samples for FACS were prepared by E.A. and analysed by G.V.I. Mitochondrial import assays were performed by P.R.-M. and T.L. Western blots and cell-based assays were performed by E.A. with assistance from S.B. Cell culture experiments, cloning and generation of stable cell lines was performed by E.A. with assistance from L.V. and V.D.W. The manuscript was written by E.A., B.A., P.R.-M., T.L. and V.T. and revised by all co-authors.

### Competing interests
The authors declare no competing interests.

### Additional information

**Extended data** is available for this paper at https://doi.org/10.1038/s41556-022-01074-9.

**Correspondence and requests for materials** should be addressed to Vincent Timmerman.

**a**

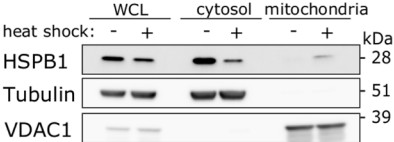

HEK293T

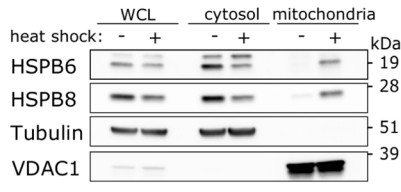

NSC34

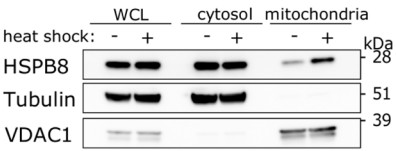

Neuro-2a

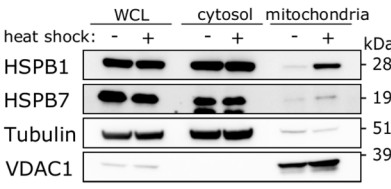

SH-SY5Y

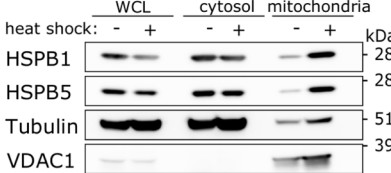

COS1

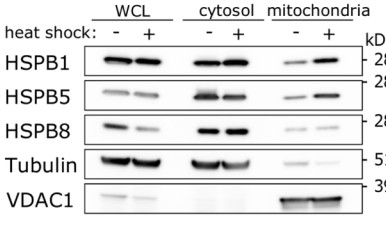

A498

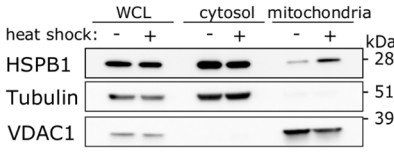

Lymphoblasts

**b**

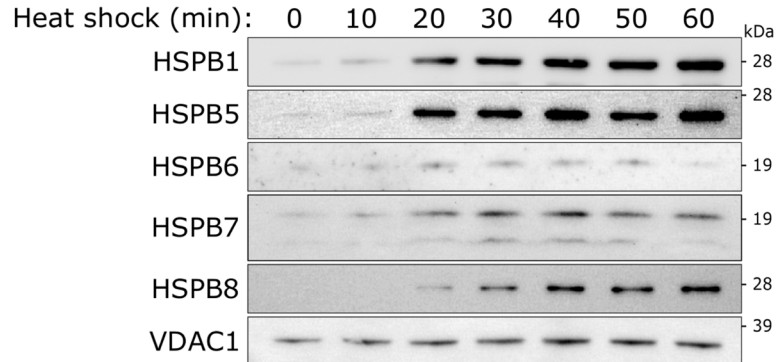

**Extended Data Fig. 1 | Small heat shock proteins enrich at mitochondria after heat shock in multiple human and mouse cell lines. a**, Mitochondria were isolated and compared to the cytosol and the NP-40-soluble whole cell lysate (WCL) from: HEK293T (human embryonic kidney cells), NSC-34 (motor neuron-like cells), Neuro-2A (neuronal crest-derived), SH-SY5Y (neuroblastoma cells), COS1 (African green monkey derived kidney cells), A498 (renal carcinoma cells), and healthy volunteer-derived human lymphoblasts. Cells were either left untreated or heat shocked (42 °C) for 1 h. Samples were analyzed by SDS-PAGE followed by immunoblotting using antibodies against sHSPs, Tubulin (cytosol), and VDAC1 (mitochondria). **b**, Mitochondria were isolated from C2C12 cells after heat shock (42 °C) for the indicated times. Samples were analyzed by SDS-PAGE followed by immunoblotting using antibodies against sHSPs or VDAC1 (loading control). Results are representative of two replicates (**a,b**). Unprocessed blots are available in source data.

**a**

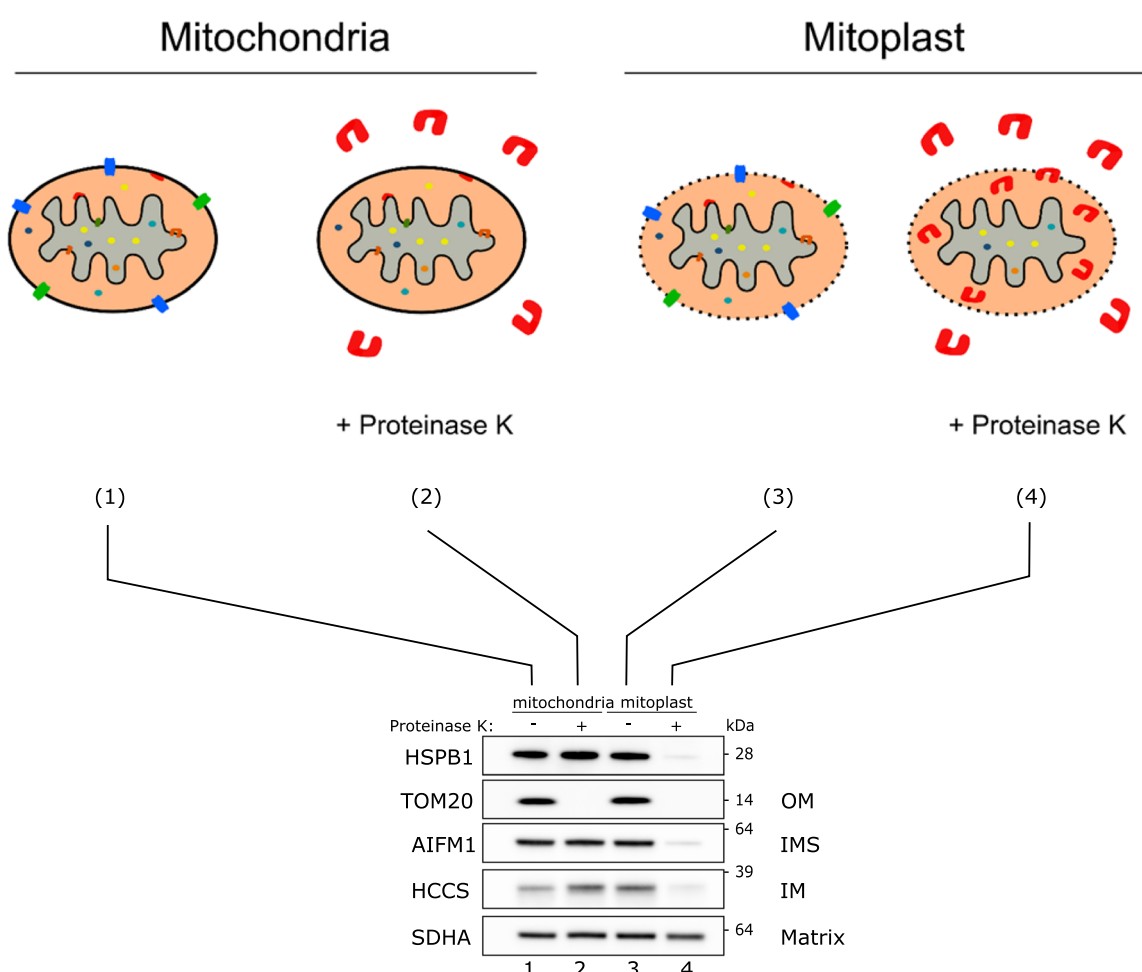

**b**

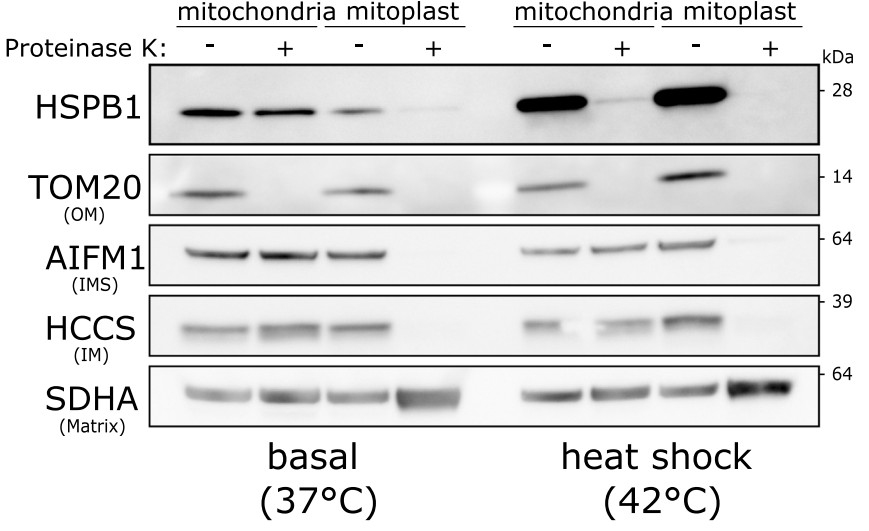

**Extended Data Fig. 2 | See next page for caption.**

**Extended Data Fig. 2 | Small heat shock proteins reside in the mitochondrial intermembrane space under basal conditions and accumulate at the outer membrane after heat shock. a**, Scheme of the proteinase K protection assay. Mitochondria were isolated from HEK293T cells and divided over 4 different tubes. The first two tubes contained regular mitochondrial isolation buffer (1 + 2) while the last two tubes (3 + 4) contained mitochondrial swelling buffer. Due to the absence of sucrose/mannitol in the swelling buffer, mitochondria were subjected to osmotic swelling which causes rupture of the OM (but not the IM) leading to mitoplast formation. The addition of proteinase K to one tube of each (2 and 4) resulted in the digestion of all accessible proteins. For (2) this leads to the digestion of all non-imported/OM proteins, for (4) this leads to the digestion of all OM, IMS, and IMS-facing IM proteins. Matrix proteins remain protected by the intact IM. The western blot displays the end result with markers for each of the different mitochondrial compartments. **b**, Proteinase K protection assay on mitochondria isolated from HeLa cells before and after heat shock. Isolated mitochondria were resuspended in a hypotonic (osmotic swelling) buffer to generate mitoplasts. Proteinase K (10 μg/ml) was used to digest accessible proteins. Samples were analyzed by SDS-PAGE followed by immunoblotting using anti-HSPB1, anti-TOM20 (OM), anti-AIFM1 (IMS), anti-HCCS (IM), and anti-SDHA (matrix) antibodies. Note that compared to Fig. 1 more mitochondrial protein was loaded, without WCL and cytosolic fractions on the same gel, resulting in higher detection sensitivity. Results in (**b**) are representative of five replicates. Unprocessed blots are available in source data.

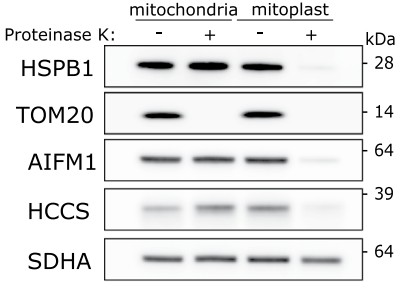

### HEK293T

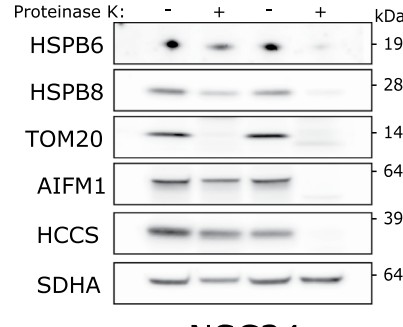

### NSC34

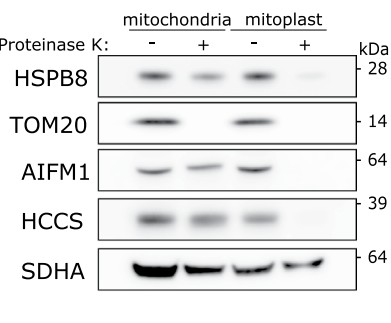

### Neuro-2a

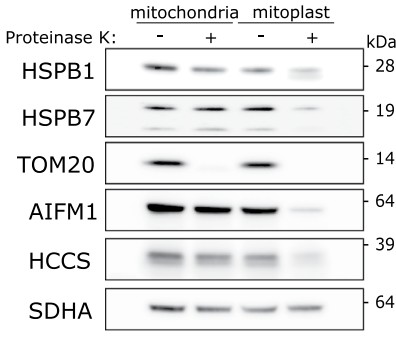

### SH-SY5Y

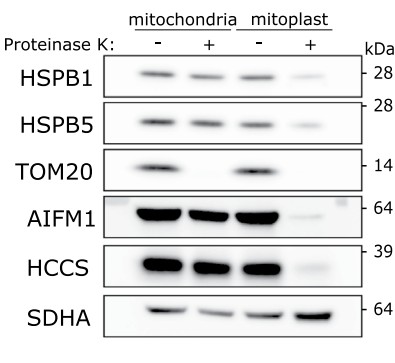

### COS1

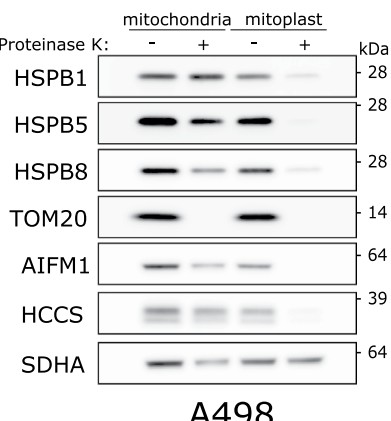

### A498

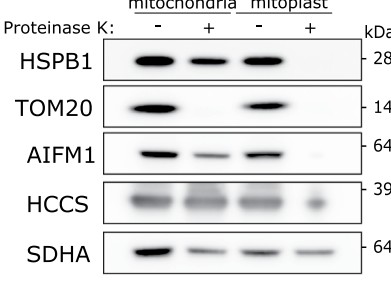

### Lymphoblasts

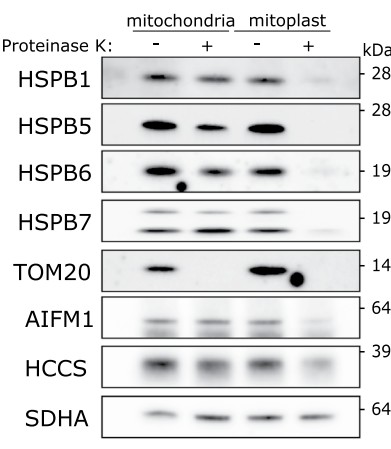

### Fibroblasts

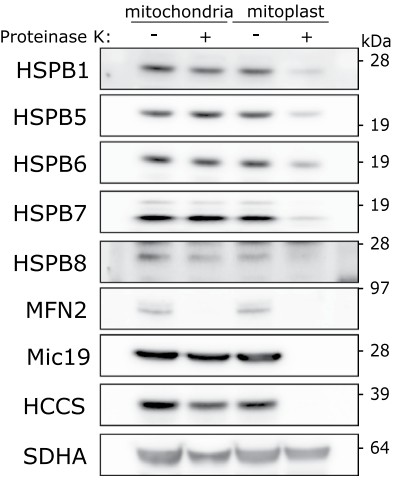

### Heart tissue

**Extended Data Fig. 3 | A fraction of most members of the small heat shock protein family resides in the mitochondrial intermembrane space.** Proteinase K protection assay on mitochondria isolated from non-heat shocked cells: HEK293T (human embryonic kidney cells), NSC-34 (motor neuron-like cells), Neuro-2A (neuronal crest-derived), SH-SY5Y (neuroblastoma cells), COS1 (African green monkey derived kidney cells), A498 (renal carcinoma cells), and healthy volunteer-derived human lymphoblasts and fibroblast, or isolated from the heart of a C57BL/6 mouse. Isolated mitochondria were resuspended in a hypotonic (osmotic swelling) buffer to generate mitoplasts. Proteinase K (10 μg/ml) was used to digest accessible proteins. Samples were analyzed by SDS-PAGE followed by immunoblotting using anti-HSPB antibodies and anti-TOM20 (OM), anti-AIFM1 (IMS), anti-HCCS (IM), and anti-SDHA (matrix) antibodies. Results are representative of two replicates. Unprocessed blots are available in source data.

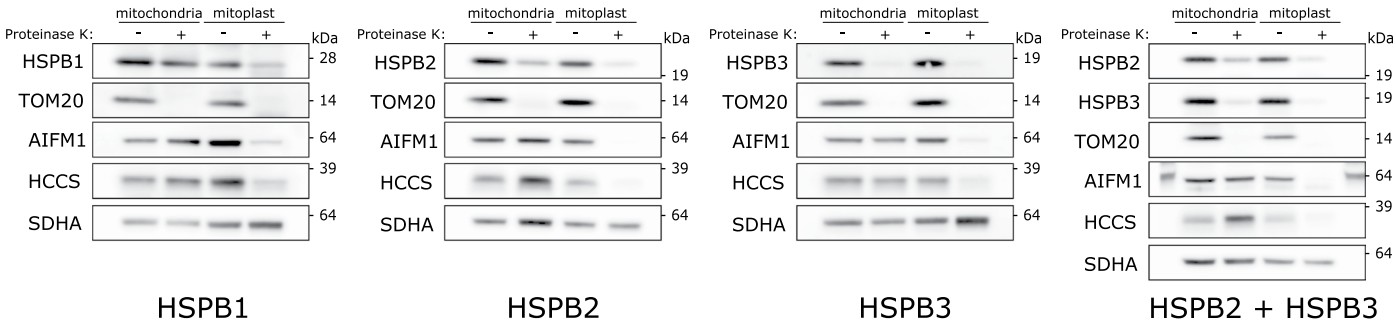

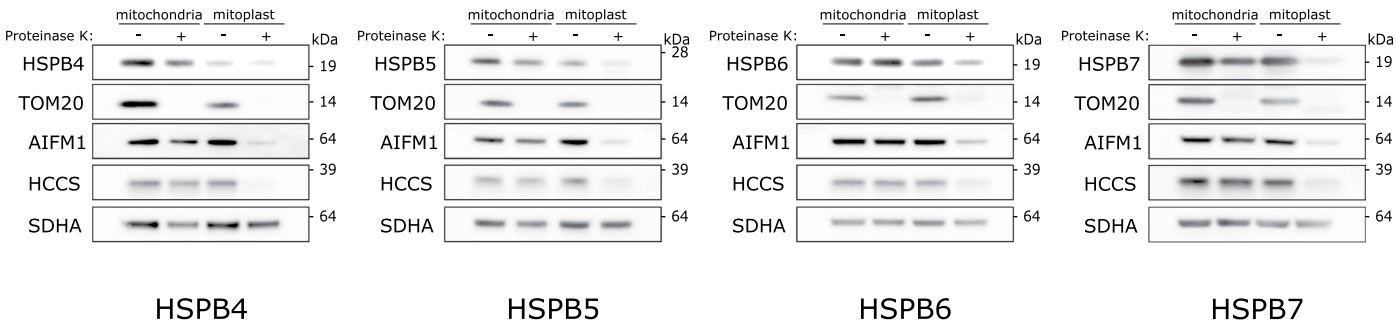

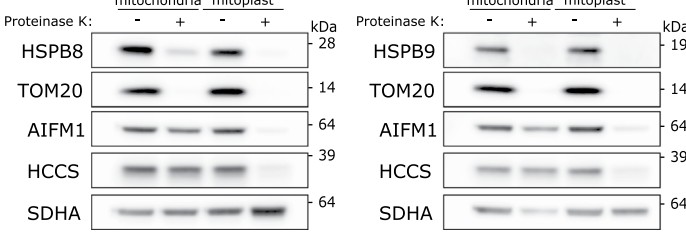

**Extended Data Fig. 4 | Overexpression of untagged small heat shock proteins demonstrates that a fraction of most family members reside in the mitochondrial intermembrane space.** Proteinase K protection assay on mitochondria isolated from non-heat shocked HeLa cells that were transiently transfected with untagged small heat shock proteins. This allowed us to assess the mitochondrial localization of tissue-specific sHSPs like HSPB4 (expressed in the eye lens) and HSPB9 (expressed in the testis). Isolated mitochondria were resuspended in a hypotonic (osmotic swelling) buffer to generate mitoplasts. Proteinase K (10 μg/ml) was added to digest accessible proteins. Samples were analyzed by SDS-PAGE followed by immunoblotting using anti-HSPB antibodies and anti-TOM20 (OM), anti-AIFM1 (IMS), anti-HCCS (IM), and anti-SDHA (matrix) antibodies. Results are representative of two replicates. Unprocessed blots are available in source data.

**a**

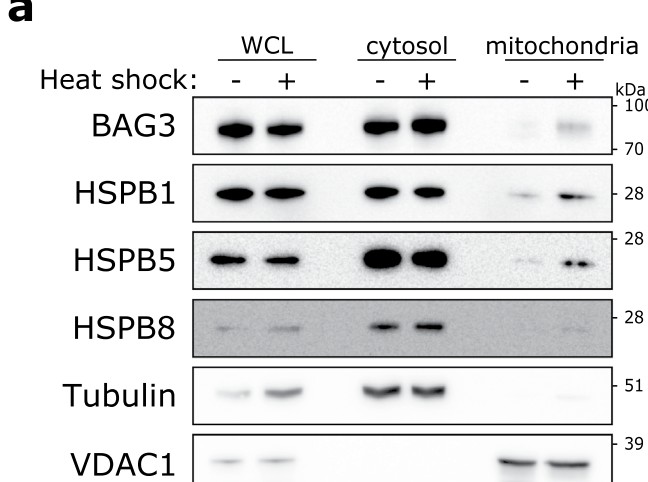

**b**

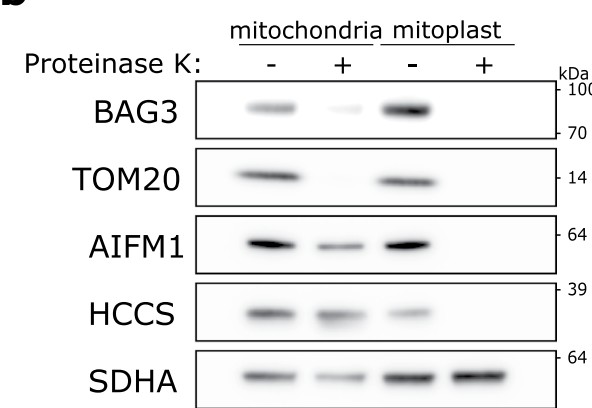

**c**

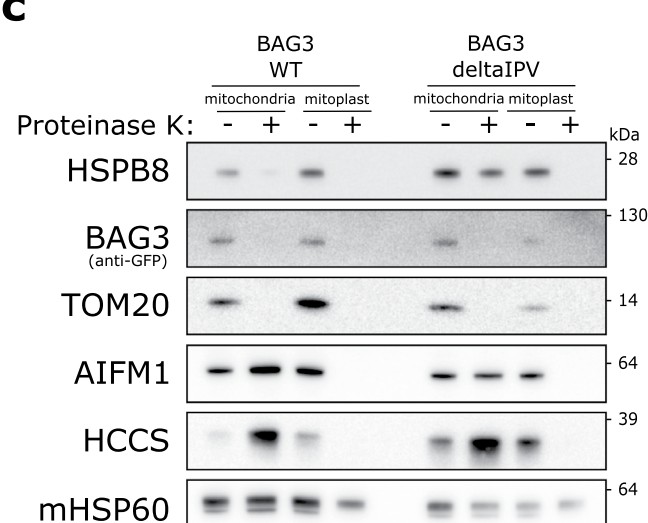

**Extended Data Fig. 5 | BAG3 is recruited to mitochondria upon heat shock but is not imported into mitochondria. a**, HeLa cells were subjected to heat shock (42 °C) for 1 h. Cells were collected afterward and the mitochondrial fraction was separated from the cytosolic fraction. Using immunoblotting we verified the abundance of BAG3 and sHSPs in each fraction. Tubulin was used as a marker for the cytosol and VDAC1 was used as a marker for mitochondria. The NP-40-soluble whole cell lysate (WCL) was used as a reference. **b**, Proteinase K protection assay on mitochondria isolated from HeLa cells. Isolated mitochondria were resuspended in a hypotonic (osmotic swelling) buffer to generate mitoplasts.

Proteinase K (10 µg/ml) was added to digest accessible proteins. Samples were analyzed by SDS-PAGE followed by immunoblotting using anti-BAG3, anti-TOM20 (OM), anti-AIFM1 (IMS), anti-HCCS (IM), and anti-SDHA (matrix) antibodies. **c**, BAG3 knockout HeLa cells were rescued with wild type or delta-IPV BAG3-GFP (deficient in binding to HSPB8) and subjected to proteinase K protection assay as in panel B. Samples were analyzed by SDS-PAGE followed by immunoblotting using anti-BAG3, anti-TOM20 (OM), anti-AIFM1 (IMS), anti-HCCS (IM), and anti-HSP60 (matrix) antibodies. Results are representative of two (**a,b**) or three (**c**) replicates. Unprocessed blots are available in source data.

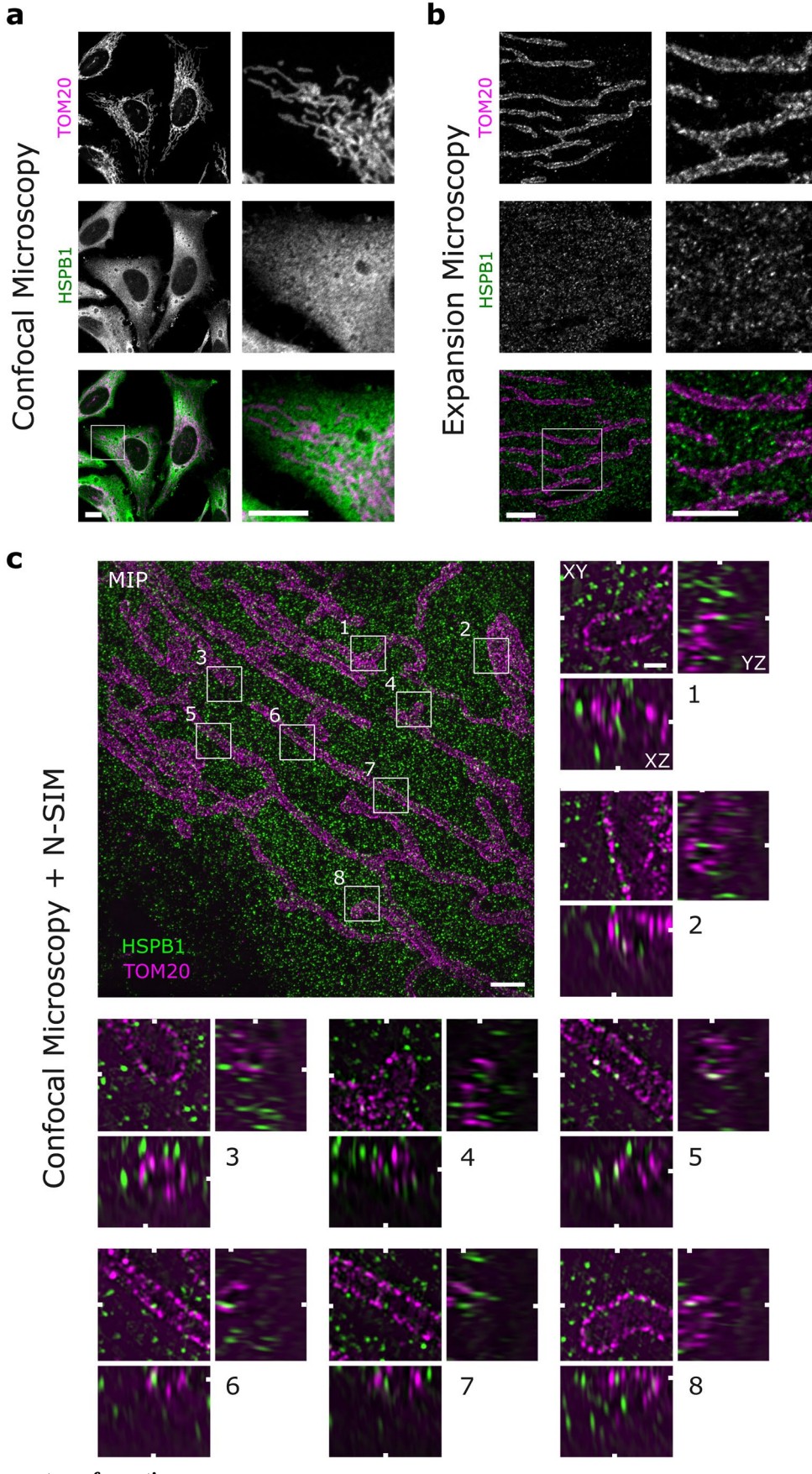

**Extended Data Fig. 6 | See next page for caption.**

**Extended Data Fig. 6 | Mitochondrial-localized HSPB1 imaged using confocal microscopy, expansion microscopy, or a combination of expansion microscopy and structured illumination microscopy.** Non-heat shocked HeLa cells were labeled by immunofluorescence staining of endogenous HSPB1 and TOM20 as an outer mitochondrial membrane marker. All samples were first fixed and stained according to our standard protocol for confocal microscopy. For expansion microscopy, immunolabeled samples were linked to a 4x expandable hydrogel. **a**, Non-expanded samples were imaged with confocal laser scanning microscopy. Scale bar is 10 µm. **b**, Expansion microscopy samples imaged with confocal microscopy. Expansion-corrected scale bar is 2.5 µm **c**, Samples prepared as in (**b**) were imaged with structured illumination microscopy (SIM). The overview image is a maximum intensity projection (MIP) of the imaged volume and the zoomed boxes (1 µm x 1 µm) show examples of HSPB1 present inside mitochondria. White markers indicate the position of the orthogonal planes in the volumes (XY, XZ, and YZ) and were sliced at HSPB1 spots fully contained by the mitochondrial TOM20 boundaries in all dimensions. Expansion-corrected scale bar is 1.0 µm and 0.20 µm (zooms). Images are representative of one (**c**) and more than three replicates (**a,b**).

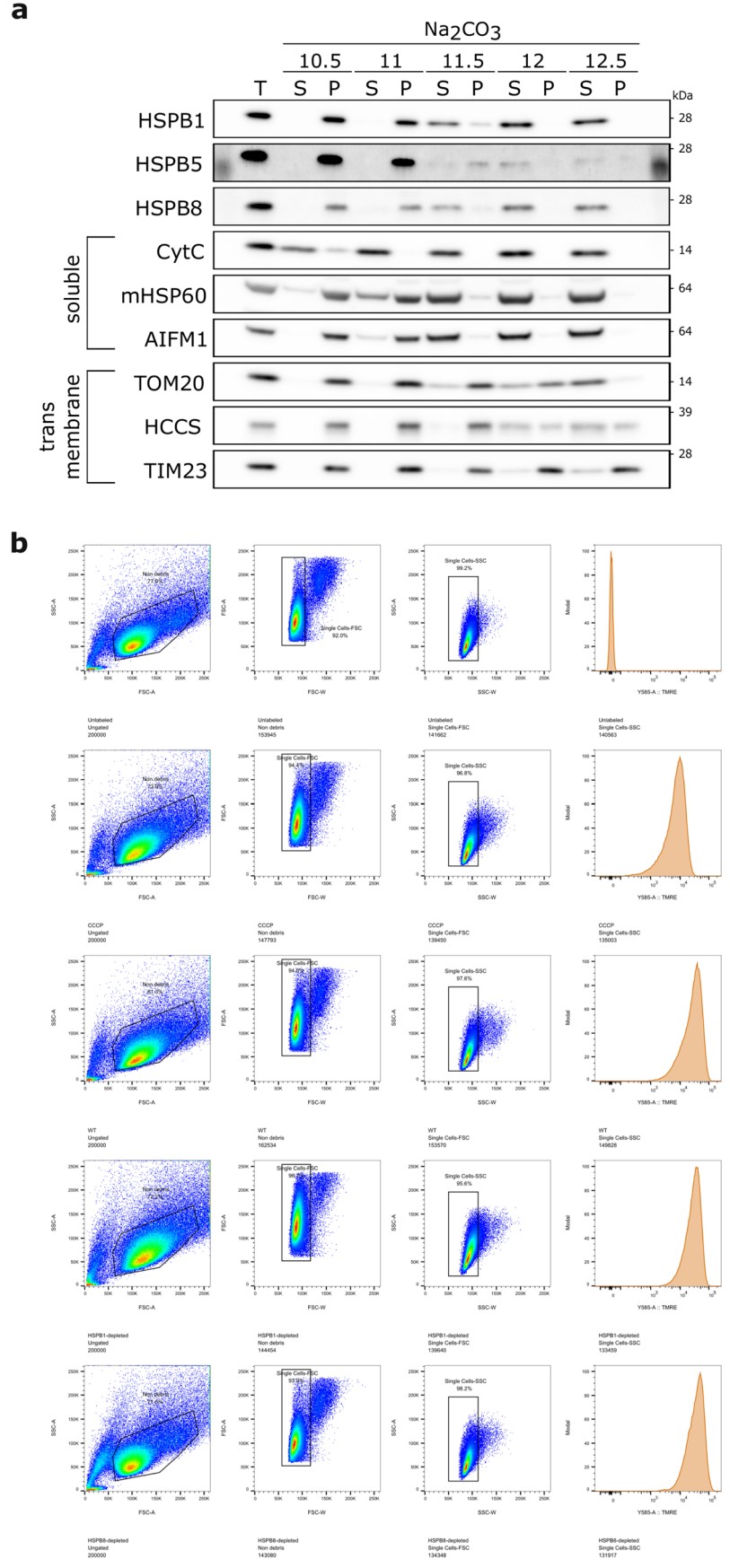

**Extended Data Fig. 7 | See next page for caption.**

**Extended Data Fig. 7 | Small heat shock proteins detach from mitochondrial membranes around pH 11.5 and have no effect on mitochondrial membrane potential. a**, Membrane association of small heat shock proteins was verified with a sodium carbonate assay. This assay allows the separation of integral membrane proteins (retained in the pellet) from peripheral membrane proteins and soluble proteins (extracted into the supernatant). Mitochondria isolated from HeLa cells were resuspended in $Na_2CO_3$ set to the indicated pH. Samples were analyzed by SDS-PAGE followed by immunoblotting using anti-HSPB1, anti-HSPB5, anti-HSPB8, anti-CytC (IMS), anti-mtHSP60 (matrix), anti-AIFM1 (IMS), anti-TOM20 (OM), anti-HCCS (IM), and anti-TIM23 (IM) antibodies. T = total; P = pellet; S = supernatant. Unprocessed blots are available in source data. **b**, Flow cytometry gating strategy and TMRE membrane potential in control or sHSP-depleted HeLa cells. As a negative control, the membrane potential was dissipated by CCCP treatment. Related to Fig. 3d: ungated, non-debris, singlets, TMRE-intensity. FSC: forward scatter, SSC: side scatter, A: Area, H: height, W: width. Results are representative of one (**b**), in which more than 130,000 cells were counted per condition, or three (**a**) replicates. Unprocessed blots are available in source data.

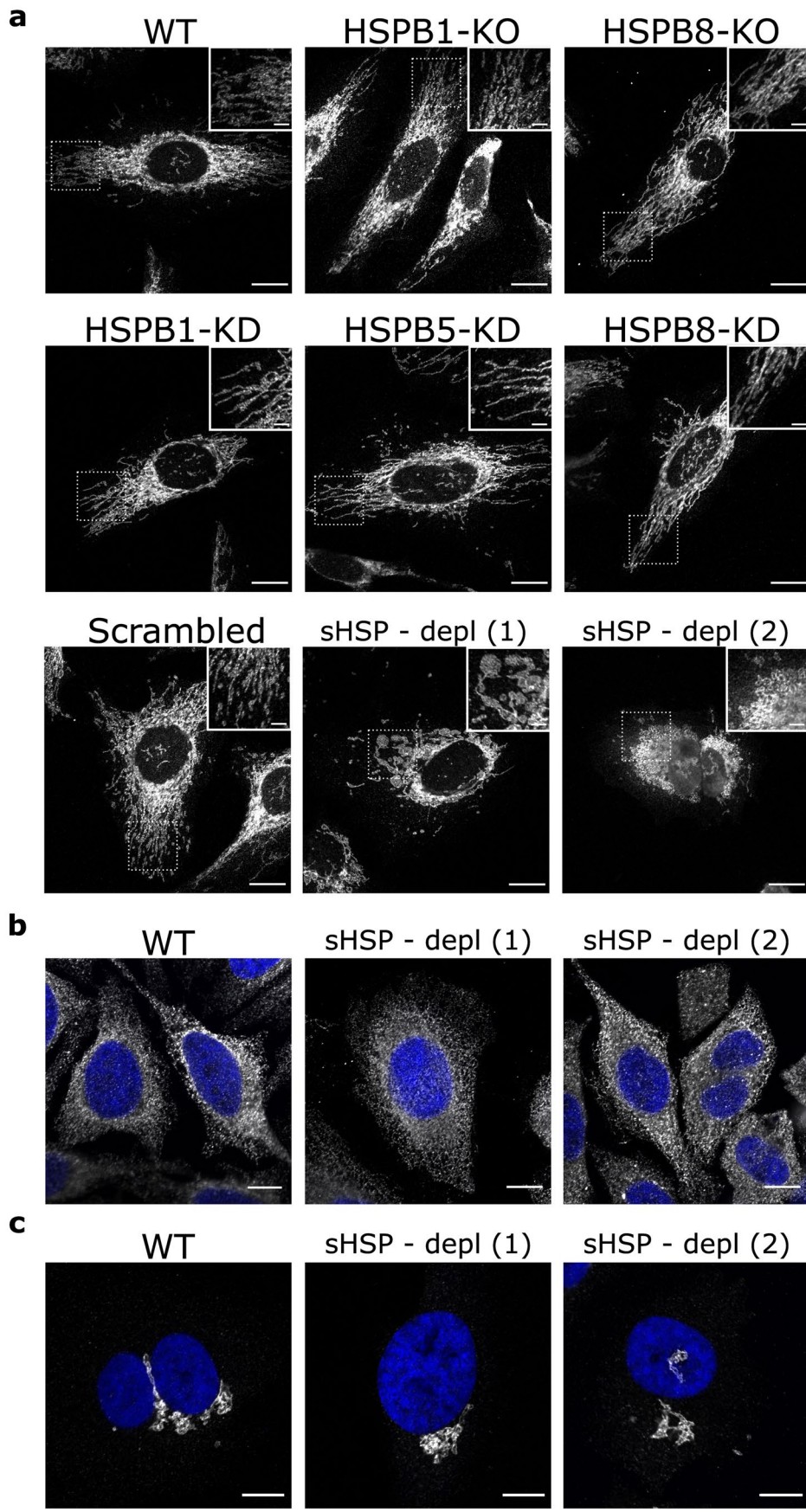

**Extended Data Fig. 8 | See next page for caption.**

**Extended Data Fig. 8 | Depletion of all sHSPs severely affects the morphology of mitochondria but not of other organelles. a**, Immunostaining (TOM20) of mitochondria in sHSP-depleted (depl), knockout (KO), or knockdown (KD) cell lines. The HeLa cell lines with reduced levels of all sHSPs and severe mitochondrial defects (used in Fig. 3) were generated by the combination of CRISPR/Cas9 knockouts and short hairpin RNAs (shRNA). Knockout or knockdown of individual components (HSPB1, HSPB5, HSPB8) did not present alterations in mitochondrial morphology compared to wild-type HeLa cells. Scale bars represent 10 μm and 2.5 μm (Zoom). **b,c**, Immunostaining of endoplasmic reticulum (KDEL) (**b**) and Golgi apparatus (Giantin) (**c**) in control (WT) and sHSP-depleted (depl) cell lines. Scale bars represent 10 μm. Images are representative of two (**b,c**) or three (**a**) replicates.

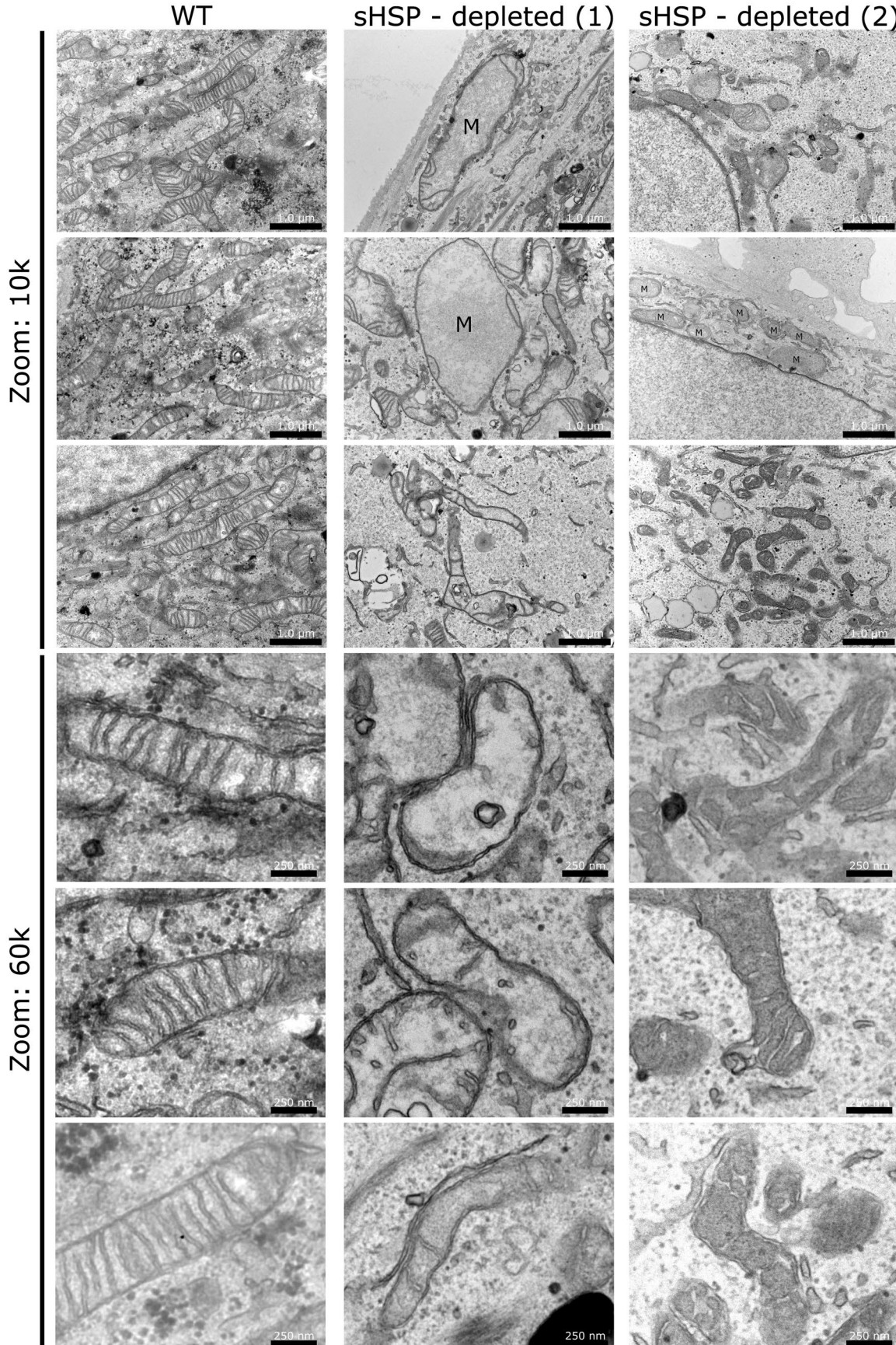

**Extended Data Fig. 9 | Transmission electron microscopy from control and sHSP-depleted cell lines.** Electron micrographs are shown from control (WT) and sHSP-depleted (depl) HeLa cell lines, as described in Fig. 3a. M = mitochondria. Notice the extremely large size of some swollen mitochondria in the depleted HeLa cells. Scale bars represent 1 μm (for 10k images) or 250 nm (for 60k images). Images are representative of two samples of one replicate EM experiment.

**a**

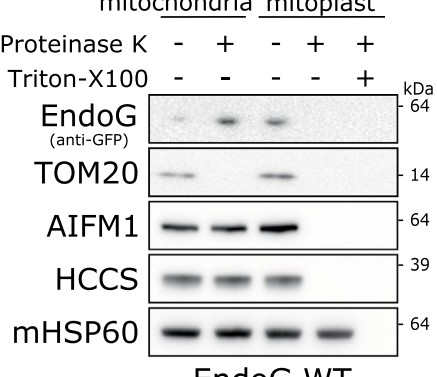
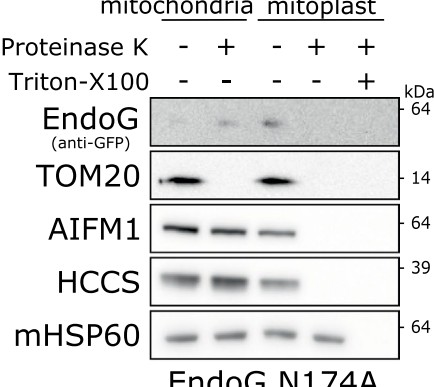

**b**

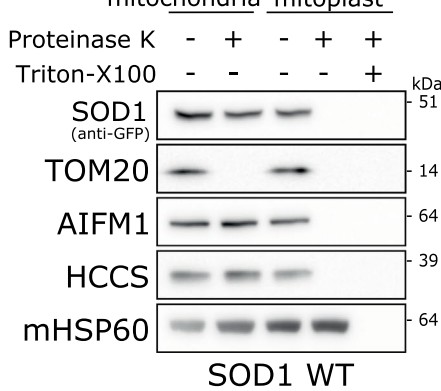
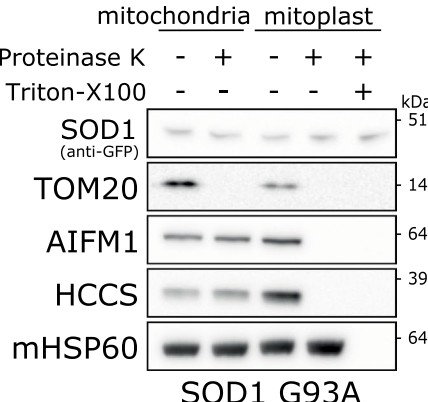

**Extended Data Fig. 10 | Proteinase K protection assay for wild-type and mutant EndoG and SOD1. a, b**, Wild type and mutant forms of EndoG (**a**) and SOD1 (**b**) were transfected in HeLa cells and analyzed for their mitochondrial import. Isolated mitochondria were resuspended in a hypotonic (osmotic swelling) buffer to generate mitoplasts. Proteinase K (10 µg/ml) was added to digest accessible proteins. Samples were analyzed by SDS-PAGE followed by immunoblotting using anti-GFP (EndoG or SOD1), anti-TOM20 (OM), anti-AIFM1 (IMS), anti-HCCS (IM), and anti-HSP60 (matrix) antibodies. Triton-X100 was added to solubilize matrix proteins and facilitate their proteinase K digestion, which revealed that mutant SOD1-G93A is partially protease-resistant as it is poorly digested even upon solubilization of all the different mitochondrial compartments. Results are representative of two replicates (**a,b**). Unprocessed blots are available in source data.

# Reporting Summary

## Statistics

For all statistical analyses, confirm that the following items are present in the figure legend, table legend, main text, or Methods section.

| n/a | Confirmed | |
|---|---|---|
| ☐ | ☒ | The exact sample size (*n*) for each experimental group/condition, given as a discrete number and unit of measurement |
| ☐ | ☒ | A statement on whether measurements were taken from distinct samples or whether the same sample was measured repeatedly |
| ☐ | ☒ | The statistical test(s) used AND whether they are one- or two-sided *Only common tests should be described solely by name; describe more complex techniques in the Methods section.* |
| ☒ | ☐ | A description of all covariates tested |
| ☐ | ☒ | A description of any assumptions or corrections, such as tests of normality and adjustment for multiple comparisons |
| ☐ | ☒ | A full description of the statistical parameters including central tendency (e.g. means) or other basic estimates (e.g. regression coefficient) AND variation (e.g. standard deviation) or associated estimates of uncertainty (e.g. confidence intervals) |
| ☐ | ☒ | For null hypothesis testing, the test statistic (e.g. *F*, *t*, *r*) with confidence intervals, effect sizes, degrees of freedom and *P* value noted *Give P values as exact values whenever suitable.* |
| ☒ | ☐ | For Bayesian analysis, information on the choice of priors and Markov chain Monte Carlo settings |
| ☒ | ☐ | For hierarchical and complex designs, identification of the appropriate level for tests and full reporting of outcomes |
| ☒ | ☐ | Estimates of effect sizes (e.g. Cohen's *d*, Pearson's *r*), indicating how they were calculated |

*Our web collection on statistics for biologists contains articles on many of the points above.*

## Software and code

Policy information about availability of computer code

| Data collection | Confocal microscopy images were collected using ZEN software version 2022 Black SP7 FP3 (Carl Zeiss Microscopy, GmbH, Germany) connected to a LSM700 with release version 14.0.26.201. |
|---|---|
| Data analysis | 1. FlowJo10 software (Tree Star Inc., Ashland, OR, USA) for FACS-data analysis.<br>2. MaxQuant (version 1.5.3.30 and version 2.1.0.0) for mass spectrometry data analysis.<br>3. PRISM 9 software (GraphPad Software, La Jolla, CA, USA) for statistical analysis and generate graphs.<br>4. ImageJ software (Schindelin et al. 2015) for immunofluorescence and electron microscopy image analysis.<br>5. Wave software version 2.6 (Agilent Technologies) for seahorse measurements. |

For manuscripts utilizing custom algorithms or software that are central to the research but not yet described in published literature, software must be made available to editors and reviewers. We strongly encourage code deposition in a community repository (e.g. GitHub). See the Nature Portfolio guidelines for submitting code & software for further information.

## Data

Policy information about availability of data

All manuscripts must include a data availability statement. This statement should provide the following information, where applicable:
- Accession codes, unique identifiers, or web links for publicly available datasets
- A description of any restrictions on data availability
- For clinical datasets or third party data, please ensure that the statement adheres to our policy

The Uniprot/Swiss-Prot database (database release version of April 2015) was accessed for proteomics data analysis. Proteomics from HSPB1-S135F affinity-enriched mass spectrometry has been deposited to the ProteomeXchange Consortium via the PRIDE partner repository with the dataset identifier PXD038275.

# Field-specific reporting

Please select the one below that is the best fit for your research. If you are not sure, read the appropriate sections before making your selection.

☒ Life sciences   ☐ Behavioural & social sciences   ☐ Ecological, evolutionary & environmental sciences

For a reference copy of the document with all sections, see nature.com/documents/nr-reporting-summary-flat.pdf

# Life sciences study design

All studies must disclose on these points even when the disclosure is negative.

| | |
|---|---|
| Sample size | No statistical methods were applied to pre-evaluate sample size. Experiments were performed at least as three replicates, according to current practices in the field. Statistical analysis was only performed on experiments for which the sample size included at least 3 biological replicates. Sample sizes were based on our previous experience (for microscopy see doi: 10.15252/embj.2019103811) and current standards in the field. |
| Data exclusions | No data were excluded from the analyses. |
| Replication | All experiments were replicated at least three times with similar findings. Sample sizes are provided in each figure legend. |
| Randomization | Samples were allocated into experimental groups by genotype and shRNA condition.Covariates were controlled for by maintaining all samples in the same growth and media conditions. |
| Blinding | The investigators were not blinded to treatment or genotype allocations in this study. For cell based experiments it was not possible to blind the experimenter. |

# Reporting for specific materials, systems and methods

We require information from authors about some types of materials, experimental systems and methods used in many studies. Here, indicate whether each material, system or method listed is relevant to your study. If you are not sure if a list item applies to your research, read the appropriate section before selecting a response.

## Materials & experimental systems

| n/a | Involved in the study |
|---|---|
| ☐ | ☒ Antibodies |
| ☐ | ☒ Eukaryotic cell lines |
| ☒ | ☐ Palaeontology and archaeology |
| ☐ | ☒ Animals and other organisms |
| ☐ | ☒ Human research participants |
| ☒ | ☐ Clinical data |
| ☒ | ☐ Dual use research of concern |

## Methods

| n/a | Involved in the study |
|---|---|
| ☒ | ☐ ChIP-seq |
| ☐ | ☒ Flow cytometry |
| ☒ | ☐ MRI-based neuroimaging |

# Antibodies

| | |
|---|---|
| Antibodies used | HSPB1 (1:1000, SPA-800, Enzo Life Sciences, Farmingdale, NY, USA)<br>HSPB2 (1:1000, SAB4501458, Sigma-Aldrich, Saint Louis, MI, USA)<br>HSPB3 (1:500, ab213591, Abcam, Cambridge, UK)<br>HSPB4 (1:1000, PA5-72139, Thermo Fisher Scientific, Waltham, MA, USA)<br>HSPB5/CRYAB (1:500, MAB4849, R&D Systems, Minneapolis, MN, USA)<br>HSPB6 (1:2500, MAB4200, R&D Systems, Minneapolis, MN, USA)<br>HSPB7 (1:200, sc-393739, Santa Cruz Biotechnology Inc., Santa Cruz, CA, USA)<br>HSPB8 (1:1000, #3059, Cell Signaling Technology, Danvers, MA, USA)<br>HSPB9 (1:1000, PA5-49139, Thermo Fisher Scientific, Waltham, MA, USA)<br>Hsp90 (1:200, sc-13119, Santa Cruz Biotechnology Inc., Santa Cruz, CA, USA)<br>BAG3 (1:1000, A302-806A, Bethyl Laboratories Inc., Montgomery, TX, USA)<br><br>VDAC1 (1:1000, ab14734, Abcam, Cambridge, UK)<br>Tubulin (1:10,000, ab7291, Abcam, Cambridge, UK)<br>TOM20 (1:1000, ab56783, Abcam, Cambridge, UK)<br>AIFM1 (1:1000, GTX102399, Genetex, Irvine, CA, USA)<br>AIFM1 (1:1000, #4642, Cell Signaling Technology, Danvers, MA, USA)<br>HCCS (1:2000, 15118-1-AP, Proteintech Group, Rosemont, IL, USA) |

SDHA (1:3000, GTX632636, Genetex, Irvine, CA, USA)
mHSP60 (1:200, sc-13115, Santa Cruz Biotechnology Inc., Santa Cruz, CA, USA)

CytC (1:1000, #4272, Cell Signaling Technology, Danvers, MA, USA)
V5 (1:1000, R96025, Invitrogen, Carlsbad, CA, USA)
GFP (1:1000, ab290, Abcam, Cambridge, UK)
GFP (1:25, ab6556, Abcam. Cambridge, UK)
GFP (1:1000, 1:5000, GTX113617, Genetex, Irvine, CA, USA)
FLAG (1:1000, F7424, Sigma-Aldrich, Saint Louis, MI, USA)

Mic19/CHCHD3 (1:1000, GTX119821, Genetex, Irvine, CA, USA)
MFN2 (1:1000, M6319, Sigma-Aldrich, Saint Louis, MI, USA)
TIM23 (1:200, sc-514463, Santa Cruz Biotechnology Inc., Santa Cruz, CA, USA)
SLC25A11 (1:200, sc-515593, Santa Cruz Biotechnology Inc., Santa Cruz, CA, USA)
SLC25A12 (1:1000, ab200201, Abcam, Cambridge, UK)
SLC25A13 (1:200, sc-393303, Santa Cruz Biotechnology Inc., Santa Cruz, CA, USA)
SLC25A22 (1:1000, SAB2702048, Sigma-Aldrich, Saint Louis, MI, USA)
IMMT/Mitofillin (1:500, GTX115523, Genetex, Irvine, CA, USA)
DNAJC11 (1:1000, 17331-1-AP, Proteintech, Rosemont, IL, USA)
ATP5C1 (1:1000, 60284-1-AP, Proteintech, Rosemont, IL, USA)
UQCRC2 (1:200, sc-390378, Santa Cruz Biotechnology Inc., Santa Cruz, CA, USA)
MAIP1 (1:500 ,24930-1-AP, Proteintech, Rosemont, IL, USA)
ECHS1 (1:200, sc-515270, Santa Cruz Biotechnology Inc., Santa Cruz, CA, USA)
MTHFDL1L (1:200, sc-376722, Santa Cruz Biotechnology Inc., Santa Cruz, CA, USA)

SLP2 (1:1000, 10348-1-AP, Proteintech, Rosemont, IL, USA)
CHCHD4 (1:1000, 21090-1-AP, Proteintech, Rosemont, IL, USA)

YMEL1 (1:1000, 11510-1-AP, Proteintech, Rosemont, IL, USA)
PARL (1:1000, 26679-1-AP, Proteintech, Rosemont, IL, USA)
ClpB (1:1000, A9130, ABclonal, Woburn, MA, USA)

pHSPB1 phospho-ser15 (1:500, A00343, Genscript Corporation, Piscataway, NJ, USA)
pHSPB1 phospho-ser15 (1:1000, ab5581, Abcam, Cambridge, UK)
pHSPB1 phospho-ser78 (1:500, A00528, Genscript Corporation, Piscataway, NJ, USA)
pHSPB1 phospho-ser78 (1:1000, ab32501, Abcam, Cambridge, UK
pHSPB1 phospho-ser82 (1:500, A00530, Genscript Corporation, Piscataway, NJ, USA)
pHSPB1 phospho-ser82 (1:1000, ab155987, Abcam, Cambridge, UK)

anti-Giantin (1:100, sc-46993, Santa Cruz Biotechnology Inc., Santa Cruz, CA, USA)
anti-KDEL (1:100, ADI-SPA-827, Enzo Life Sciences, Farmingdale, NY, USA)

HRP Goat anti-Rabbit (1:10,000, 111-035-144, Jackson ImmunoResearch Laboratories)
HRP Goat anti-Mouse (1:10,000, 115-035-146, Jackson ImmunoResearch Laboratories)

Alexa Fluor 488 Goat anti-Mouse (1:500, A11001, Life Technologies)
Alexa Fluor 488 Goat anti-Mouse G1 (1:500, A21121, Life Technologies)
Alexa Fluor 488 Donkey anti-Mouse (1:500, A21206, Life Technologies)
Alexa Fluor 488 Donkey anti-Goat (1:500, A11055, Life Technologies)
Alexa Fluor 488 Donkey anti-Rabbit (1:500, A21206, Life Technologies)
Alexa Fluor 546 Goat anti-Mouse G1 (1:500, A21123, Life Technologies)
Alexa Fluor 594 Goat anti-Mouse G1 (1:500, A21125, Life Technologies)
Alexa Fluor 647 Goat anti-Mouse G2a (1:500, A21241, Life Technologies)

Protein A gold 10nm (1:25, Department of Cell Biology, University of Utrecht)

| Validation | Antibodies were selected based on their use in other publications and/or validation by the manufacturers for their respective application. Where possible we used knockdown or knockout cell lines to validate specificity further. Alternatively, some antibodies were validated by verifying their known response to stress stimuli such as increased expression upon heat shock. A last layer of validation was obtained through cell fractionation experiments in which we separated mitochondria from cytosol, allowing us to verify the specificity of antibodies based on the cellular localization of the target protein. |
|---|---|

# Eukaryotic cell lines

Policy information about cell lines

| Cell line source(s) | All cell lines (HeLa, HEK293T, NSC34, Neuro-2a, SH-SY5Y, COS1, A498,C2C12) were acquired from the American Type Culture Collection (ATCC). Primary human-derived lines (lymphoblasts and fibroblasts) were obtained from healthy volunteers according to protocols approved by the local Medical Ethics Committee (University of Antwerp, Belgium). Tafazzin knockout HEK293 Flp-In cells were generously shared by Steven Claypool (The Johns Hopkins University School of Medicine, USA). The origina HEK293 Flp-In cells were obtained from Invitrogen. |
|---|---|
| Authentication | Authentication was performed upon first arrival in the lab (through morphological and STR analysis). |

| Mycoplasma contamination | All cell lines were routinely tested for mycoplasma contamination. All cell lines were negative. |
| Commonly misidentified lines (See ICLAC register) | The cell lines used in this study are not listed as commonly misidentified cell lines. This was verified in the ICLAC table of commonly misidentified cell lines (version 11). |

# Animals and other organisms

Policy information about studies involving animals; ARRIVE guidelines recommended for reporting animal research

| Laboratory animals | Mouse C57BL6/J (Female - 6 months old) |
| Wild animals | This study did not involve wild animals. |
| Field-collected samples | This study did not involve field-collected samples. |
| Ethics oversight | Animal procedures complied with all relevant ethical regulations and were carried out in accordance with European, national and institutional guidelines and with approval by the Medical Ethics Committee (20/36/461) and Ethical Committee for Animal Experiments (2022-13) of the University of Antwerp, Belgium. |

Note that full information on the approval of the study protocol must also be provided in the manuscript.

# Human research participants

Policy information about studies involving human research participants

| Population characteristics | Whole blood was collected from healthy donor (female) for generation of lymphoblast cell lines. Fibroblast cell lines were established from punch biopsies of healthy donor (male). |
| Recruitment | Donors were recruited as volunteers and received no remuneration. These are adult volunteers interested to contribute to scientific research, and have been recruited on free will. |
| Ethics oversight | Blood and skin collection was performed according to the guidelines of the the local Medical Ethics Committee (University of Antwerp, Belgium). Healthy donors are anonymous individuals who signed an informed consent prior to blood/skin collection for research use only and at no cost. |

Note that full information on the approval of the study protocol must also be provided in the manuscript.

# Flow Cytometry

## Plots

Confirm that:

☒ The axis labels state the marker and fluorochrome used (e.g. CD4-FITC).

☒ The axis scales are clearly visible. Include numbers along axes only for bottom left plot of group (a 'group' is an analysis of identical markers).

☒ All plots are contour plots with outliers or pseudocolor plots.

☒ A numerical value for number of cells or percentage (with statistics) is provided.

## Methodology

| Sample preparation | HeLa cells were loaded with 100nM TMRE (tetramethylrhodamine ethyl ester perchlorate, ENZ-52309, Enzo Life Sciences, Farmingdale, NY, USA) for 30 min at 37°C. Cells were washed in PBS, trypsinized and single cells were resuspended in PBS with 2% (v/v) fetal bovine serum prior to analysis. Flow cytometry was performed on a LSR Fortessa cytometer (BD Biosciences). Data analysis was performed with FlowJo10 software (Tree Star Inc., Ashland, OR, USA). |
| Instrument | LSR Fortessa cytometer (BD Biosciences) |
| Software | FlowJo10 software (Tree Star Inc., Ashland, OR, USA) |
| Cell population abundance | All cells were included except those that were not viable. |
| Gating strategy | Gating was optimized for wild type control HeLa cells and subsequently applied to the sHSP-depleted lines using the same parameter settings. Viable singlet cells were analyzed and separated from potential dead cells or doublets based on scatter. |

☒ Tick this box to confirm that a figure exemplifying the gating strategy is provided in the Supplementary Information.

