## [Peer Review File · Nature Cell Biology]

Peer Review Information

Journal: Nature Cell Biology

Manuscript Title: Small heat shock proteins operate as molecular chaperones in the mitochondrial intermembrane space

Corresponding author name(s): Professor Vincent Timmerman

Editorial Notes:

Reviewer Comments & Decisions:

Decision Letter, initial version:

*Please delete the link to your author homepage if you wish to forward this email to co-authors.

Dear Professor Timmerman,

Thank you for submitting your manuscript, "Small heat shock proteins operate as molecular chaperones in the mitochondrial intermembrane space", to Nature Cell Biology. Thank you for your patience with the peer review process. The manuscript has now been seen by 3 referees, who are experts in mitochondrial biology, mitochondrial import (Referee #1); chaperones (Referee #2); and chaperones (Referee #3). As you will see from their comments (attached below), they find this work of potential interest, but have raised substantial concerns, which in our view would need to be

addressed with considerable revisions before we can consider publication in Nature Cell Biology.

As you may know, Nature Cell Biology editors discuss the referee reports in detail within the editorial team, including the chief editor, to identify key referee points that should be addressed with priority, as opposed to requests that are beyond the scope of the current study. To guide the scope of the revisions, I have listed these points below. Our typical revision period is six months; please do not hesitate to contact us to discuss the resubmission timeline or if you anticipate any issues addressing the points of major concerns. We are committed to providing a fair and constructive peer-review process and would be happy to discuss the referee comments further.

In our view, for resubmission to NCB, it would be essential to:

A. Provide additional, strong evidence for the IMS localization of the chaperones and for their functional association of the HSPBs with IMS proteins as delineated by Reviewer #1 in point #1 and by Reviewer #2. Efforts should be made to rule out contributions from non-IMS-localized protein quality control systems, as suggested by Rev#2. For Rev#2, this ties into a greater conceptual concern that it is unclear how ATP-independent HSPB proteins in the IMS may contribute to mitochondrial protection in the absence of ATP-dependent chaperones, and insight into this question should be provided in revision.

B. Bolster the functional analyses of HSPB1's role in protein aggregation as suggested by Rev#3 in point #2 and addressing Rev#2's questions about Figure 5

C. Experimentally address the referees' questions about the lower-mass band of HSPB1 and its functional relevance (point #1 from Rev#3 and point #2 from Rev#1)

D. Provide more clarity on the roles of small HSPs basally vs under heat shock (Rev #2)

E. All other referee concerns pertaining to strengthening existing data, providing controls, methodological details, clarifications and textual changes should also be addressed.

F. Finally please pay close attention to our guidelines on statistical and methodological reporting (listed below) as failure to do so may delay the reconsideration of the revised manuscript. In particular please provide:

We would be happy to consider a revised manuscript that would satisfactorily address these points, unless a similar paper is published elsewhere, or is accepted for publication in Nature Cell Biology in the meantime.

- ensure that it conforms to our format instructions and publication policies (see below and <https://www.nature.com/nature/for-authors>).
- provide a point-by-point rebuttal to the full referee reports verbatim, as provided at the end of this letter.
- provide the completed Reporting Summary (found here <https://www.nature.com/documents/nr-reporting-summary.pdf>). This is essential for reconsideration of the manuscript will be available to editors and referees in the event of peer review. For more information see <http://www.nature.com/authors/policies/availability.html> or contact me.

When submitting the revised version of your manuscript, please pay close attention to our [Digital Image Integrity Guidelines](https://www.nature.com/nature-research/editorial-policies/image-integrity). and to the following points below:

Nature Cell Biology is committed to improving transparency in authorship. As part of our efforts in this direction, we are now requesting that all authors identified as 'corresponding author' on published papers create and link their Open Researcher and Contributor Identifier (ORCID) with their account on the Manuscript Tracking System (MTS), prior to acceptance. ORCID helps the scientific community achieve unambiguous attribution of all scholarly contributions. You can create and link your ORCID from the home page of the MTS by clicking on 'Modify my Springer Nature account'. For more information please visit www.springernature.com/orcid.

This journal strongly supports public availability of data. Please place the data used in your paper into a public data repository, or alternatively, present the data as Supplementary Information. If data can only be shared on request, please explain why in your Data Availability Statement, and also in the correspondence with your editor. Please note that for some data types, deposition in a public repository is mandatory - more information on our data deposition policies and available repositories appears below.

[Redacted]

We hope that you will find our referees' comments and editorial guidance helpful. Please do not hesitate to contact me if there is anything you would like to discuss. Thank you for considering NCB for your work.

Best wishes,

Melina

Melina Casadio, PhD
Senior Editor, Nature Cell Biology
ORCID ID: <https://orcid.org/0000-0003-2389-2243>

Reviewers' Comments:

Reviewer #1:

Remarks to the Author:

The mitochondrial intermembrane space (IMS) is an important compartment for the biogenesis of mitochondria, the transfer of metabolites and the communication of mitochondria with the remainder of the cell. The functional organization of its protein complement is only poorly understood. Molecular chaperones of the well established chaperone families play crucial roles in nearly all cellular compartments, however, in our current understanding the mitochondrial IMS is largely devoid of general chaperone machineries. IMS-specific chaperones dedicated to protein translocation such as the small TIM chaperones have been characterized in detail and a recent study provided evidence that the ATP-dependent Skd3/ClpB may function as disaggregase in the IMS, though it is open how general the role of Skd3/ClpB may be.

This manuscript reports the unexpected observation that small heat shock proteins, a large family of ATP-independent cytosolic chaperones, possess a dual localization. In addition to their predominant localization in the cytosol, a fraction of numerous family members are shown to reside in the IMS. By careful fractionation and protein import experiments the authors demonstrate the dual localization of small heat shock proteins. As important control they observe that many family members, but not all, display a dual localization in cytosol and IMS. Depletion of small heat shock proteins leads to profound alterations of mitochondrial morphology and functions and to the aggregation of proteins located in or exposed to the IMS, including numerous transmembrane proteins of the inner mitochondrial membrane. In addition, this study reveals mechanistic insight in the striking link of Charcot-Marie-Tooth disease-causing mutations, specific small heat shock proteins and mitochondria. The experiments are performed very well and the paper is clearly structured and written.

The following points should be addressed:

1. The protein aggregation assay shown in Figure 5d is important for demonstrating the role of small heat shock proteins in the IMS, yet the presented dataset is limited. Experiments using this assay should be expanded, including quantifications in order to present these important findings not only with single gel lanes.

2. Import of small heat shock proteins into purified mitochondria (Figure 2). The authors conclude: "The import assay with recombinantly produced HSPB1 also revealed the appearance of a lower band for HSPB1, which is proteinase K resistant. This could suggest that HSPB1 undergoes processing inside mitochondria."

To suggest a processing of HSPB1 inside mitochondria further experimental data would be required. The authors should either provide further evidence or mitigate this conclusion.

3. The headings of supplemental Figures 5 and 6 may be misleading (S5: The majority of small heat shock proteins reside in the mitochondrial intermembrane space; S6: Overexpression of untagged small heat shock proteins demonstrates that most of them reside in the mitochondrial intermembrane space).

I see what the authors intend to state but this may be misinterpreted as a major localization of small heat shock proteins in the IMS. As the authors clearly demonstrate in the paper and outline in the main text, the bulk of small heat shock proteins resides in the cytosol and a fraction of numerous small heat shock proteins is located in the IMS. Therefore the headings should be rewritten.

Reviewer #2:

Remarks to the Author:

The authors describe that certain small Hsp may be present in the inter membrane space of mitochondria (and/or go there upon heat stress) to protect against the noxious effects of protein misfolding/aggregation on mitochondrial function. While the mitochondria matrix does contain several chaperones to ensure proper protein quality control and maintain protein homeostasis, the mitochondrial intermembrane space (IMS) lacks such chaperones, except for a few AAA+protease Skd3 (ClpB) that also can help dealing with protein aggregates in this compartment (Cupo & Shorter, Elife 2020). The small HspB members are mere 'holdases' and thus distinct from Skd3, although a (functional) connection between Skd3 and the HspBs in the IMS may exist (see major comments). While HSPB proteins have generally not been found in mitochondria of mammalian cells, some related small HSPs have been suggested to be mitochondria-localized in metazoan, albeit to my understanding this usually was suggested to be in the mitochondrial matrix and not the IMS.

My difficulty in understanding the data in the current paper relates to how these ATP-independent HSPB proteins in the IMS may actually lead to mitochondrial protection when ATP-dependent chaperones are absent here. The dogma is that HSPBs require such ATP-dependent chaperones for 'substrate processing'. The authors propose a nice analogy with the periplasm of Ecoli (for good reasons assumed to be the equivalent of the eukaryotic IMS) where ATP-independent chaperones like Spy also have protective properties in an environment without both ATP and (thus) ATP -dependent chaperones. However, the properties of proteins like Spy that make them independent of ATP-dependent chaperones for client release/refolding, have -to my knowledge- not been described for HSPB proteins so far. Alternatively, one could assume that HSPB-chaperoned clients may be a

substrate for processing by e.g. the Skd3 proteases: however, whereas the authors also speculated on this possibility in their discussion, this was not tested here.

Finally, unlike the IMC protease Skd3, the mammalian HSPB members lack a mitochondrial leader sequence, which makes one wonder how (mechanistically) they end up in the IMC. The authors recognized this and proposed a Mia40 (i.e leader-independent) translocation. Whereas this is -of course- possible, this means that this than unequivocally must be demonstrated. However, I am not fully convinced by the data yet (but could be wrong) that (some of) the HSPB members are truly resident inside the IMS.

Having said so, the idea is attractive, and the combined data set would all be consistent with such a possibility.

However, as indicated below, I have questions and doubts about (parts in) each of the individual main figures. Combined with the above-mentioned, more theoretical considerations, this yet refrains me from being convinced that some HSPBs indeed reside inside the IMS and that the observed functional features are directly due to chaperoning activities inside the IMS.

Major comments:

Figure 1:

As HSPBs lack mitochondrial leader sequences, it is of greatest importance for conclusively demonstrate that they really get INSIDE the IMS, to exclude the possibility that much of the (great) functional data presented later can be due to indirect effects. Figure 1 is therefore key to this paper and several observations made me not fully convinced and I have a whole series of questions on this figure, many of which might be easily explained but I just feel it is important to make sure...

- What wonders me is that there is always only a very minor fraction of the small HSPBs associated with the mitochondrial fraction in unheated cells: these truly only seem minor traces. To me, this and other arguments suggest the possibility of (post-lysis) association of these HSPs with mitochondrial membranes. In fact, the data in figure 1e,f argue in that direction (also indicating they are not integral membrane proteins, but still...). Such associations with membranes may be next augmented when cells are heat shocked due to temperature induced fluidization of membranes and maybe even consequential collapse/aggregation of certain integral membrane proteins to which the HSPBs next bind as holdases. In turn, all such interactions with (heat-altered) membranes may also affect what happens during the isolation mitoblasts and the proteinase K digestibility.

What I certainly disagree with is the statement (lines 115-118) that this suggests "that HSPBs may fulfil a dual function which is different under basal and stress conditions". Whereas I do not exclude the possibility of moonlighting cell biological impact of HSPBs, it seems extremely unlikely (and it certainly cannot be deduced from these data) that the biochemical functions of HSPBs differ when in these different cellular locales).

- What is somewhat bothersome is the inconsistency between different experiments as HSPB7 is found to be mostly largely absent the mitochondria of unheated C2C12 cells in figure 1, but if fact presented at even higher levels in mitochondrial of control than of heated cells in figure S2.

- What is also striking, yet also confusing, is that the proteinase K treatment was effective in

removing the “extra” HSPB1 from intact mitochondria from heated cells (figure 1B) but had less effect on the HSPB1 that was associated with mitochondria from non-heated cells (Figure S4). But also here, not all data are consistent. First, the amount of HSPB1 in the mitochondrial fraction of control cells seems much higher in Figure S4 than in Figure 1. In addition, the amount of HSPB2 and HSPB8 detected in the intact mitochondria of heat shocked cells was generally reduced largely upon proteinase K treatment (Figure 1C,D, figure S5). In other cell lines, some other HSPBs also showed (some) proteinase K sensitivity in intact mitochondria (e.g. HSPB6 in NSC34 and fibroblasts; HSPB1 in SH-SY5Y and lymphoblasts), hinting to the possibility of non-specific associations with (heat altered) mitochondrial membranes. In any case, for different HSPBs and in different cell types, different patterns were apparently seen. Whereas this is, as such, OK it should not remain unnoted, and conclusions should not be generalized.

- Finally, HSPB2 and HSPB3 are almost always found to partner in mixed oligomers. It is therefore striking to see HSPB2 but not HSPB3 to be enriched in heat shocked mitochondria, but yet see that they did both became associated when overexpressed, but sensitive to proteinase K in intact mitochondria (Figure S6), meaning that are mitochondrial associated, but unlikely in the IMS. BTW: it is unclear from the legend in figure S6, but I assume this is all from heat-shocked cells? In fact, control, non-heat shocked samples should also be shown in case of these overexpression studies.

Obviously, from these data, the generic conclusion (lines 138-141) that “these biochemical fractionation experiments show that most sHSPs reside in the mitochondrial IMS but interactions with binding partners may reduce their mitochondrial import” is not (yet) warranted.

The subsequent data with expansion light microscopy and especially the EM (both nice!), do indeed show occasional HSPB1 dots that could be consistent with IMS localization, but most of the HSPB1 material seems to be outside the mitochondria with again some in close vicinity to the mitochondria. What neither figure 1c,d and figure S8 indicate is whether these images are from control or from heat shocked cells. In fact, it would be highly worthy to compare the two conditions and get some quantitative data showing more IMS-localized dots after heat shock compared to controls.

Finally on this figure (panel g) whereas it is interesting to see that the mitochondrial associated HSPB1 seems to more in a DTT-sensitive, dimer form than in the cytoplasm, I am not sure how that would add to the evidence that it is truly resident in the IMS.

Figure 2:

Whereas this figure nicely demonstrates that a minor fraction of purified HSPB1 can become protease resistant when incubated with mitochondria, it is mostly gone upon PK treatment and thus behaves largely different from LIAs. Inversely, if I am correct there is even some of this minimal amount of HSPB1 material present after CCCP treatment present, whereas the originally much more abundant, LIAs was no longer present in the mitochondrial treated with CCCP. So, also these data do not fully yet convince me that HSPB1 has really accumulated inside the IMS.

Figure 3:

This figure shows that association with mitochondria is not associated with or influenced by the phosphorylation status. Whereas this could suggest that these mitochondrial association are not directly linked to the holdase functions of HSPB1 (which are strongly reduced in the for the 3D/non-phosphorylated forms), it does raise another question: the non-phosphorylated forms are usually present in much bigger complexes and do not/less well dissociate into monomers/dimers upon heat

shock for which it even seems more difficult now to understand how they could pass the OM to reach the IMS.

In panel A of this figure, it is clear in that this concerns heat-shocked cells. Is this also true for the other panels? Non-heat shock data should also be presented as controls (or the reversed in the other panels concerns non-heated samples).

Furthermore, the protease K protection was done on mitochondria (not also mitoplasts) and can thus certainly do not warrant the conclusion that the various HSPB1 mutants were imported (lines 262-266). The same is true for the suggestive remark on HSPB1 remaining phosphorylated during import (lines 275-278). Even though the authors may not mean to suggest this is required (it is not: the 3A variants show similar features s the wildtype protein), this statement may confuse the reader.

Figure 4:

I am a bit confused by the labelling of panel A, but if I am correct, the authors used single knockouts (of B1 or B8) and next silenced the other HSPB (B8 or B1) by shRNA. The obvious first question is why HSPB5 is also lost in these cells. The second one is, why the others not just combined the shRNA for all HSPs to transiently knockdown all together and minimize all kinds of adaptations that may be associated with the permanent knockout of these genes.

For the rest, these experiments clearly revealed that reductions in all 3 of these HSPB proteins affected mitochondrial (but not ER or Golgi) morphology and function. Whereas this is striking, it cannot be excluded that this is not related to direct effects inside the IMS, especially given the minute fraction of these HSPBs being associated with the mitochondria in non-heat shock conditions.

Figure 5:

Panel A is confusing as it shows no SOD1 signal in the whole cell lysate of cells transfected with SOD1 G93A. Also, this mutant is not found in the mitochondria.

Other panel: I do agree that for both the mutant SOD G93A and mutant EndoG N174A (but not the OTC mutant: nice control!) there is a slight increase in HSPB1 association with mitochondria. But, do these proteins truly aggregate in the IMS? This should be shown.

In fact, I know of several papers especially on mutant SOD1 in which it aggregates inside the cytosol and that other non-mitochondrial chaperones can reduced this (and hence likely downstream) and damage to and co-aggregation with mitochondria. Can this not also explain the data in panel D?

Furthermore, this panel is incomplete as it lacks the panel of the insoluble material for SOD1-WT and the input fraction before detergent lysis. Also, where is soluble mutant SOD1 in the WT cells and the sHSP depl (2) cells? It is (to me) impossible to evaluate this panel in terms of aggregation.

Figure 6 & 7:

The use of HSPB1-mutants as trap to identify putative clients is smart at first sight but also comes with several questions. It assumes that the mutant is unaffected in both its substrate specificity and has not lost (but rather increased) its binding affinity to all substrates in general. However, even if so, do to lost chaperone activity the mutant protein may aggregate with its substrates such that now other proteins become secondary interactors with these primary misfolded HSPB1-bound substrates. Given that the affinity-enriched MS was next done on whole cell lysates, this even further increases the risks of post-lysis association with already sticky mutant HSPB1-aggregates...

Yet, I agree it is striking that the interactome was enriched for mitochondrial proteins. Question again is also here whether this was done for heat shocked or non-heat shocked cells and whether one should thus not compare these here, as the most important argument from most of the above is that the presumed translocation to the IMS occurs in response to heat shock (or other disturbances in IMS protein homeostasis).

Finally, the findings in panel E that HspB1 binds to the transmembrane domain of SLC25A12 (that should not be exposed in intact mitochondria) to me further supports that much of these data may be due to a post-lysis effect in which HSPB1 (or mutant-associated aggregates) binds to everything that exposes hydrophobic patches. It also refers to my previous remark in figure 1, where I propose that the associations of the HSPBs may related to fluidization of membranes after heat shock that may actually lead to exposure of such hydrophobic transmembrane segments where HSPB proteins next might interact with.

Finally, I do not think the data yet allow the conclusion (discussion lines 625-644) that HSPB1-mutants cause CMT directly by a loss of mitochondrial protein quality control (one, but certainly not the only feature in cells from CMT-derived cells) rather than (also) via defects in cytoplasmic protein quality control.

Minor comments

Lines 43-51: Whereas bacteria indeed also have two small HSP's, there are -to my knowledge so far no data that the reside in between the inner and outer membrane. This would imply that the reported localization in the mitochondrial intermembrane space in metazoan must have occurred later in evolution? Authors comment?

Lines 52-65: This is a bit a circular argument. Whereas the provided reasoning is one possibility, many other non-direct effects of HDAC in relation to (e.g. non-mitochondrially localized HSPB mutants) could also explain the various phenotypes cited here.

Line 66-72: If I am correct, all the studies in plants and metazoans on sHSP in mitochondria refer to matrix-localized versions. This should be checked and if true, the current finding may thus be specific for mammals?

Reviewer #3:

Remarks to the Author:

This work aims to explore the relationship between small heat shock proteins (sHSPs) and mitochondria. Using fractionation and expansion microscopy, they find that HSPB1, 2, 5, 6, 7 and 8 are found in the mitochondria and that a fraction of the total pool is in the inter-membrane space. This is an interesting finding, as a recent study identified a disaggregase in the same sub-compartment, but little else is known about the chaperones of the IMS. The authors proceed to explore the functional relevance of this localization, using knockdowns/knockouts, metabolic measurements and expression of aggregation-prone mutant proteins. Finally, they find that CMT-related mutations alter the localization patterns of HSPB1, suggesting that changes in HSPB1 mitochondrial import may contribute to disease pathology.

Overall, this is a carefully designed set of experiments, with excellent controls and well-described logic. For example, the use of the SOD, EndoG and OTC mutants is a rigorous way to explore protein aggregation in the mitochondria with precision. The finding that sHSPs are an important chaperone of the IMS is an important finding – and the extension to the CMT mutations further improves the impactful.

1. Fig 2. The lower mass band of HSPB1 is interesting and unexpected, but not further explored in the subsequent studies. Because later experiments suggest that this pool of IMS-localized HSPB1 is active as a chaperone, it seems worth exploring what this band represents by mass spectrometry and/or Edmond sequencing. Is this change in molecular mass upon IMS localization observed for other sHSPs?
2. Fig 5. The knockout and knockdown studies (termed sHSP^{-/-}) are interesting and supportive of a key role for the proteins in mitochondrial quality control. However, the study would be strengthened by analysis of an over-expression phenotype. Does OE of HSPB1 protect against SOD1 aggregation? Such a finding would draw a more direct relationship between the limiting activity of the chaperones and protein aggregation.
3. It seems that the authors used three biological replicates for each experiment - but this was not always clear. It would be worth adding specific information about the number of technical and biological replicates to each figure legend.

Minor:

1. BAG3 is a stress responsive protein, whose levels are significantly elevated by heat stress (42°C). Thus, it seems likely that heat shock reduces partitioning of sHSPs into the IMS, in part, because BAG3 levels are elevated. The authors nearly suggest this possibility, but it would be worth blotting for BAG3 in the heat shocked lysates to confirm and strengthen this conclusion.
2. The position of molecular weight markers should be included in the western blots.

Methods should be written concisely, but should contain all elements necessary to allow interpretation and replication of the results. As a guideline, Methods sections typically do not exceed 3,000 words.

The Methods should be divided into subsections listing reagents and techniques. When citing previous methods, accurate references should be provided and any alterations should be noted. Information must be provided about: antibody dilutions, company names, catalogue numbers and clone numbers for monoclonal antibodies; sequences of RNAi and cDNA probes/primers or company names and catalogue numbers if reagents are commercial; cell line names, sources and information on cell line identity and authentication. Animal studies and experiments involving human subjects must be reported in detail, identifying the committees approving the protocols. For studies involving human subjects/samples, a statement must be included confirming that informed consent was obtained. Statistical analyses and information on the reproducibility of experimental results should be provided in a section titled "Statistics and Reproducibility".

All Nature Cell Biology manuscripts submitted on or after March 21 2016 must include a Data availability statement as a separate section after Methods but before references, under the heading "Data Availability". For Springer Nature policies on data availability see <http://www.nature.com/authors/policies/availability.html>; for more information on this particular policy see <http://www.nature.com/authors/policies/data/data-availability-statements-data-citations.pdf>. The Data availability statement should include:

- Accession codes for primary datasets (generated during the study under consideration and designated as "primary accessions") and secondary datasets (published datasets reanalysed during the study under consideration, designated as "referenced accessions"). For primary accessions data should be made public to coincide with publication of the manuscript. A list of data types for which submission to community-endorsed public repositories is mandated (including sequence, structure, microarray, deep sequencing data) can be found here <http://www.nature.com/authors/policies/availability.html#data>.
- Unique identifiers (accession codes, DOIs or other unique persistent identifier) and hyperlinks for datasets deposited in an approved repository, but for which data deposition is not mandated (see here for details <http://www.nature.com/sdata/data-policies/repositories>).
- At a minimum, please include a statement confirming that all relevant data are available from the authors, and/or are included with the manuscript (e.g. as source data or supplementary information), listing which data are included (e.g. by figure panels and data types) and mentioning any restrictions on availability.
- If a dataset has a Digital Object Identifier (DOI) as its unique identifier, we strongly encourage including this in the Reference list and citing the dataset in the Methods.

We recommend that you upload the step-by-step protocols used in this manuscript to the Protocol Exchange. More details can found at www.nature.com/protocolexchange/about.

FIGURES – Colour figure publication costs \$600 for the first, and \$300 for each subsequent colour figure. All panels of a multi-panel figure must be logically connected and arranged as they would

appear in the final version. Unnecessary figures and figure panels should be avoided (e.g. data presented in small tables could be stated briefly in the text instead).

All imaging data should be accompanied by scale bars, which should be defined in the legend. Cropped images of gels/blots are acceptable, but need to be accompanied by size markers, and to retain visible background signal within the linear range (i.e. should not be saturated). The boundaries of panels with low background have to be demarked with black lines. Splicing of panels should only be considered if unavoidable, and must be clearly marked on the figure, and noted in the legend with a statement on whether the samples were obtained and processed simultaneously. Quantitative comparisons between samples on different gels/blots are discouraged; if this is unavoidable, it should only be performed for samples derived from the same experiment with gels/blots were processed in parallel, which needs to be stated in the legend.

Regardless of format, all figures must be vector graphic compatible files, not supplied in a flattened raster/bitmap graphics format, but should be fully editable, allowing us to highlight/copy/paste all text and move individual parts of the figures (i.e. arrows, lines, x and y axes, graphs, tick marks, scale bars etc.). The only parts of the figure that should be in pixel raster/bitmap format are photographic

images or 3D rendered graphics/complex technical illustrations.

The total number of Supplementary Figures (not including the “unprocessed scans” Supplementary Figure) should not exceed the number of main display items (figures and/or tables (see our Guide to Authors and March 2012 editorial <http://www.nature.com/ncb/authors/submit/index.html#suppinfo>; <http://www.nature.com/ncb/journal/v14/n3/index.html#ed>). No restrictions apply to Supplementary Tables or Videos, but we advise authors to be selective in including supplemental data.

Each Supplementary Figure should be provided as a single page and as an individual file in one of our accepted figure formats and should be presented according to our figure guidelines (see above). Supplementary Tables should be provided as individual Excel files. Supplementary Videos should be provided as .avi or .mov files up to 50 MB in size. Supplementary Figures, Tables and Videos must be

accompanied by a separate Word document including titles and legends.

GUIDELINES FOR EXPERIMENTAL AND STATISTICAL REPORTING

REPORTING REQUIREMENTS – We are trying to improve the quality of methods and statistics reporting in our papers. To that end, we are now asking authors to complete a reporting summary that collects information on experimental design and reagents. The Reporting Summary can be found here <https://www.nature.com/documents/nr-reporting-summary.pdf> If you would like to reference the guidance text as you complete the template, please access these flattened versions at <http://www.nature.com/authors/policies/availability.html>.

Author Rebuttal to Initial comments

Reviewers' Comments:

Reviewer #1:

Remarks to the Author:

The mitochondrial intermembrane space (IMS) is an important compartment for the biogenesis of mitochondria, the transfer of metabolites and the communication of mitochondria with the remainder of the cell. The functional organization of its protein complement is only poorly understood. Molecular chaperones of the well-established chaperone families play crucial roles in nearly all cellular compartments, however, in our current understanding the mitochondrial IMS is largely devoid of general chaperone machineries. IMS-specific chaperones dedicated to protein translocation such as the small TIM chaperones have been characterized in detail and a recent study provided evidence that the ATP-dependent Skd3/ClpB may function as disaggregase in the IMS, though it is open how general the role of Skd3/ClpB may be.

This manuscript reports the unexpected observation that small heat shock proteins, a large family of ATP-independent cytosolic chaperones, possess a dual localization. In addition to their predominant localization in the cytosol, a fraction of numerous family members are shown to reside in the IMS. By careful fractionation and protein import experiments the authors demonstrate the dual localization of small heat shock proteins. As important control they observe that many family members, but not all, display a dual localization in cytosol and IMS. Depletion of small heat shock proteins leads to profound alterations of mitochondrial morphology and functions and to the aggregation of proteins located in or exposed to the IMS, including numerous transmembrane proteins of the inner mitochondrial membrane. In addition, this study reveals mechanistic insight in the striking link of Charcot-Marie-Tooth disease-causing mutations, specific small heat shock proteins and mitochondria. The experiments are performed very well and the paper is clearly structured and written.

We would like to thank the reviewer for taking the time to read our manuscript carefully and we appreciate the kind words about our manuscript. Please find our response to the points below.

The following points should be addressed:

1. The protein aggregation assay shown in Figure 5d is important for demonstrating the role of small heat shock proteins in the IMS, yet the presented dataset is limited. Experiments using this assay should be expanded, including quantifications in order to present these important findings not only with single gel lanes.

We thank the reviewer for this comment and have followed the suggestion to extend this section substantially.

First, we added the quantifications related to Figure 5d, which confirm that the mutant SOD1-G93A protein aggregates more in the absence of small heat shock proteins (sHSPs). The quantification has now been added to the Figure 5 (figure legend is adapted in the revised manuscript).

e

We next verified whether overexpression of sHSPs would reduce the level of aggregated SOD1-G93A. This was indeed the case. Upon overexpression of HSPB1-B5-B8, we observed reduced insoluble SOD1-G93A and this was accompanied by increased soluble SOD1-G93A. This further supports our data obtained with sHSP-depleted cell lines, which suggested a direct interaction between the chaperones and mutant SOD1. We added this result as a new panel to Figure 5.

d

We next wondered whether we could also find evidence for the accumulation of other substrates in mitochondria that lack sHSPs. To this end, we purified mitochondria from control and sHSP-depleted HeLa cells and analyzed the insoluble fractions of these mitochondria with mass spectrometry. Unfortunately, we could not identify substrates that accumulate in the absence of sHSPs. In contrast to the SOD1-G93A mutant, which is a pathological mutant and known to be resistant to degradation, we suspect that these non-pathological substrates may still be degraded by the (very potent) protease system of mitochondria (Deshwal et al. 2020 Annu Rev Biochem). This notion is supported by the observation that even in cells in which individual mitochondrial proteases were knocked out, substrates did not accumulate either (unpublished data from the Langer lab). An alternative explanation for the absence of accumulating substrates could be that more abundant proteins masked the (less abundant) aggregating proteins. We presume the latter is unlikely as our proteomics experiments obtained considerable depth but we cannot exclude this possibility.

As proteomics could not provide additional support for an important chaperone-role of sHSPs in mitochondria, we decided to analyze additional conditions in which we would perturb

mitochondrial proteostasis and then evaluate whether this would provoke a response from sHSPs:

- (i) In the first experiment we depleted the IMS-chaperone ClpB. Knockdown of ClpB does indeed lead to increased recruitment of sHSPs to mitochondria, suggesting an interplay and potential compensation mechanism between both IMS chaperone systems. We added this experiment as a new panel to Figure 5 (figure legend is adapted in the revised manuscript).

- (ii) Earlier work of the Langer lab revealed that in retinas of mice with a neuron-specific deletion of the IMS-facing YME1L protease, HSPB1 is one of the most upregulated proteins (as detected by proteomics in Figure 1 from Sprenger et al. 2021 Nat Metab). This suggests that dysregulation of mitochondrial proteostasis may lead to activation of mitochondrial chaperones like the sHSPs. We therefore verified whether we could phenocopy this in our HeLa cell model (for consistency with our other stress experiments) and developed cell lines in which we depleted mitochondrial proteases YME1L, PARL, and OMA1. This revealed that also in HeLa cells, we could detect increased amounts of sHSPs in mitochondria with reduced levels of YME1L/PARL and YME1L/PARL/OMA1. We added these data as a new panel to Figure 5.

- (iii) As we were able to identify candidate substrates that are chaperoned by sHSPs in Figure 6, we used this knowledge to see if we could find further evidence for a chaperone-role for these substrates specifically. To this end, we used TAZ-knockout HEK293 Flp-In cells (obtained from Steven Claypool, Johns Hopkins, Baltimore, USA) as the deletion

of *Tafazzin* prevents the synthesis of the mitochondrial phospholipid cardiolipin, which results in stress for transmembrane proteins that interact with cardiolipin. Among these, the mitochondrial solute carriers (SLC25-family) were identified as sensitive to Tafazzin-depletion as cardiolipin stabilizes the carrier's folded state (Senoo et al. 2020 Sci Adv). We therefore hypothesized that partial unfolding of mitochondrial solute carriers would be sensed by sHSPs. Indeed, we found increased levels of sHSPs in mitochondrial fraction of TAZ-knockout cells. The different clones had varying levels of HSPB1, possibly the result of clonal variability, but even in clones with drastically reduced total levels of soluble HSPB1, we still observed increased mitochondrial translocation. This therefore supports the hypothesis that sHSPs respond to protein stress inside mitochondria. We added these data as a new panel to Figure 6.

(iv) Finally, we also provided additional evidence for the chaperone-client interaction between EndoG and HSPB1. We verified whether the HSPB1 3A- and 3D-mutant engaged in equal amounts with misfolded EndoG-N174A. To this end, we generated stable cell lines overexpressing HSPB1 WT/3A/3D and transiently transfected the cells with EndoG-N174A. We hypothesized that the 3D-variant (mimicking the hyperactive chaperone) would engage most with the misfolded client protein while the 3A-mutant (mimicking the hypoactive chaperone) would engage the least with the misfolded client protein. To verify this, we purified mitochondria from the different cell lines and separated the soluble and insoluble fraction. In line with our hypothesis, we observed that the 3D-mutant (mimicking the hyperactive variant) engaged most with the misfolded substrate in the insoluble fraction. The 3A-mutant engaged the least with the misfolded substrate, as evidenced by the reduced presence in the insoluble fraction. This therefore provides further evidence for a direct chaperone-substrate interaction with the misfolded client model substrate EndoG_N174A, similarly as the SOD1-G93A.

2.Import of small heat shock proteins into purified mitochondria (Figure 2). The authors conclude: “The import assay with recombinantly produced HSPB1 also revealed the appearance of a lower band for HSPB1, which is proteinase K resistant. This could suggest that HSPB1 undergoes processing inside mitochondria.” To suggest a processing of HSPB1 inside mitochondria further experimental data would

be required. The authors should either provide further evidence or mitigate this conclusion.

We thank the reviewer for this comment and have addressed this experimentally. After some optimization, we were able to immunoprecipitate sufficient amounts of V5-tagged HSPB1 from purified mitochondria (pretreated with proteinase K to degrade any cytosolic contamination) and cut the band from a Coomassie stained gel before submitting the sample for mass spec analysis.

Using chymotrypsin for protein digestion, we were able to obtain sufficient coverage of the N-terminus which was not possible with a standard trypsin digest. This experiment revealed that imported HSPB1 is modified at its N-terminus. The N-terminal methionine is cleaved off and the remaining peptide is modified by acetylation, which might explain the slight shift on the gel as reported in our initial manuscript.

We have added the MS/MS spectrum as supplementary figure and modified the text accordingly:

Mass spectrometric analysis of this lower band revealed that mitochondrial HSPB1 had undergone an N-terminal modification, possibly explaining the size shift on gel (**Figure S11**).

3.The headings of supplemental Figures 5 and 6 may be misleading (S5: The majority of small heat shock proteins reside in the mitochondrial intermembrane space; S6: Overexpression of untagged small heat shock proteins demonstrates that most of them reside in the mitochondrial intermembrane space). I see what the authors intend to state but this may be misinterpreted as a major localization of small heat shock proteins in the IMS. As the authors clearly demonstrate in the paper and outline in the main text, the bulk of small heat shock proteins resides in the cytosol and a fraction of numerous small heat shock proteins is located in the IMS. Therefore the headings should be rewritten.

We thank the reviewer for pointing this out, as we had not thought of this ourselves, and agree that the figure titles could be interpreted incorrectly. We have therefore amended the titles to avoid they would mislead the reader and hope that the new figure titles are better.

Suppl Figure 5

Original:

Suppl. Figure 5. The majority of small heat shock proteins reside in the mitochondrial intermembrane space

Changed to:

Suppl. Figure 5. A fraction of most members of the small heat shock protein family reside in the mitochondrial intermembrane space

Suppl Figure 6

Original:

Suppl. Figure 6. Overexpression of untagged small heat shock proteins demonstrates that most of them reside in the mitochondrial intermembrane space

Changed to:

Suppl. Figure 6. Overexpression of untagged small heat shock proteins demonstrates that a fraction of most family members reside in the mitochondrial intermembrane space

Reviewer #2:

Remarks to the Author:

The authors describe that certain small Hsp may be present in the inter membrane space of mitochondria (and/or go there upon heat stress) to protect against the noxious effects of protein misfolding/aggregation on mitochondrial function. While the mitochondria matrix does contain several chaperones to ensure proper protein quality control and maintain protein homeostasis, the mitochondrial intermembrane space (IMS) lacks such chaperones, except for a few AAA+protease Skd3 (ClpB) that also can help dealing with protein aggregates in this compartment (Cupo & Shorter, Elife 2020). The small HspB members are mere 'holdases' and thus distinct from Skd3, although a (functional) connection between Skd3 and the HspBs in the IMS may exist (see major comments). While HSPB proteins have generally not been found in mitochondria of mammalian cells, some related small HSPs have been suggested to be mitochondria-localized in metazoan, albeit to my understanding this usually was suggested to be in the mitochondrial matrix and not the IMS.

My difficulty in understanding the data in the current paper relates to how these ATP-independent HSPB proteins in the IMS may actually lead to mitochondrial protection when ATP-dependent chaperones are absent here. The dogma is that HSPBs require such ATP-dependent chaperones for 'substrate processing'. The authors propose a nice analogy with the periplasm of Ecoli (for good reasons assumed to be the equivalent of the eukaryotic IMS) where ATP-independent chaperones like Spy also have protective properties in an environment without both ATP and (thus) ATP -dependent chaperones. However, the properties of proteins like Spy that make them independent of ATP-dependent chaperones for client release/refolding, have -to my knowledge- not been described for HSPB proteins so far. Alternatively, one could assume that HSPB-chaperoned clients may be a substrate for processing by e.g. the Skd3 proteases: however, whereas the authors also speculated on this possibility in their discussion, this was not tested here.

Finally, unlike the IMC protease Skd3, the mammalian HSPB members lack a mitochondrial leader sequence, which makes one wonder how (mechanistically) they end up in the IMC. The authors recognized this and proposed a Mia40 (i.e leader-independent) translocation. Whereas this is -of course- possible, this means that this than unequivocally must be demonstrated. However, I am not fully convinced by the data yet (but could be wrong) that (some of) the HSPB members are truly resident inside the IMS.

Having said so, the idea is attractive, and the combined data set would all be consistent with such a possibility.

However, as indicated below, I have questions and doubts about (parts in) each of the individual main figures. Combined with the above-mentioned, more theoretical considerations, this yet refrains me from being convinced that some HSPBs indeed reside inside the IMS and that the observed functional features are directly due to chaperoning activities inside the IMS.

We would like to thank the reviewer for taking the time to carefully read our manuscript and are grateful for the comments described above and below. We have addressed the comments with additional experiments and feel that this has strengthened the manuscript considerably. One element we would already like to point out here, and to which we will come back below, is that sHSPs accumulate on the OMM after heat shock and not in the IMS. This is evidenced by their sensitivity to proteinase K after heat shock, after which their pattern mimics that of outer mitochondrial membrane protein TOMM20. Some of the questions and remarks below can therefore already be explained by this. The other points have been addressed experimentally and have been added to the manuscript.

Major comments:

Figure 1:

As HSPBs lack mitochondrial leader sequences, it is of greatest importance for conclusively demonstrate that they really get INSIDE the IMS, to exclude the possibility that much of the (great) functional data presented later can be due to indirect effects. Figure 1 is therefore key to this paper and several observations made me not fully convinced and I have a whole series of questions on this figure, many of which might be easily explained but I just feel it is important to make sure...

• What wonders me is that there is always only a very minor fraction of the small HSPBs associated with the mitochondrial fraction in unheated cells: these truly only seem minor traces. To me, this and other arguments suggest the possibility of (post-lysis) association of these HSPs with mitochondrial membranes. In fact, the data in figure 1e,f argue in that direction (also indicating they are not integral membrane proteins, but still...). Such associations with membranes may be next augmented when cells are heat shocked due to temperature induced fluidization of membranes and maybe even consequential collapse/aggregation of certain integral membrane proteins to which the HSPBs next bind as holdases. In turn, all such interactions with (heat-altered) membranes may also affect what happens during the isolation mitoblasts and the proteinase K digestibility. What I certainly disagree with is the statement (lines 115-118) that this suggests “that HSPBs may fulfil a dual function which is different under basal and stress conditions”. Whereas I do not exclude the possibility of moonlighting cell biological impact of HSPBs, it seems extremely unlikely (and it certainly cannot be deduced from these data) that the biochemical functions of HSPBs differ when in these different cellular locales).

We thank the reviewer for this comment and agree that the statement (lines 115-118) might come across as confusing. We intended to state that the localization of sHSPs differs under basal versus heat shock conditions. Where all of HSPB1 is protected from proteinase K under basal conditions, it becomes subject to proteinase K digestion upon heat shock. The enrichment

of HSPB1 therefore occurs on the OMM after heat shock and not in the IMS. However, we agree that our sentence may make this point insufficiently clear and instead suggest that they would fulfil a function other than chaperoning which may have caused confusion. We have therefore removed the sentence from the manuscript and made other textual changes to avoid any confusion.

The large increase in proteinase K sensitive sHSPs in the mitochondrial fraction upon heat shock is indeed also to us remarkable. The focus of our manuscript is, however, on the basal (non-heat shock) condition and the far majority of our results were obtained in absence of heat shock - ensuring also that for example heat-induced membrane fluidization or other factors cannot influence the interpretation. As stated above, we have made adjustments to the manuscript to make this distinction more explicit.

While it is tempting to conclude from the Western blots that the fraction of total sHSPs that is associated with mitochondria under basal (non-heat shock) conditions is very small, it is important to keep in mind that the mitochondrial IMS represents only a small fraction of the total mitochondrial volume, while the mitochondrial matrix occupies a much larger fraction, so a strong diluting effect of IMS proteins can be expected on Western blot. To overcome this, we used our immuno-EM images to quantify the amount of HSPB1, as it allows to correct for the relative size of the different cellular compartments. These measurements revealed that the concentration of HSPB1 is at least as high, if not higher, in the mitochondrial IMS compared to the cytosol (Figure 1d).

The reviewer's concern that sHSPs would associate post-lysis with mitochondria is a very fair and important criticism. Below we describe the points that made us conclude that our results are not the result of non-specific post-lysis association with mitochondrial membranes. Besides these theoretical points, we also performed additional control experiments to rule out that our results would stem from a technical error.

- (i) if sHSPs would associate with mitochondrial membranes in a non-specific (post-lysis) manner, then one would expect that all sHSPs would react identical in the biochemical assays. However, HSPB3 for instance cannot be detected under basal conditions in the mitochondrial fraction while most other sHSPs actually can. If our results would have stemmed from non-specific post-lysis association, then HSPB3 should also have appeared in the mitochondrial fraction (Fig. 1a). In particular since it is known that HSPB3 can form hetero-oligomers with HSPB2 (Den Engelsman et al. 2009) and we did find HSPB2 in the mitochondrial fraction.
- (ii) if the mitochondrial association would only have occurred post-lysis, then our light and electron microscopy would have revealed this as these methods are performed by fixation or high-pressure freezing of intact cells (without lysis). However, these experiments confirmed our results obtained by biochemical fractionation assays.
- (iii) A last argument that would argue against non-specific post-lysis association comes from the proteinase K experiments as it is not plausible that post-lysis associations become proteinase K resistant.

All of these arguments therefore argue in favour of most sHSPs residing inside mitochondria.

However, as we agree with the reviewer that this is a very important concern, we have performed an experiment to directly test whether sHSPs could associate with mitochondria post-lysis, to rule out that our results would stem from such non-specific mitochondrial association. To this end, we isolated mitochondria from naïve HeLa cells (non-heat shocked

cells) and incubated each mitochondrial pellet for increasing amounts of time with the cytosolic fraction isolated from HeLa cells overexpressing V5-tagged HSPB1. In case sHSPs would associate post-lysis with mitochondria, then one would expect to see an increasing amount of HSPB1-V5 with increasing time of co-incubation. We did however find no evidence for this, as none of the HSPB1-V5 that was co-incubated with purified mitochondria became resistant to proteinase K, further arguing against post-lysis association.

Finally, we also extended our expansion microscopy data. By employing a combination of expansion microscopy and structured illumination microscopy (SIM) we were able to increase resolution (down to approximately 25nm), and show with an even higher level of detail that HSPB1 is present inside mitochondria. We added additional representative images as supplementary figure 9.

Suppl. Figure 9. Mitochondrial-localized HSPB1 imaged using a combination of expansion microscopy and structured illumination microscopy

Non-heat shocked HeLa cells with immunofluorescence staining of endogenous HSPB1 (green) and TOM20 (magenta) as an OMM marker. Samples were fixed and stained according to our standard

protocol, prior to linkage in an expandable hydrogel and imaged with structured illumination microscopy. The overview image is a maximum intensity projection (MIP) of the imaged volume and the zoomed boxes (1 μm x 1 μm) show examples of HSPB1 present inside mitochondria. White markers indicate the position of the orthogonal planes in the volumes (XY, XZ and YZ) and were sliced at HSPB1 spots fully contained by the mitochondrial TOM20 boundaries in all dimensions.

We therefore feel confident based on the arguments above that the conclusions of our biochemical assays are not the result of non-specific post-lysis association. This is further supported by our light and electron microscopy experiments where such post-lysis effects are naturally excluded.

• What is somewhat bothersome is the inconsistency between different experiments as HSPB7 is found to be mostly largely absent the mitochondria of unheated C2C12 cells in figure 1, but if fact presented at even higher levels in mitochondrial of control than of heated cells in figure S2.

We thank the reviewer for this comment and have repeated this experiment to confirm that our results are consistent across experiments. In fact, the experiment shown in Figure S2 is technically very challenging, both at the level of collecting the samples but also at the level of detecting all the different sHSPs by western blot. We are therefore afraid that HSPB7 may have been an outlier in Figure S2. We suspect due to a technical reason such as unequal loading of this blot, as all other proteins show very consistent results.

Below we show the repetition of this experiment and show that HSPB7 also shows the same pattern of other sHSPs as previously shown in Figure S2. We have replaced supplementary Figure 2 with this new replicate.

• What is also striking, yet also confusing, is that the proteinase K treatment was effective in removing the “extra” HSPB1 from intact mitochondria from heated cells (figure 1B) but had less effect on the HSPB1 that was associated with mitochondria from non-heated cells (Figure S4). But also here, not all data are consistent. First, the amount of HSPB1 in the mitochondrial fraction of control cells seems much higher in Figure S4 than in Figure 1. In addition, the amount of HSPB2 and HSPB8 detected in the intact mitochondria of heat shocked cells was generally reduced largely upon proteinase K treatment (Figure 1C,D, figure S5). In other cell lines, some other HSPBs also showed (some) proteinase K sensitivity in intact mitochondria (e.g. HSPB6 in NSC34 and fibroblasts; HSPB1 in SH-SY5Y and lymphoblasts), hinting to the possibility of non-specific associations with (heat

altered) mitochondrial membranes. In any case, for different HSPBs and in different cell types, different patterns were apparently seen. Whereas this is, as such, OK it should not remain unnoted, and conclusions should not be generalized.

We thank the reviewer for carefully checking our data, and the apparent inconsistencies indeed need some clarification.

Firstly, the observation that proteinase K is effective in removing HSPB1 from intact mitochondria from heated cells but not from non-heated cells, goes back to what we stated above, and is explained by the altered dominant subcellular localization of sHSPs after heat shock. The sHSPs are no longer imported after heat shock but, instead, accumulate on the OMM. Hence, they become proteinase K sensitive after heat shock while they are proteinase K resistant under basal conditions. The reviewer's observation is therefore absolutely correct, but is overlooking our point in the manuscript where we state that sHSPs accumulate on the OMM after heat shock (lines 114-115), explaining why they become proteinase K sensitive after heat shock and providing an interpretation for these results.

Secondly, the apparent difference in the amount HSPB1 in the mitochondrial fraction across experiments (Figure 1 versus Figure S4) can easily be explained. It is impossible to compare quantities across different western blot experiments, as the method does not allow such absolute read out. At best, one could make a relative comparison between heat shock and non-heat shocked conditions and compare this across western blots. When one does this, there appears no major difference between both experiments. We suspect that the impression the reviewer got, that there is the amount of HSPB1 in the mitochondrial fraction of control cells is much higher in Figure S4 than in Figure 1, stems from the fact that we were able to expose the western blot from Figure S4 longer as there is no WCL (whole cell lysate) or cytosolic fraction on the membrane, which would overshadow the mitochondrial HSPB1-bands, as those fractions contain bigger bands and would get overexposed. We therefore understand the concern but, as we hoped to clarify, this is the result of technical differences between both blots representing the same underlying changes. We have added a short additional sentence to the legend of figure S4 to avoid any confusion.

With regard to the other mentioned minor differences in proteinase K sensitivity, we would like to mention that, in general, sensitivity to proteinase K would lead to complete degradation of the protein and therefore no band on the gel (or at least a major reduction in signal on the western blot). Specifically, the bands for NSC34 appear a bit unregular in shape for the HSPB6 antibody, and perhaps this may have caused the reviewer to reason that there is proteinase K sensitivity, but this is due to the low expression of HSPB6 in NSC34 cells, which forced us to load a large amount of protein on the gel, resulting in distortion of the bands (note that we run a separate gel for HSPB6 with more protein/lane to be able to detect HSPB6). For HSPB6 in the fibroblast cells, this is only a very minor reduction and falls within the regular variability of the assay. Note that there are some technical challenges associated to the proteinase K assay and having to load protein multiple times on gel to be able to detect the different sHSPs may have resulted in the minor variation of the displayed blot for HSPB6 – however this would not qualify as reduced import and we would also not attach any biological conclusion to such minor reductions. A meaningful reduction would be at minimum what we consistently observed for HSPB8. However, smaller differences than those for HSPB8 would simply be due to the technical challenges associated with the assay and would also not reproduce across replicates – whereas we always observed this same level of reduction for HSPB8 in the condition of intact mitochondria.

The reviewer's comment that the amount of HSPB2 and HSPB8 is reduced upon proteinase K treatment is correct. This was and also our interpretation and we conclude that both proteins are incompletely imported into the mitochondrial IMS and a fraction of them remains on the

outside of mitochondria (hence they are partially sensitive to proteinase K). In our manuscript we speculate this could be due to the fact that these two particular sHSPs have stable binding partners (HSPB3 and BAG3, respectively) and both of these binding partners are not imported into mitochondria, which could lead to import retention of HSPB2 and HSPB8. We have referred to this in the manuscript on lines 136-144.

• Finally, HSPB2 and HSPB3 are almost always found to partner in mixed oligomers. It is therefore striking to see HSPB2 but not HSPB3 to be enriched in heat shocked mitochondria, but yet see that they did both become associated when overexpressed, but sensitive to proteinase K in intact mitochondria (Figure S6), meaning that are mitochondrial associated, but unlikely in the IMS. BTW: it is unclear from the legend in figure S6, but I assume this is all from heat-shocked cells? In fact, control, non-heat shocked samples should also be shown in case of these overexpression studies.

We thank the reviewer for this question and are happy to clarify that these were non-heat shocked samples in figure S6. In fact, other than at the very beginning of our manuscript, all experiments were done with mitochondria isolated from non-heat shocked cells as we observed that sHSPs accumulate on the OMM after heat shock and we decided to focus this manuscript on the role of sHSPs in the IMS (in other words during basal/non-heat shocked conditions). To make sure this is clear, we have added this to the figure legends of Figure S5 and S6.

We agree with the reviewer that there appears to be a dogma in the field that HSPB2 and HSPB3 always have to form mixed oligomers. However, in comparison to some other sHSPs such as HSPB1 or HSPB5, there is much less literature on HSPB2 and HSPB3 and this dogma of mixed oligomers appears to stem from just a handful of older *in vitro* studies. We think that, based on our results and some more recent literature, that HSPB2 and HSPB3 can indeed form mixed oligomers (as initially suggested by den Engelsman et al. 2009), but that they may also act independent from one another (potentially as mixed oligomers with other sHSPs). For instance, in C2C12 cells we observe absolutely no correlation between the translocation of HSPB2 and HSPB3 to mitochondria after heat shock (Fig. 1a). While HSPB2 clearly enriches in the mitochondrial fraction after heat shock, HSPB3 remains undetectable. This in fact, would form another argument why our results do not stem from post-lysis artefacts as in that case one would expect HSPB2 and HSPB3 to have the same pattern of translocation, given that they can form mixed hetero-oligomers. Only upon overexpressing HSPB3 in HeLa cells (Figure S6), which would not naturally express HSPB3, we are able to detect HSPB3 in the mitochondrial fraction. HSPB3 remains however nearly completely sensitive to proteinase K, unlike HSPB2 which is partially protected from proteinase K, suggesting that at least a significant fraction of HSPB2 is imported into the mitochondrial IMS independent from HSPB3.

Our data therefore suggest that HSPB2 and HSPB3 may have roles independent from each other and that perhaps only a pool of HSPB2 is engaged in hetero-oligomers with HSPB3. This idea, that HSPB2 and HSPB3 do not always form mixed hetero-oligomers, is supported by some more recent work from the lab of Serena Carra (Tiago et al. 2021 Cell Death & Dis) where the authors stained HSPB2 and HSPB3 in Figure 1B and one can clearly see that both proteins have distinct subcellular localization, confirming our findings in this manuscript, that HSPB2 and HSPB3 can also exist in separate pools and thereby potentially performing different functions. The authors concluded this as following in their manuscript (Tiago et al. 2021):

Although HSPB2 and HSPB3 form a complex²², these two proteins displayed a different localization in differentiating myoblasts. HSPB2 formed intranuclear phase-separated condensates;¹⁸ HSPB3 was enriched at the NE and formed nuclear filaments, reminiscent of the nuclear lamin meshwork (Fig. 1B). Quantification of endogenous HSPB2 and HSPB3 colocalization in multinucleated myotubes with HSPB2 foci confirmed their distinct subcellular pattern (Fig. 1C). As a control, colocalization of myc and HSPB3 stainings in myoblasts overexpressing myc-HSPB3 was significantly higher; this significance was lost by rotating of 90° the red channel (Fig. 1C). Thus, HSPB2 and HSPB3 exist in separate pools that may exert distinct functions.

More work will thus be required to understand the roles and interdependence of HSPB2 and HSPB3 better. However, at least from our results and those of the Carra lab, it becomes clear that HSPB2 and HSPB3 can function independently from each other and that the dogma of ‘almost always found to partner in mixed oligomers’ is possibly incorrect.

Obviously, from these data, the generic conclusion (lines 138-141) that “these biochemical fractionation experiments show that most sHSPs reside in the mitochondrial IMS but interactions with binding partners may reduce their mitochondrial import” is not (yet) warranted.

We agree with the reviewer that the sentence above, as stated in the original submitted manuscript, was perhaps too speculative. We have therefore attempted to address this experimentally in the following way: we generated BAG3 CRISPR/Cas9 knockout HeLa cells and verified whether the deletion of BAG3 would alter the proteinase K pattern of HSPB8 in the mitochondrial fraction (in other words: whether HSPB8 would be imported more upon loss of its interaction with BAG3). To this end, we rescued the BAG3-KO line with BAG3-WT and BAG3-deltaIPV as the latter is unable to bind HSPB8 (Fuchs et al. 2010). Using these new cell lines, we observed that HSPB8 import retention is drastically reduced in cells expressing BAG3-deltaIPV supporting our hypothesis from the originally submitted manuscript. Interestingly, BAG3-KO cells rescued with BAG3-WT show what could be interpreted as hyper-import retention as we could hardly see any proteinase K resistant HSPB8 in the condition of intact mitochondria (while in naïve HeLa cells approx. 50% of HSPB8 shows resistance to proteinase K). However, as we cannot rule out this could also be due to clonal side effects associated with the CRISPR/Cas9 genome editing, we have left this statement out of the manuscript. We have added this figure as a new supplementary figure S7 and added the conclusion to our manuscript that BAG3-binding may indeed lead to mitochondrial import retention of HSPB8.

The subsequent data with expansion light microscopy and especially the EM (both nice!), do indeed show occasional HSPB1 dots that could be consistent with IMS localization,

but most of the HSPB1 material seems to be outside the mitochondria with again some in close vicinity to the mitochondria. What neither figure 1c,d and figure S8 indicate is whether these images are from control or from heat shocked cells. In fact, it would be highly worthy to compare the two conditions and get some quantitative data showing more IMS-localized dots after heat shock compared to controls.

We thank the reviewer for appreciation of these experiments but we think this comment may concern a misunderstanding as one would not expect more IMS-localized dots after heat shock (as the accumulation occurs at the OMM and not the IMS). To clarify this better, we have amended this in the figure legends in the manuscript and now clearly state that all these images are taken under basal (non-heat shock) conditions as it is the focus of our manuscript.

Finally on this figure (panel g) whereas it is interesting to see that the mitochondrial associated HSPB1 seems to more in a DTT-sensitive, dimer form than in the cytoplasm, I am not sure how that would add to the evidence that it is truly resident in the IMS.

We agree with the reviewer that this experiment on itself is not sufficient to claim IMS-residence. However, it does show that these sHSPs find themselves in a different redox environment and this is consistent with our other observations showing their IMS localization. We therefore think that, combined with our other localization experiments, this forms another piece of indirect evidence for mitochondrial localization.

Figure 2:

Whereas this figure nicely demonstrates that a minor fraction of purified HSPB1 can become protease resistant when incubated with mitochondria, it is mostly gone upon PK treatment and thus behaves largely different from Lias. Inversely, if I am correct there is even some of this minimal amount of HSPB1 material present after CCCP treatment present, whereas the originally much more abundant, Lias was no longer present in the mitochondrial treated with CCCP. So, also these data do not fully yet convince me that HSPB1 has really accumulated inside the IMS.

We thank the reviewer for this comment and are happy to explain this better. The reviewer correctly states that a fraction of HSPB1 becomes proteinase K resistant in this assay, which would indicate mitochondrial import as only then it is protected from proteinase K. HSPB1 is indeed not sensitive to CCCP treatment, which indicates that its mitochondrial import is not dependent on the mitochondrial membrane potential. This is often observed for IMS-localized proteins (Hansen and Herrmann 2019 Protein J) and would therefore be consistent with the idea that sHSPs reside in the IMS. The mitochondrial import of matrix proteins, like Lias, is on the other hand often strictly dependent on the presence of a mitochondrial membrane potential. Dissipation of the membrane potential by the addition of CCCP therefore prevents their mitochondrial import. Our results are thus in line with what would be expected for a mitochondrial IMS-localized protein.

One aspect that has been discussed intensely between the co-authors of this manuscript is why only a fraction of HSPB1 becomes proteinase K sensitive in this assay. While we did not include this in the final manuscript as we felt it was too speculative at this point, we hypothesized that the HSPB1 produced with this rabbit reticulocyte lysate would readily form oligomers of various sizes. In absence of co-factors that regulate oligomer size, one would therefore expect a natural distribution across the different oligomer sizes as observed for

recombinantly purified HSPB1. This implies that at least a significant fraction will be captured in the oligomers at any given time and these HSPB1 molecules are therefore possibly unavailable for mitochondrial import. This idea is supported by our results obtained with the disease-causing P182L mutant of HSPB1, for which we previously showed (Alderson and Adriaenssens et al. 2021 EMBO J) that it forms larger oligomers, and which, as our results indicate, have drastically reduced mitochondrial import in this manuscript. It therefore appears that dissociation from large oligomeric complexes is required for mitochondrial import, which sounds reasonable given the size of the TOMM40-channel which can simply not import large oligomeric complexes and would require monomeric subunits for import. However, in absence of further supporting evidence we did not feel comfortable enough to introduce this idea in the manuscript and have therefore decided to leave it out.

Figure 3:

This figure shows that association with mitochondria is not associated with or influenced by the phosphorylation status. Whereas this could suggest that these mitochondrial association are not directly linked to the holdase functions of HSPB1 (which are strongly reduced in the for the 3D/non-phosphorylated forms), it does raise another question: the non-phosphorylated forms are usually present in much bigger complexes and do not/less well dissociate into monomers/dimers upon heat shock for which it even seems more difficult now to understand how they could pass the OM to reach the IMS.

The reviewer is correct in stating that it is thought that the HSPB1-3A mutant and HSPB1-3D mutant form larger and smaller oligomeric complexes, respectively. At the same time, sHSPs are known for their high degree of plasticity, which allows them to form very dynamic oligomeric assemblies that continuously reshape by ejecting and incorporating subunits. This polydispersity suggests that at any given time one will find oligomers of different sizes and *in vitro* the size of oligomeric complexes increases by increasing protein concentration. However, it appears that in cells there are more regulating factors such as kinases (like the MAPK-pathway) but also the fact that oligomeric assemblies can be heterogenous and incorporate other sHSPs. For instance, the work from Johannes Buchner elegantly showed that the incorporation of HSPB6 subunits into HSPB1 oligomers leads to dissociation of the oligomer, resulting in the formation of smaller subunits and this was accompanied by increased chaperone-activity (Mymrikov et al. 2020 JBC). This therefore suggests that our experiments with HSPB1-3A/D mutants in HeLa cells is to be interpreted in a context that is far more complicated than the *in vitro* setting (from which most of the knowledge on the 3A/D mutant stems). Given the presence of multiple regulating factors in cells, our results may therefore not always translate 1:1 with the conclusions obtained in isolated and well-controlled *in vitro* experiments.

Keeping this in mind, we have observed here (but also in other unpublished work) that the 3A/D mutants behave to a large extent like the WT protein. This can be explained by the fact that phosphorylation is only one way to regulate oligomer size and that other regulating factors may still compensate in absence of N-terminal phosphorylation.

However, it is also possible that overexpressing HSPB1-3A/D mutants in naïve HeLa cells that still have endogenous HSPB1-WT expression, which results in the formation hetero-oligomers, still leads to ejection from the oligomeric assemblies as it can be driven by the HSPB1-WT protein. This would allow the HSPB1-3A mutant to form smaller assemblies, facilitating its

mitochondrial import. This forms a plausible explanation for our data and can therefore explain why even the HSPB1-3A mutant can still be imported in sufficient amounts.

Please note that while we observed many similarities in the behavior of the HSPB1-3A/D mutants in our cellular experiments, we also generated one new piece of data showing differences between the WT, 3A, and 3D mutants. In a chaperone-assay, using mutant EndoG as a probe, the 3D mutant, which is predicted to be the most chaperone-active followed by the WT and then 3A mutant, did indeed engage the most with aggregating EndoG in the insoluble fraction while soluble levels are near equal with WT and 3A.

Altogether, this shows that our data recapitulate key aspects previously described in isolated *in vitro* settings for these mutants, while other aspects like the oligomer size may be regulated in a more complicated way inside cells due to the integration of multiple regulating factors. Our results therefore reproducibly show that the 3A/D mutants are still imported into mitochondria and this can be explained by what we discussed above.

In panel A of this figure, it is clear in that this concerns heat-shocked cells. Is this also true for the other panels? Non-heat shock data should also be presented as controls (or the reversed if the other panels concerns non-heated samples).

We apologize in case we have not stated this sufficiently clear in the figure legends but this all concerns non-heat shocked conditions. The reason is that we focused this manuscript on the non-heat shock condition (as stated at lines 118-120: *We decided to focus on the role of sHSPs in the IMS under basal (non-heat shocked) conditions as, so far, only a single protein disaggregase has been identified in the mammalian IMS [20].*) and only in certain experiments we took the heat shock condition along as a control. For instance, here in panel A, we used the heat shock condition to prove that our antibodies and assay is working correctly. However, subsequent experiments in panels B and C were all done under non-heat shock conditions, as the focus of our work on the role of sHSPs in the IMS and not on the OMM (where they accumulate after heat shock).

We have however still performed the experiments as requested by the reviewer and this revealed that:

- (a) all three variants (WT/3A/3D) translocate to the mitochondrial surface after heat shock in equal amounts. While this may seem unexpected, because it is thought that the 3A mutant is locked in oligomers (at least based on *in vitro* data), we have made similar observations in other experiments (not part of this manuscript) that in cells the 3A and 3D mutant still largely behave as the WT protein in cells. To rule out that we would have swapped samples we also resequenced the plasmids and confirmed they were correct. Moreover, on the blot the 3A mutant runs a tiny bit lower on the gel, further confirming this is the 3A mutant. It is therefore possible

that the N-terminal phosphorylation is not the only factor that determines oligomer size in cells and that in cells a more complex level of regulation occurs. Alternatively, it is also possible that in cells, hetero-oligomerization with WT HSPB1 may still allow their ejection from oligomers (in the form of dimers) and that the effect of 3A/3D variants therefore is reduced.

(b) We also repeated the experiment of panel C and added the heat shock condition. This experiment showed that HSPB1 is indeed phosphorylated upon heat shock and this is accompanied by an increased translocation of HSPB1 to the mitochondrial fraction. Inhibiting the MAPK pathway with SB203580 does indeed lead to reduced HSPB1 phosphorylation but this does not prevent HSPB1 from accumulating in the mitochondrial fraction (neither under basal nor heat shock conditions).

While both results are interesting in their own respect, we think that they may dilute the key message of the manuscript, which is focused on the role of sHSPs in the mitochondrial IMS under basal conditions. Adding additional experiments with the heat shock condition (in which sHSPs accumulate on the OMM) could potentially lead to confusion of the reader, we think. We would therefore like to ask approval of the editor to leave these additional experiments out of the manuscript.

Furthermore, the protease K protection was done on mitochondria (not also mitoplasts) and can thus certainly do not warrant the conclusion that the various HSPB1 mutants were imported (lines 262-266). The same is true for the suggestive remark on HSPB1 remaining phosphorylated during import (lines 275-278). Even though the authors may

not mean to suggest this is required (it is not: the 3A variants show similar features s the wildtype protein), this statement may confuse the reader.

We thank the reviewer for pointing this out and agree that the statements may confuse the reader. We have therefore removed the sentence from the manuscript (lines 286-289) and repeated the proteinase K protection assay for each of the HSPB1-variants, including mitoplasts. We have added this as a new supplementary figure S12, to complement the quantitative panel B from Figure 3:

Figure 4:

I am a bit confused by the labelling of panel A, but if I am correct, the authors used single knockouts (of B1 or B8) and next silenced the other HSPB (B8 or B1) by shRNA. The obvious first question is why HSPB5 is also lost in these cells. The second one is, why the others not just combined the shRNA for all HSPs to transiently knockdown all together and minimize all kinds of adaptations that may be associated with the permanent knockout of these genes.

The reviewer is correct; the cell lines were knockout for one sHSP and then further depleted by shRNAs for the others. Why HSPB5 is spontaneously lost? That is something that we indeed observed and that was also striking to us. As far as we know, no prior studies reported similar observations. At the same time, we have also not come across studies that attempted to reduce all sHSPs like we did here. So, although we did not investigate this loss of HSPB5 in further detail, we speculate that the absence of HSPB1 and HSPB8 renders HSPB5 rather unstable on its own, since HSPB1-HSPB5-HSPB8 form hetero-oligomeric complexes (as shown by published and unpublished work from a collaborator). The precise underlying mechanism is to be unraveled but it shows interesting parallels with the HSPB8-BAG3 relationship, where HSPB8 also shows reduced expression in absence of BAG3. Whether this HSPB5 instability is specific to HeLa cells or is a general principle remains to be determined.

The reviewer makes an excellent suggestion to use acute depletion (via shRNAs) over CRISPR-KOs. In fact, this is exactly what we did in our first trials. However, this did not work so well due to molecular compensation. If we had a cell line that was depleted for HSPB1 (for >90%) and we attempted to transduce this line with additional shRNAs against HSPB5 and HSPB8, this led to a reduction of B5/B8 but a concomitant increase in HSPB1. While we did not investigate the underlying mechanism of this compensation, it appeared that the loss of

certain sHSPs is rapidly (and strongly) compensated by other sHSPs (see below). For instance, upon depletion of HSPB8, HSPB5 would be compensating the most. To overcome this compensation, we combined HSPB8-shRNAs with HSPB1-KOs. This approach turned out to be effective and as a consequence led to a natural reduction of HSPB5. We therefore exploited the knowledge we obtained from our transient knockdown experiments and generated combinations of shRNA-knockdowns with CRISPR/Cas9 knockouts to achieve the highest level of reduction. Attempts to reduce HSPB5 levels even further in this background were unsuccessful and led to increased levels of apoptosis, indicating that a complete loss of all sHSPs is possibly not viable for this cell line. Therefore, after testing various combinations of shRNAs with and without CRISPR-KOs, the two most successful depleted cell lines were selected and used in this study (as shown in Figure 4a).

Figure. Combined shRNA regimes result in molecular compensation of sHSPs. (a) In this experiment we first established stable individual knockdown lines and used the HSPB1-KD line for subsequent HSPB5 and HSPB8 shRNA treatment to establish the triple KD line. To our surprise we observed a reversion of the HSPB1-KD at the cost of minor reduction of the two other targets. (b) In this experiment we used the stable HSPB5-KD line to establish the triple KD line and once again we observed reversion of the HSPB5-KD and absolute minimal reduction of HSPB1/HSPB8 while on their own they generate more potent knockdowns. Based on these results we decided to proceed with CRISPR/Cas9 knockout lines to overcome the compensatory responses. We also changed the shRNA for HSPB8 to a more potent one.

For the rest, these experiments clearly revealed that reductions in all 3 of these HSPB proteins affected mitochondrial (but not ER or Golgi) morphology and function. Whereas this is striking, it cannot be excluded that this is not related to direct effects inside the IMS, especially given the minute fraction of these HSPBs being associated with the mitochondria in non-heat shock conditions.

The reviewer is absolutely correct and we have therefore made sure that our manuscript contains no suggestive sentences that would indicate such overinterpretation. We are well aware of the limitation of these experiments and agree that such direct link is not warranted. Our manuscript was therefore written very carefully in this respect.

Figure 5:

Panel A is confusing as it shows no SOD1 signal in the whole cell lysate of cells transfected with SOD1 G93A. Also, this mutant is not found in the mitochondria. Other panel: I do agree that for both the mutant SOD G93A and mutant EndoG N174A (but not the OTC mutant: nice control!) there is a slight increase in HSPB1 association with mitochondria. But, do these proteins truly aggregate in the IMS? This should be shown.

The reduced levels of mutant SOD1 in the WCL and mitochondrial fraction are due to the fact that this mutant aggregates. What is shown in panel A is the NP-40 soluble fraction and the reduced solubility of the mutant leads to a reduction on this western blot. We have enhanced the contrast so that the limited amount of soluble SOD1-G93A also becomes visible in the WCL (see panel A below). In addition, we added following sentence to the manuscript:

Note that the reduced signal for SOD1-G93A is due to its tendency to aggregate, as only the soluble fraction is displayed here.

The aggregation propensity of the SOD1-G93A also becomes clear from an experiment requested by reviewer 3. Upon suggestion, we verified whether SOD1 aggregation can be countered by overexpressing sHSPs. To this end, we compared SOD1 WT/G93A across soluble and insoluble mitochondrial fractions and found that SOD1-G93A tendency to aggregate and become insoluble is countered by overexpression of sHSPs. We think that in addition to our modifications to the text and figure, as outlined above, this panel will further clarify the distribution of SOD1-G93 across both fractions.

We also verified whether both mutants aggregate in the IMS and have added the new data as a new supplemental figure S16. Note that SOD1-G93A mutant appears resistant to proteinase K, which would be indicative for it being aggregated (in contrast to SOD1-WT which is proteinase K sensitive). We conclude that the SOD1-mutant is resistant to proteinase K because even under conditions where we added Triton X-100 and thereby opened all mitochondrial compartments (incl. mitochondrial matrix), the mutant could still not be degraded. For this mutant we can therefore conclude that it is aggregated but its precise mitochondrial localization cannot be confirmed. For EndoG however, we were able to determine its mitochondrial localization and confirmed this mutant is still imported into the mitochondrial IMS.

a**b**
In fact, I know of several papers especially on mutant SOD1 in which it aggregates inside the cytosol and that other non-mitochondrial chaperones can reduced this (and hence likely downstream) and damage to and co-aggregation with mitochondria. Can this not also explain the data in panel D?

When we scanned the literature on this, we found that the results have not always been very consistent and there remains some controversy. For instance, some studies report that mutant SOD1 can aggregate on the mitochondrial surface when recombinant SOD1 is co-incubated with mitochondria (e.g. Israelson et al. 2015 Neuron). An important note here is that this particular set-up probably favors accumulation on the OMM over mitochondrial import. Indeed, the same group reported earlier that mutant SOD1 accumulates both on the OMM and in the IMS (Liu et al. 2004 Neuron). In particular the immuno-EM images from this study convincingly show that at least a significant fraction of mutant SOD1 translocate into the mitochondrial IMS. This would then be in line with a series of other studies who came to the same conclusion (e.g. Igoudjil et al. 2011 J Neurosci, Kawamata et al. 2008 Hum Mol Gen, etc.).

To ensure that our data would not be biased by these issues, we decided to undertake an orthogonal approach that would overcome these limitations. To this end, we verified the amount of sHSP-recruitment to mitochondria in cells in which we depleted mitochondrial proteases (PARL/YME1L/OMA1). This approach was inspired by an observation of the

Langer lab who found in a proteomics dataset that HSPB1 is the most upregulated gene in the retina of YME1L knockout mice, suggesting that mitochondrial proteostasis-stress results in activation of HSPB1. In this way, we would be able to probe for sHSPs recruitment upon mitochondrial proteostasis impairment independent from overexpressing mutant proteins like EndoG or SOD1.

Figure 1 from Sprenger et al. 2021 Nat Metabolism

To this end, we depleted YME1L/PARL and YME1L/PARL/OMA1 in our HeLa cell model and could indeed confirm that reducing the expression of mitochondrial proteases of the IMM (which face the IMS) leads to increased recruitment of sHSPs to mitochondria. This supports the notion further that disturbing the mitochondrial proteostasis in the IMS leads to activation of sHSPs.

Following on these interesting results, we wondered whether sHSPs would also react to depletion of ClpB, until now the only known chaperone-component of the IMS. We therefore generated ClpB-depleted cell lines and did indeed observe increased accumulation of sHSPs in the mitochondrial fraction. Together with the results from YME1L/PARL/OMA1 depletion,

this forms another line of strong evidence that proteostasis-stress in the mitochondrial IMS results in accumulation of sHSPs in the mitochondrial fraction.

We added both results to our manuscript as new panels to Figure 5f and 5g.

Furthermore, this panel is incomplete as it lacks the panel of the insoluble material for SOD1-WT and the input fraction before detergent lysis. Also, where is soluble mutant SOD1 in the WT cells and the sHSP depl (2) cells? It is (to me) impossible to evaluate this panel in terms of aggregation.

We thank the reviewer for this question and have updated our figure with a longer exposure panel so that SOD1-GFP is visible in all wells where we could detect it. The absence of SOD1-G93A-GFP signal in the soluble fraction of sHSP-depl (2) is most likely the result of reduced mitochondrial import of the SOD1 in this line and a large proportion that aggregates in the insoluble fraction. The insoluble material for SOD1-WT was omitted from this gel because it would exceed the maximum limit of wells we can run with our NuPage precast system. We therefore opted for leaving these conditions out, as they were least informative for this experiment. There were multiple reasons why we wanted to include and display the SOD1-WT soluble fraction. First, it provides insight into the reduced mitochondrial localization of SOD1 in the sHSP-depl (2) line which provides an explanation for why we could not detect it in the soluble fraction for this particular line. However, we also added the quantification to show this in a more quantitative way. The other aspect that can be appreciated from comparing SOD1-WT versus SOD1-G93A (in the WT HeLa line) is the enrichment of sHSPs in the mitochondrial soluble fraction upon expression of the mutant SOD1, in line with our results from Fig. 5a.

Figure 6 & 7:

The use of HSPB1-mutants as trap to identify putative clients is smart at first sight but also comes with several questions. It assumes that the mutant is unaffected in both its substrate specificity and has not lost (but rather increased) its binding affinity to all substrates in general. However, even if so, do to lost chaperone activity the mutant protein may aggregate with its substrates such that now other proteins become secondary interactors with these primary misfolded HSPB1-bound substrates. Given that the affinity-enriched MS was next done on whole cell lysates, this even further increases the risks of post-lysis association with already sticky mutant HSPB1-aggregates... Yet, I agree it is striking that the interactome was enriched for mitochondrial proteins. Question again is also here whether this was done for heat shocked or non-heat shocked cells and whether one should thus not compare these here, as the most important argument from most of the above is that the presumed translocation to the IMS occurs in response to heat shock (or other disturbances in IMS protein homeostasis). Finally, the findings in panel E that HspB1 binds to the transmembrane domain of SLC25A12 (that should not be exposed in intact mitochondria) to me further supports that much of these data may be due to a post-lysis effect in which HSPB1 (or mutant-associated aggregates) binds to everything that exposes hydrophobic patches. It also refers to my previous remark in figure 1, where I propose that the associations of the HSPBs may related to fluidization of membranes after heat shock that may actually lead to exposure of such hydrophobic transmembrane segments where HSPB proteins next might interact with.

We thank the reviewer for this comment but would like to point out that all experiments were performed under basal (non-heat shock) conditions. As we explained above, small heat shock proteins accumulate on the OMM after heat shock instead of in the IMS. For this reason, we decided to focus on the non-heat shock condition throughout the manuscript, so that we could focus on the role and contribution of sHSPs to IMS-proteostasis. The risk of increased membrane fluidization is therefore not of concern here.

Regarding the concern about performing the affinity-enriched MS on whole cell lysate samples, we agree with the reviewer that this required careful validation to rule out the possibility that the interactions occurred post-lysis. We therefore performed the validation experiments on very stringent conditions by submitting the purified mitochondria to a proteinase K treatment, to retain only proteins imported into mitochondria, before proceeding with the

immunoprecipitation and western blot analysis. The fact that we can still confirm the top hits from our whole-cell MS under these stringent conditions supports the notion that the hits identified in whole-cell lysate are specific. The rationale being that only a small fraction of the total cellular pool of HSPB1 is located in mitochondria (we calculated this is approx. 1 per 275 molecules of HSPB1) and if the interactions would have been non-specific, then these mitochondrial interactions would have been snowed under by other non-specific interactions coming from the far larger proportion of HSPB1 in other cellular compartments.

Whether all the interactions we identified using the molecular trap HSPB1-S135F mutant are not due to altered substrate specificity due to the mutation is a very relevant question. At least for an important class in our dataset (the most enriched class being mitochondrial solute carriers) we have confirmed that the interactions also happen with HSPB1-WT (see Figure 6d). In fact, in some of our earlier work we also always found that increased binding by disease-causing HSPB1-mutants occurs with substrates that are naturally handled by HSPB1-WT (see Almeida-Souza et al. 2010 JBC, Almeida-Souza et al. J Neurosci 2011, Geuens et al. 2017 Acta Neuro Comm, Alderson & Adriaenssens et al. 2021 EMBO J). In particular in our most recent EMBO J publication (Alderson & Adriaenssens et al. 2021) we unraveled a framework that provides an explanation for why there is reason to believe these are substrates that are also handled by HSPB1-WT. The surface of HSPB1 contains different grooves which form binding sites for different substrates. Elegant work by others (e.g. some of Johannes Buchner or Rachel Klevit's latest papers) has shown that the different binding grooves interact with different substrates. The lab of Jason Gestwicki complemented this concept by demonstrating that back-folding of the C-terminus of HSPB1 to the B4/B8 binding groove allows the C-terminus to act as a gatekeeper for incoming clients, through direct competition between self-binding of the C-term and binding of the incoming client protein. Our EMBO publication validated this concept and revealed that disease-causing mutations in the C-terminus of HSPB1 (like the severe P182L mutation) can impair this gatekeeper function due to reduced self-binding of the C-terminus, which leaves the binding groove more frequently unbound and this results in enhanced and prolonged client binding. In our earlier work we already observed that different mutants bind to different substrates. For instance, the binding partners are very different between C-terminal mutants versus ACD-mutants. As clients bind more stably with the mutants, it allowed us to detect and identify clients through affinity-enriched MS. In contrast, the binding of clients to the WT protein is transient, most likely due to competition for binding grooves with elements in the N- or C-terminus of HSPB1 that can fold back and compete for binding grooves on the surface of the ACD. This gatekeeper/competition allows the timely release of bound clients (we hypothesize to facilitate their transfer to other chaperones like HSP70) and therefore it has been notoriously difficult to identify clients of HSPB1 with regular approaches such as affinity-enriched MS, as the interactions with clients are typically too short-lived. While we have experimental validation for this model for C-terminal mutations, we are still working on mechanistic validation of this hypothesis for mutations in the ACD, it does seem likely that the increased affinity of the HSPB1-S135F mutant and related ACD-mutants can also be explained a reduced stringency for incoming substrates and as a result in long-lasting chaperone-client complexes.

Based on this emerging framework, the increased binding to clients is not due to altered substrate specificity but rather due to decreased off-rates of natural clients that would otherwise also bind to the WT protein (but simply release faster from the WT and have therefore been more difficult to capture). All of our earlier work would support this notion, as we have so far not come across any substrate for which we could not also find evidence that it also interacts with HSPB1-WT. So while we have not been able to validate this for every individual candidate

protein in our MS dataset shown in Fig. 6a, we did provide evidence that the largest family in our MS dataset also binds the WT protein, and based on an emerging framework we hypothesize that this can be extended to most (if not all) other proteins in our dataset.

Finally, to rule out that our results would solely rely on a single method we decided to undertake an orthogonal approach with the aim to validate our mass spec findings in a way we would not rely on the mutant proteins. To this end, we came across work on *Tafazzin* knockout cells in which a key enzyme for the production of the mitochondria-specific lipid cardiolipin is inactivated. As a result, *Tafazzin* knockout cells lack cardiolipin and this causes stress to mitochondrial membrane proteins. An important class of proteins affected by the absence of cardiolipin are the mitochondrial solute carriers and we therefore requested *Tafazzin* knockout cells and used these to verify whether this mitochondrial proteostasis-stress would result in enhanced recruitment of sHSPs to mitochondria. Indeed, in both HEK293T knockout clones we detected increased amounts of HSPB1 in the mitochondrial fraction. We also observed significant clonal variation which resulted in altered expression levels of HSPB1 in the different clones, with significantly reduced amounts in particular in clone 19. However, while it was difficult to detect HSPB1 in the WCL and cytosol fractions, we could still potently detect HSPB1 in the mitochondrial fraction. This implied that the relative amounts of HSPB1 in the mitochondrial fraction was very high with respect to the total pool of HSPB1 present in these cells. With only a limited number of HSPB1 molecules available in these cells, it seems that it was judged that protecting mitochondria was a particularly important task for these chaperones as a significant proportion of HSPB1 was located in the mitochondrial fraction.

Finally, I do not think the data yet allow the conclusion (discussion lines 625-644) that HSPB1-mutants cause CMT directly by a loss of mitochondrial protein quality control (one, but certainly not the only feature in cells from CMT-derived cells) rather than (also) via defects in cytoplasmic protein quality control.

We thank the reviewer for pointing this out and we fully agree. We have therefore added following line at the end of this paragraph: *Our work therefore suggests that mutations in HSPB1 and HSPB8 may result in protein quality control defects both in the cytosol and mitochondria.*

Minor comments

Lines 43-51: Whereas bacteria indeed also have two small HSP's, there are -to my knowledge so far no data that the reside in between the inner and outer membrane. This would imply that the reported localization in the mitochondrial intermembrane space in

metazoan must have occurred later in evolution? Authors comment?

We may not be the best positioned to comment on this, as we ourselves have not worked on bacterial sHSPs and we do not know whether this has yet been investigated in detail or not. From what we found in literature, the bacterial sHSPs have so far not been reported to reside between the inner and outer membrane, which would indeed suggest that the IMS-translocation of sHSPs occurred later in evolution. Given that many other species (incl. plants) have mitochondrial sHSPs, it appears that having sHSPs in one or more mitochondrial compartments seems beneficial. It would therefore hint at a rather conserved mechanism of mitochondrial translocation, however the discrepancies between species and mitochondrial compartments require further investigation and applying an evolutionary perspective to this could be very insightful. To point out that at least some sHSPs have been found in the matrix, we have added this info to line 649 of our discussion.

Lines 52-65: This is a bit a circular argument. Whereas the provided reasoning is one possibility, many other non-direct effects of HDAC in relation to (e.g. non-mitochondrially localized HSPB mutants) could also explain the various phenotypes cited here.

We have rephrased this in the following way:

Original:

*While it remains unclear how these cytosolic chaperones can impair mitochondrial function, the observation that rescuing mitochondrial defects **leads to** phenotypic amelioration, indicates that mitochondrial dysfunction is an important contributor to the neurodegeneration [39-41].*

New version:

*While it remains unclear how these cytosolic chaperones can impair mitochondrial function, the observation that a rescue of mitochondrial defects **is accompanied by** phenotypic amelioration, suggests that mitochondrial dysfunction could be an important contributor to the neurodegeneration [39-41].*

Line 66-72: If I am correct, all the studies in plants and metazoans on sHSP in mitochondria refer to matrix-localized versions. This should be checked and if true, the current finding may thus be specific for mammals?

Yes, this is indeed correct. So far, all mitochondrial sHSPs that have been studied have been found to localize to the mitochondrial matrix. That would therefore indeed imply that our current finding would be specific to mammals. However, we have not incorporated such explicit statement in our discussion as we judged that mitochondrial-targeted sHSPs in plants (and other species) have remained understudied and from the current knowledge it is unclear whether all mitochondrial-targeted sHSPs are matrix-localized in plants or not. Therefore, as it is unclear whether this is extrapolatable to all plants, or whether there also exist IMS-localized sHSPs in certain species, we have decided to leave this hypothesis out of the manuscript.

Reviewer #3:

Remarks to the Author:

This work aims to explore the relationship between small heat shock proteins (sHSPs) and mitochondria. Using fractionation and expansion microscopy, they find that HSPB1, 2, 5, 6, 7 and 8 are found in the mitochondria and that a fraction of the total pool is in the inter-membrane space. This is an interesting finding, as a recent study identified a disaggregase in the same sub-compartment, but little else is known about the chaperones of the IMS. The authors proceed to explore the functional relevance of this localization, using knockdowns/knockouts, metabolic measurements and expression of aggregation-prone mutant proteins. Finally, they find that CMT-related mutations alter the localization patterns of HSPB1, suggesting that changes in HSPB1 mitochondrial import may contribute to disease pathology.

Overall, this is a carefully designed set of experiments, with excellent controls and well-described logic. For example, the use of the SOD, EndoG and OTC mutants is a rigorous way to explore protein aggregation in the mitochondria with precision. The finding that sHSPs are an important chaperone of the IMS is an important finding – and the extension to the CMT mutations further improves the impactfulness.

We would like to thank the reviewer for taking the time to carefully read our manuscript and we appreciate the kind words about our manuscript. We have addressed the points raised by the reviewer with additional experiments and have them outlined below.

1. Fig 2. The lower mass band of HSPB1 is interesting and unexpected, but not further explored in the subsequent studies. Because later experiments suggest that this pool of IMS-localized HSPB1 is active as a chaperone, it seems worth exploring what this band represents by mass spectrometry and/or Edmond sequencing. Is this change in molecular mass upon IMS localization observed for other sHSPs?

We thank the reviewer for this question and have performed the suggested experiment (also raised by reviewer 1). After some optimization, we were able to immunoprecipitate sufficient amounts of V5-tagged HSPB1 from purified mitochondria (pretreated with proteinase K to degrade any cytosolic contamination) and cut the band from a Coomassie stained gel before submitting the sample for mass spec analysis.

Using chymotrypsin for protein digestion, we were able to obtain sufficient coverage of the N-terminus which was not possible with a standard trypsin digest. This experiment revealed that imported HSPB1 is modified at its N-terminus. The N-terminal methionine is cleaved off and the remaining peptide is modified by acetylation, which might explain the slight shift on the gel as reported in our initial manuscript.

We have added the MS/MS spectrum as supplementary figure and modified the text accordingly:

Mass spectrometric analysis of this lower band revealed that mitochondrial HSPB1 had undergone an N-terminal modification, possibly explaining the size shift on gel (**Figure S11**).

2. Fig 5. The knockout and knockdown studies (termed sHSP-/-) are interesting and supportive of a key role for the proteins in mitochondrial quality control. However, the study would be strengthened by analysis of an over-expression phenotype. Does OE of HSPB1 protect against SOD1 aggregation? Such a finding would draw a more direct relationship between the limiting activity of the chaperones and protein aggregation.

We agree and would like to thank the reviewer for this excellent suggestion. To this end, we generated cell lines expressing either SOD-WT or SOD-G93A and compared them to cells overexpressing HSPB1/B5/B8. We then isolated mitochondria from these cells and compared the NP-40 soluble to insoluble fraction. As expected, SOD1-WT was mostly soluble and SOD1-G93A was insoluble. However, addition of HSPB1/B5/B8 reduced SOD1-G93A's propensity to aggregate and become insoluble. As we agree with the reviewer that this would

further confirm a more direct relationship between the chaperones and this substrate in the mitochondrial IMS, we added this as a new panel to Figure 5d.

3. It seems that the authors used three biological replicates for each experiment - but this was not always clear. It would be worth adding specific information about the number of technical and biological replicates to each figure legend.

We thank the reviewer for pointing this out, we have now added this information to the figure legends of each figure.

Minor:

1. BAG3 is a stress responsive protein, whose levels are significantly elevated by heat stress (42°C). Thus, it seems likely that heat shock reduces partitioning of sHSPs into the IMS, in part, because BAG3 levels are elevated. The authors nearly suggest this possibility, but it would be worth blotting for BAG3 in the heat shocked lysates to confirm and strengthen this conclusion.

We thank the reviewer for this comment and have performed the corresponding experiment to verify this. We indeed find that along with the sHSPs, also BAG3 accumulates in the mitochondrial fraction after heat shock. We agree that this would be a plausible explanation for why the partitioning of sHSPs into the IMS is reduced. We have therefore added this new piece of data as a supplementary figure to our manuscript as Figure S7.

Please note that we collected and isolated mitochondria already after 1 hour, the protein levels of BAG3 had not yet increased. This is also the case for sHSPs which also require up to 4 hours of recovery after the heat shock before protein levels increase detectably (we suspect that transcriptional upregulation happens nearly immediately during heat shock but it then still requires several hours before this increased transcription translates into detectable increased protein levels). So, the increased mitochondrial translocation occurs before BAG3 levels raised in heat shocked cells.

2. The position of molecular weight markers should be included in the western blots.

We thank the reviewer for this comment and apologize this was not yet available. Molecular weight markers have now been added.

Decision Letter, first revision:

26th October 2022

Dear Dr. Timmerman,

Thank you for submitting your revised manuscript "Small heat shock proteins operate as molecular chaperones in the mitochondrial intermembrane space" (NCB-A46820A) and for your patience with the process.

Thank you also very much for getting back to me regarding your thoughts on Rev#2's additional comments after our editorial discussion with the referee.

As you know, the revision has now been seen by the original referees and their comments are below. The reviewers find that the paper has improved in revision, and therefore we'll be happy in principle to publish it in Nature Cell Biology, pending minor revisions to satisfy the referees' final requests and to comply with our editorial and formatting guidelines. In particular, we have talked about Rev#2's hesitations further editorially and, in light of your response, we agree that text edits will be beneficial to address Rev#2's remaining points since further experiments are not possible. Please do see the full comments from this expert below -- we are hopeful that revisions to the text and clarifications can address all their remaining concerns. Please do provide a point-by-point response to the full comments recapitulating the changes made to the paper when resubmitting your files.

Please note that the current version of your manuscript is in a PDF format - could you please email us a copy of the file in an editable format (Microsoft Word or LaTeX)? We cannot proceed with PDFs at this stage. Many thanks in advance.

When the Word file is received, we will get started and perform detailed checks on your paper and will send you a checklist detailing our editorial and formatting requirements within ~2 weeks. Please do not upload the final materials and make any revisions until you receive this additional information from us.

Thank you again for your interest in Nature Cell Biology. Please do not hesitate to contact me if you have any questions.

Sincerely,

Melina

Melina Casadio, PhD
Senior Editor, Nature Cell Biology
ORCID ID: <https://orcid.org/0000-0003-2389-2243>

Reviewer #1 (Remarks to the Author):

The authors have addressed the comments of the reviewers in a detailed and careful revision. I recommend publication of this very interesting study with novel findings on small heat shock proteins and mitochondrial function in NCB.

Reviewer #2 (Remarks to the Author):

Referee NCB

First of all, I would like to state that I was impressed by the detailed, frank and careful reply of the authors.

Hereby several of my technical concerns have been answered.

However, there are still some major concerns and things that I am still confused about. This no longer concern questions as to whether small Hsp's might be present in the IMS of mitochondria, but rather to if/how these can be linked to the observed functional endpoints and hence what their actual function inside the IMS is.

First of all, as the authors carefully replied, the presence of small HSP in the inter membrane space of mitochondria of unstressed cells drastically differs from how they get associated with mitochondria upon heat stress (prot-K sensitivity as the most convincing difference). The authors also convinced me now that under basal conditions some sHSPs indeed reside within the inner membrane space of mitochondria and that this is not due to post-lysis artifacts.

The heat shock work indeed was part of my confusion and the cause of some of the concerns I raised. The authors indicate (now also clearly in their MS) that their major aim is to focus on the presence of the sHSP inside the IMS under non-stress conditions. Yet, they persist to include the heat shock-induced associations in various figures (1, 3) as if the "this increased recruitment of small heat shock proteins is functionally related to protein misfolding in the mitochondrial intermembrane space" (see abstract). Whereas I can envision that protein stress in the IMS can e.g. lead to impaired import and hence accumulation/aggregation of mitochondrial proteins near import sites and that this attracts sHSP, I do not see how this relates to the functionality of the sHSP that are inside the IMS under basal conditions. In fact, the sHSP that associated with (but are not in) mitochondria upon stress could perform their chaperone action upon mitochondrial stress in the cytosol.

Regarding the presence inside the IMS under basal condition, I now also have a minor question HOW they get in. The data on retention of import via partner interactions is an interesting but negative one, supported by most data. The finding that entry seems independent on membrane potential (CCCP) also is clear. But the question remains how these proteins that lack classical leaders do get in. Related to the latter, I am also still confused about the data on the phosphorylation mutants (Figure 3, Figure S12) and why the authors again show the heat shock experiment. HSPB1 phosphorylation is indeed a typically stress-induced event, but as mentioned above, the authors wanted to focus on IMS localization under basal condition. Indeed, stress was found to enhance heat shock induced accumulation at mitochondria (the phosphorylated forms showing stronger associations than the bulk). But, not surprisingly, nothing changed for the entry INSIDE mitochondria nor were any changes found for phosphorylation mutants or inhibitors. Another surprising finding here is that the WCL

sample seem to indicate that HSPB1 is already heavily phosphorylated under basal growth conditions and hardly show the typical heat-induced increase in phosphorylation. So, these phosphorylation data raise more questions than that they answer with regards to localization of sHSP inside the IMS under basal conditions.

Whereas the small HSP depletion data clearly show that this results in mitochondrial defects, this could still be all indirect (chaperone depletion from the cytosol) rather than due to an chaperoning effect inside the IMS. A minor concern I still also have relates to the differential effects of HSPB knockouts (due to compensatory effects) and transient knockdown. Whereas I truly appreciate the honest responses to these questions (that largely addressed my concerns), the provided answers highlights how complex it is to interpret all these data in a conclusive manner.

My major problem, however, still concerns the effects on the mutant SOD-1 G93A. As clearly is shown in figure 5 panel a, this mutant no longer accumulates in mitochondria (or only a minor portion); this is likely due to its cytoplasmatic aggregation and the prot-K experiments here are indeed non-conclusive. Hence, the effects of the small HSP on SOD-1 G93A aggregation (Figure 5 panels d,e,) is most likely unrelated to their presence inside the IMS and may equally well reflect the action of the bulk of the sHSP in the cytosol that modulates mutant SOD-1 aggregation in the cytosol reducing its accumulation at mitochondria. So, the title of figure 5 is not fully proven by the data. The endoG and delta OTC do still accumulate in mitochondria, but here no functional data are shown that can be related to the function of the IMS-located small HSP.

A minor comment that also still remained relates to the mere absence of a SOD1 signal in the whole cell lysate of cells transfected with SOD1 G93A in Figure 5. I do understand that the mutant aggregates (as the authors show), but to my knowledge SOD1 G93A aggregates are SDS-insoluble and should be detectable in westerns of WCL. If, however, this fraction concerns a soluble fraction after a lysis (which detergent?), it should not be labelled as WCL, which refers to a fraction of total cells (not just the supernatant).

The added findings that the fraction of small HSPs increase inside the IMS increases upon the depletion of YMLE1 and PARL or ClpB depletion are interesting, but what is still needed here is to be shown the prot-K insensitivity to really proof their IMS localization. Moreover, of course, a function of the HSPs inside the IMS is not directly proven hereby.

The final use of a CMT-associated HSPB1 mutant (S135F) as a trap to identify clients remains nice, even more so since I now know this was done without heat shock (and the authors indeed included prot-K in their protocols). These data do suggest that substrates for the IMS HSPB1 likely includes transmembrane proteins. The subsequent conclusion that this is why the various CMT mutants causes disease is again premature, as also most of those mutant proteins are not inside the IMS but in the cytosol. In the cytosol, they also will engage in abnormal interaction with clients (trapping/aggregation) that on one hand leads to mitochondrial dysfunction and other hand (maybe as a result) impairs the import of the mutants into the IMS. The authors now acknowledge this and adapted some of their text accordingly, yet in their abstract the suggested link between CMT and a (failed) IMS function of sHSP is still made. This should be corrected.

ADDITIONAL COMMENTS AFTER DISCUSSION WITH REV#2:

As indicated, I am excited about the first formal demonstration of IMS localisation of small HSP in metazoan. I also think the approach with the trap mutant is very nice as it suggests that certain transmembrane proteins may require the chaperone activity of IMS-located small HSPs.

Beyond that, I think that the effects on mutant SOD1 and the conclusion that CMT-mutants cause disease (just) because of failed IMS-related actions of the small HSPs are premature. How to conclusively demonstrate this indeed is a major challenge.

Maybe key experiments could be to show EM-data on the HSP effects on SOD-1 G93A aggregates at/in the IMS. These experiments could be done under HSP depletion conditions without and with re-introducing either wildtype or CMT-mutant HSPB1.

Similarly these experiments could be done without mutant SOD-1 expression to test what happens with the (structure of) IMS of mitochondria and if this is restored by re-introducing either wildtype or CMT-mutant HSPB1.

I am not sure how feasible these experiments are; especially the latter set of HSPB1 depletion may take a long time to reveal a mitochondrial phenotype.

I hope these suggestions are of help, but I also could agree with accepting the MS with tuning down some of the conclusions as I think the key findings that they do localise inside the IMS are novel and interesting indeed

Reviewer #3 (Remarks to the Author):

The authors have done a good job addressing all of the reviewer concerns. I particularly enjoyed the experiment in which they explored SOD1-G93A binding (Fig 5D) and the re-distribution upon ClpB knockdown (Fig 5G). This is a well done study on an important topic.

Decision Letter, final checks:

Our ref: NCB-A46820A

3rd November 2022

Dear Dr. Timmerman,

Thank you for your patience as we've prepared the guidelines for final submission of your Nature Cell Biology manuscript, "Small heat shock proteins operate as molecular chaperones in the mitochondrial intermembrane space" (NCB-A46820A). Please carefully follow the step-by-step instructions provided

in the attached file, and add a response in each row of the table to indicate the changes that you have made. Please also check and comment on any additional marked-up edits we have proposed within the text. Ensuring that each point is addressed will help to ensure that your revised manuscript can be swiftly handed over to our production team.

In recognition of the time and expertise our reviewers provide to Nature Cell Biology's editorial process, we would like to formally acknowledge their contribution to the external peer review of your manuscript entitled "Small heat shock proteins operate as molecular chaperones in the mitochondrial intermembrane space". For those reviewers who give their assent, we will be publishing their names alongside the published article.

Nature Cell Biology offers a Transparent Peer Review option for new original research manuscripts submitted after December 1st, 2019. As part of this initiative, we encourage our authors to support increased transparency into the peer review process by agreeing to have the reviewer comments, author rebuttal letters, and editorial decision letters published as a Supplementary item. When you submit your final files please clearly state in your cover letter whether or not you would like to participate in this initiative. Please note that failure to state your preference will result in delays in accepting your manuscript for publication.

Cover suggestions

As you prepare your final files we encourage you to consider whether you have any images or illustrations that may be appropriate for use on the cover of Nature Cell Biology.

Nature Cell Biology has now transitioned to a unified Rights Collection system which will allow our Author Services team to quickly and easily collect the rights and permissions required to publish your work. Approximately 10 days after your paper is formally accepted, you will receive an email in providing you with a link to complete the grant of rights. If your paper is eligible for Open Access, our Author Services team will also be in touch regarding any additional information that may be required to arrange payment for your article.

Please note that *Nature Cell Biology* is a Transformative Journal (TJ). Authors may publish their research with us through the traditional subscription access route or make their paper immediately open access through payment of an article-processing charge (APC). Authors will not be required to make a final decision about access to their article until it has been accepted. Find out more about Transformative Journals

Please use the following link for uploading these materials:
[Redacted]

Best regards,

Kendra Donahue
Staff
Nature Cell Biology

On behalf of

Melina Casadio, PhD
Senior Editor, Nature Cell Biology

ORCID ID: <https://orcid.org/0000-0003-2389-2243>

Reviewer #1:

Remarks to the Author:

The authors have addressed the comments of the reviewers in a detailed and careful revision. I recommend publication of this very interesting study with novel findings on small heat shock proteins and mitochondrial function in NCB.

Reviewer #2:

Remarks to the Author:

Referee NCB

First of all, I would like to state that I was impressed by the detailed, frank and careful reply of the authors.

Hereby several of my technical concerns have been answered.

However, there are still some major concerns and things that I am still confused about.

This no longer concern questions as to whether small Hsp's might be present in the IMS of mitochondria, but rather to if/how these can be linked to the observed functional endpoints and hence what their actual function inside the IMS is.

First of all, as the authors carefully replied, the presence of small HSP in the inter membrane space of mitochondria of unstressed cells drastically differs from how they get associated with mitochondria upon heat stress (prot-K sensitivity as the most convincing difference). The authors also convinced me now that under basal conditions some sHSPs indeed reside within the inner membrane space of mitochondria and that this is not due to post-lysis artifacts.

The heat shock work indeed was part of my confusion and the cause of some of the concerns I raised. The authors indicate (now also clearly in their MS) that their major aim is to focus on the presence of the sHSP inside the IMS under non-stress conditions. Yet, they persist to include the heat shock-induced associations in various figures (1, 3) as if the "this increased recruitment of small heat shock proteins is functionally related to protein misfolding in the mitochondrial intermembrane space" (see abstract). Whereas I can envision that protein stress in the IMS can e.g. lead to impaired import and hence accumulation/aggregation of mitochondrial proteins near import sites and that this attracts sHSP, I do not see how this relates to the functionality of the sHSP that are inside the IMS under basal conditions. In fact, the sHSP that associated with (but are not in) mitochondria upon stress could perform their chaperone action upon mitochondrial stress in the cytosol.

Regarding the presence inside the IMS under basal condition, I now also have a minor question HOW they get in. The data on retention of import via partner interactions is an interesting but negative one, supported by most data. The finding that entry seems independent on membrane potential (CCCP) also is clear. But the question remains how these proteins that lack classical leaders do get in. Related to the latter, I am also still confused about the data on the phosphorylation mutants (Figure 3, Figure S12) and why the authors again show the heat shock experiment. HSPB1 phosphorylation is indeed a typically stress-induced event, but as mentioned above, the authors wanted to focus on IMS

localization under basal condition. Indeed, stress was found to enhance heat shock induced accumulation at mitochondria (the phosphorylated forms showing stronger associations than the bulk). But, not surprisingly, nothing changed for the entry INSIDE mitochondria nor were any changes found for phosphorylation mutants or inhibitors. Another surprising finding here is that the WCL sample seem to indicate that HSPB1 is already heavily phosphorylated under basal growth conditions and hardly show the typical heat-induced increase in phosphorylation. So, these phosphorylation data raise more questions than that they answer with regards to localization of sHSP inside the IMS under basal conditions.

Whereas the small HSP depletion data clearly show that this results in mitochondrial defects, this could still be all indirect (chaperone depletion from the cytosol) rather than due to an chaperoning effect inside the IMS. A minor concern I still also have relates to the differential effects of HSPB knockouts (due to compensatory effects) and transient knockdown. Whereas I truly appreciate the honest responses to these questions (that largely addressed my concerns), the provided answers highlights how complex it is to interpret all these data in a conclusive manner.

My major problem, however, still concerns the effects on the mutant SOD-1 G93A. As clearly is shown in figure 5 panel a, this mutant no longer accumulates in mitochondria (or only a minor portion); this is likely due to its cytoplasmatic aggregation an the prot-K experiments here are indeed non-conclusive. Hence, the effects of the small HSP on SOD-1 G93A aggregation (Figure 5 panels d,e,) is most likely unrelated to their presence inside the IMS and may equally well reflect the action of the bulk of the sHSP in the cytosol that modulates mutant SOD-1 aggregation in the cytosol reducing its accumulation at mitochondria. So, the title of figure 5 is not fully proven by the data. The endoG and delta OTC do still accumulate in mitochondria, but here no functional data are show that can be related to the function of the IMS-located small HSP.

A minor comment that also still remained relates to the mere absence of a SOD1 signal in the whole cell lysate of cells transfected with SOD1 G93A in Figure 5. I do understand that the mutant aggregates (as the authors show), but to my knowledge SOD1 G93A aggregates are SDS-insoluble and should be detectable in westerns of WCL. If, however, this fraction concerns a soluble fraction after a lysis (which detergent?), it should not be labelled as WCL, which refers to a fraction of total cells (not just the supernatant).

The added findings that the fraction of small HSPs increase inside the IMS increases upon the depletion of YMLE1 and PARL or ClpB depletion are interesting, but what is still needed here is to be shown the prot-K insensitivity to really proof their IMS localization. Moreover, of course, a function of the HSPs inside the IMS is not directly proven hereby.

The final use of a CMT-associated HSPB1 mutant (S135F) as a trap to identify clients remains nice, even more so since I now know this was done without heat shock (and the authors indeed included prot-K in their protocols). These data do suggest that substrates for the IMS HSPB1 likely includes transmembrane proteins. The subsequent conclusion that this is why the various CMT mutants causes disease is again premature, as also most of those mutant proteins are not inside the IMS but in the cytosol. In the cytosol, they also will engage in abnormal interaction with clients (trapping/aggregation) that on one hand leads to mitochondrial dysfunction and other hand (maybe as a result) impairs the import of the mutants into the IMS. The authors now acknowledge this and adapted some of their text accordingly, yet in their abstract the suggested link between CMT and a (failed) IMS function of sHSP is still made. This should be corrected.

ADDITIONAL COMMENTS AFTER DISCUSSION WITH REV#2:

As indicated, I am excited about the first formal demonstration of IMS localisation of small HSP in metazoan. I also think the approach with the trap mutant is very nice as it suggests that certain transmembrane proteins may require the chaperone activity of IMS-located small HSPs.

Beyond that, I think that the effects on mutant SOD1 and the conclusion that CMT-mutants cause disease (just) because of failed IMS-related actions of the small HSPs are premature. How to conclusively demonstrate this indeed is a major challenge.

Maybe key experiments could be to show EM-data on the HSP effects on SOD-1 G93A aggregates at/in the IMS. These experiments could be done under HSP depletion conditions without and with re-introducing either wildtype or CMT-mutant HSPB1.

Similarly these experiments could be done without mutant SOD-1 expression to test what happens with the (structure of) IMS of mitochondria and if this is restored by re-introducing either wildtype or CMT-mutant HSPB1.

I am not sure how feasible these experiments are; especially the latter set of HSPB1 depletion may take a long time to reveal a mitochondrial phenotype.

I hope these suggestions are of help, but I also could agree with accepting the MS with tuning down some of the conclusions as I think the key findings that they do localise inside the IMS are novel and interesting indeed

Reviewer #3:

Remarks to the Author:

The authors have done a good job addressing all of the reviewer concerns. I particularly enjoyed the experiment in which they explored SOD1-G93A binding (Fig 5D) and the re-distribution upon ClpB knockdown (Fig 5G). This is a well done study on an important topic.

Author Rebuttal, first revision:

Reviewer #1 (Remarks to the Author):

The authors have addressed the comments of the reviewers in a detailed and careful revision. I recommend publication of this very interesting study with novel findings on small heat shock proteins and mitochondrial function in NCB.

We would like to thank reviewer #1 for the positive evaluation of our revisions and are grateful for the constructive suggestions that helped us make our manuscript better.

Reviewer #2 (Remarks to the Author):

Referee NCB

First of all, I would like to state that I was impressed by the detailed, frank and careful reply of the authors. Hereby several of my technical concerns have been answered.

We would like to thank reviewer #2 for sharing this feedback. We are grateful for the constructive and positive approach of the reviewer toward our manuscript and this has undoubtedly improved our work significantly. Please find below our point-to-point replies to the remaining questions.

However, there are still some major concerns and things that I am still confused about. This no longer concerns questions as to whether small Hsp's might be present in the IMS of mitochondria, but rather to if/how these can be linked to the observed functional endpoints and hence what their actual function inside the IMS is.

First of all, as the authors carefully replied, the presence of small HSP in the inter membrane space of mitochondria of unstressed cells drastically differs from how they get associated with mitochondria upon heat stress (prot-K sensitivity as the most convincing difference). The authors also convinced me now that under basal conditions some sHSPs indeed reside within the inner membrane space of mitochondria and that this is not due to post-lysis artifacts.

The heat shock work indeed was part of my confusion and the cause of some of the concerns I raised. The authors indicate (now also clearly in their MS) that their major aim is to focus on the presence of the sHSP inside the IMS under non-stress conditions. Yet, they persist to include the heat shock-induced associations in various figures (1, 3) as if the “this increased recruitment of small heat shock proteins is functionally related to protein misfolding in the mitochondrial intermembrane space” (see abstract). Whereas I can envision that protein stress in the IMS can e.g. lead to impaired import and hence accumulation/aggregation of mitochondrial proteins near import sites and that this attracts sHSP, I do not see how this relates to the functionality of the sHSP that are inside the IMS under basal conditions. In fact, the sHSP that associated with (but are not in) mitochondria upon stress could perform their chaperone action upon mitochondrial stress in the cytosol.

We thank the reviewer for sharing this comment and are happy to share our thought process on incorporating/excluding the heat shock condition in figures 1 and 3. The heat shock condition was incorporated in Figure 1 because these are the experiments in which we demonstrate that the localization of sHSPs differs after heat shock. Figure 1 therefore also gives us the opportunity to explicitly state that the manuscript will only focus on the basal condition (under which the sHSPs reside in the IMS). In Figure 3, we prefer to show the heat shock condition once more, because we think that heat shock serves its role as a positive control for the phospho-antibodies. As it is well known that the phosphorylation of sHSPs increases after heat shock, we used this in panel 3A to prove the specificity of the antibodies and to help the interpretation of the weak signals in the basal (non-heat shock conditions), as the bands of p-HSPB1 (S15, S78, S82) are fairly weak for the mitochondrial samples in non-heat shock condition. We feel that the heat shock condition is an important technical control in Figure 3A, but we refrained from attaching scientific conclusions to these samples. The results of the heat shock condition in Figure 3A were therefore also not discussed in the main text, other than pointing out that the total levels of HSPB1 in the mitochondrial fraction also increase after heat shock.

Original (page 11 – line 261-263):

We first evaluated whether mitochondrial HSPB1 is phosphorylated. Using phospho-specific antibodies we detected low amounts of phosphorylated HSPB1 in the mitochondrial fraction under basal conditions, which increased after heat shock (**Figure 3a**). Note that under heat shock conditions, not only the amount of phosphorylated HSPB1 increases in the mitochondrial fraction, but also the total amount of HSPB1 due to increased recruitment of sHSPs to the mitochondrial outer membrane.

Replaced by:

We first evaluated whether mitochondrial HSPB1 is phosphorylated. **To this end, we used phospho-specific antibodies and used the heat shock condition as a positive control for increased phosphorylation.** We detected low amounts of phosphorylated HSPB1 in the mitochondrial fraction under basal conditions, which increased after heat shock (**Figure 3a**). Note that under heat shock conditions, not only does the amount of phosphorylated HSPB1 increases in the mitochondrial fraction, but also the total amount of HSPB1 due to increased recruitment of sHSPs to the mitochondrial outer membrane.

Regarding the presence inside the IMS under basal condition, I now also have a minor question HOW they get in. The data on retention of import via partner interactions is an interesting but negative one, supported by most data. The finding that entry seems independent on membrane potential (CCCP) also is clear. But the question remains how these proteins that lack classical leaders do get in.

Related to the latter, I am also still confused about the data on the phosphorylation mutants (Figure 3, Figure S12) and why the authors again show the heat shock experiment. HSPB1 phosphorylation is indeed a typically stress-induced event, but as mentioned above, the authors wanted to focus on IMS localization under basal condition. Indeed, stress was found to enhance heat shock induced accumulation at mitochondria (the phosphorylated forms showing stronger associations than the bulk). But, not surprisingly, nothing changed for the entry INSIDE mitochondria nor were any changes found for phosphorylation mutants or inhibitors. Another surprising finding here is that the WCL sample seem to indicate that HSPB1 is already heavily phosphorylated under basal growth conditions and hardly show the typical heat-induced increase in phosphorylation. So, these phosphorylation data raise more questions than that they answer with regards to localization of sHSP inside the IMS under basal conditions.

How sHSPs are imported into mitochondria is indeed an important question and one that we would have liked to be able to answer experimentally. However, there are significant challenges associated with addressing this question, as we discussed in the Discussion (page 26 – lines 644-657), such as ‘Elucidating whether mammalian sHSPs harbor an MTS is complicated further by the fact that all three protein domains are involved in the secondary, tertiary and quaternary structure, in ways that are incompletely understood. With limited information on how mutations or deletion variants affect the dynamic ensemble of sHSP-oligomers, it will be a challenging task to verify if and where mammalian sHSPs have an MTS’. Our efforts to elucidate how sHSPs are imported into mitochondria were hampered by the incomplete knowledge of the structure-function relationship of sHSPs. As this prevented us from generating conclusive results, we have refrained from incorporating these data in the final manuscript.

However, we are happy to elaborate here on the experiments we undertook to address this question. First, we ruled out that the import made use of the Mia40-import pathway (by depleting Mia40 to such an extent that known Mia40-import substrates became depleted from mitochondria, but this did not reduce the levels of mitochondrial HSPB1). This was not unexpected as HSPB1 is the only sHSP that forms a disulphide bridge and other sHSPs would therefore not match the criteria of typical Mia40-substrates.

We then proceeded with experiments in which we generated chimera constructs, using the knowledge that HSPB1 is imported into mitochondria but HSPB3 is not. However, as mentioned above, by replacing the

N- or C-terminus of HSPB1 with that from HSPB3 (and vice versa) we may have been modulating not only the mitochondrial import sequence but also the chaperone activity and oligomeric state (which is comprised of secondary, tertiary, and quaternary structures of which the dynamic chaperone ensembles are composed) in ways that are hard to predict. For these reasons, the current knowledge of the structure-function relationship of sHSPs becomes a limiting factor in the search for the mitochondrial import mechanism.

The mitochondrial targeting motif is also still unknown for many other IMS resident proteins. It might therefore well be that additional import pathways specific to the IMS are still waiting to be discovered (cfr. the recent discovery of a new import pathway for alpha-helical proteins of the OMM by the groups of Rebecca Voorhees and Jonathan Weissman; DOI: [10.1126/science.add1856](https://doi.org/10.1126/science.add1856)). With this in mind, we decided to add a specific section in the discussion of this manuscript addressing this question as we agree with the reviewer that it is an important question that would merit further research in the future (Page 26, lines 644-657).

Whereas the small HSP depletion data clearly show that this results in mitochondrial defects, this could still be all indirect (chaperone depletion from the cytosol) rather than due to an chaperoning effect inside the IMS. A minor concern I still also have relates to the differential effects of HSPB knockouts (due to compensatory effects) and transient knockdown. Whereas I truly appreciate the honest responses to these questions (that largely addressed my concerns), the provided answers highlights how complex it is to interpret all these data in a conclusive manner.

My major problem, however, still concerns the effects on the mutant SOD-1 G93A. As clearly is shown in figure 5 panel a, this mutant no longer accumulates in mitochondria (or only a minor portion); this is likely due to its cytoplasmatic aggregation and the prot-K experiments here are indeed non-conclusive. Hence, the effects of the small HSP on SOD-1 G93A aggregation (Figure 5 panels d,e,) is most likely unrelated to their presence inside the IMS and may equally well reflect the action of the bulk of the sHSP in the cytosol that modulates mutant SOD-1 aggregation in the cytosol reducing its accumulation at mitochondria. So, the title of figure 5 is not fully proven by the data. The endoG and delta OTC do still accumulate in mitochondria, but here no functional data are shown that can be related to the function of the IMS-located small HSP.

We thank the reviewer for this comment and have replaced the title from Figure 5. By replacing the word 'prevent' with 'respond' the title better captures the results from EndoG and OTC, without overinterpreting the SOD1 results.

Original:

Figure 5. Small heat shock proteins prevent the aggregation of client proteins in the mitochondrial intermembrane space**Replaced by:****Figure 5. Small heat shock proteins respond to the aggregation of client proteins in the mitochondrial intermembrane space**

A minor comment that also still remained relates to the mere absence of a SOD1 signal in the whole cell lysate of cells transfected with SOD1 G93A in Figure 5. I do understand that the mutant aggregates (as the authors show), but to my knowledge SOD1 G93A aggregates are SDS-insoluble and should be detectable in westerns of WCL. If, however, this fraction concerns a soluble fraction after a lysis (which detergent?), it should not be labelled as WCL, which refers to a fraction of total cells (not just the supernatant).

The detergent we used here was NP-40. The whole cell lysate (WCL) indeed consisted of the soluble fraction of the WCL. However, for reasons of clarity, we would prefer to stick to the abbreviation WCL in the figures (to make it clear this fraction contains both the cytosol and mitochondrial proteins). We have, however, mentioned this now more explicitly in the M&M as well as the figure legends.

The added findings that the fraction of small HSPs increase inside the IMS increases upon the depletion of YMLE1 and PARL or ClpB depletion are interesting, but what is still needed here is to be shown the prot-K insensitivity to really proof their IMS localization. Moreover, of course, a function of the HSPs inside the IMS is not directly proven hereby.

We thank the reviewer for this comment and agree that it would be consistent with, but not directly prove that their localization is in the IMS under these proteostasis-compromising conditions. However, as we showed for other substrates of proteostasis-disruption, such as SOD1, this enrichment does effectively occur in the mitochondrial IMS. Therefore, while we have not worked this out for every proteostasis perturbation tested in Figure 5, all of our data are consistent with the idea that the enrichment occurs in the IMS. Moreover, in the manuscript (page 17-18), we made sure to refer to these data as ‘resulted in sHSP-recruitment to the mitochondrial fraction’ and not ‘in the IMS’ as we agree with the reviewer that these data do not allow us to extend this conclusion to the IMS but only to the mitochondrial fraction.

The final use of a CMT-associated HSPB1 mutant (S135F) as a trap to identify clients remains nice, even more so since I now know this was done without heat shock (and the authors indeed included prot-K in

their protocols). These data do suggest that substrates for the IMS HSPB1 likely includes transmembrane proteins. The subsequent conclusion that this is why the various CMT mutants causes disease is again premature, as also most of those mutant proteins are not inside the IMS but in the cytosol. In the cytosol, they also will engage in abnormal interaction with clients (trapping/aggregation) that on one hand leads to mitochondrial dysfunction and other hand (maybe as a result) impairs the import of the mutants into the IMS. The authors now acknowledge this and adapted some of their text accordingly, yet in their abstract the suggested link between CMT and a (failed) IMS function of sHSP is still made. This should be corrected.

We thank the reviewer for this comment and agree, as we also mentioned in our previous rebuttal, that the jury is still out on whether the chaperone-dysfunction results in CMT symptoms through dysfunction in the cytosol, mitochondria, or both. We have made sure this message is clear from the manuscript by stressing on this in the discussion, following the excellent suggestion from this reviewer. We have now also slightly amended the sentence in the abstract to make sure this is equally balanced and in line with current literature, including the findings from this manuscript.

Original:

Charcot-Marie-Tooth (CMT) disease-causing mutations disturb the mitochondrial function of HSPB1, linking previously observed mitochondrial dysfunction in CMT type 2F to its role in the mitochondrial intermembrane space.

Replaced by:

Charcot-Marie-Tooth (CMT) disease-causing mutations disturb the mitochondrial function of HSPB1, **potentially** linking previously observed mitochondrial dysfunction in CMT type 2F to its role in the mitochondrial intermembrane space.

ADDITIONAL COMMENTS AFTER DISCUSSION WITH REV#2:

As indicated, I am excited about the first formal demonstration of IMS localisation of small HSP in metazoan. I also think the approach with the trap mutant is very nice as it suggests that certain transmembrane proteins may require the chaperone activity of IMS-located small HSPs.

Beyond that, I think that the effects on mutant SOD1 and the conclusion that CMT-mutants cause disease (just) because of failed IMS-related actions of the small HSPs are premature. How to conclusively demonstrate this indeed is a major challenge.

Maybe key experiments could be to show EM-data on the HSP effects on SOD-1 G93A aggregates at/in the IMS. These experiments could be done under HSP depletion conditions without and with re-introducing either wildtype of CMT-mutant HSPB1. Similarly these experiments could be done without mutant SOD-1 expression to test what happens with the (structure of) IMS of mitochondria and if this is restored by re-introducing either wildtype of CMT-mutant HSPB1. I am not sure how feasible these experiments are; especially the latter set of HSPB1 depletion may take a long time to reveal a mitochondrial phenotype.

I hope these suggestions are of help, but I also could agree with accepting the MS with tuning down some of the conclusions as I think the key findings that they do localise inside the IMS are novel and interesting indeed.

We would like to thank the reviewer for the feedback and the very constructive reviewing process. We very much appreciate the reviewer's input and are grateful for the reviewer's effort to help us make our manuscript better.

While we feel fully confident about our conclusions regarding the SOD1-mutants, we also understand the reservations the reviewer brings up on this topic. In short, we unfortunately have no meaningful additional data that would be useful regarding this item. As the reviewer rightfully mentions, further full experimental proof of the IMS chaperone function is not a trivial challenge. The specific experiments proposed surely seem very interesting, and we have often thought ourselves about performing these kinds of setups, but in the end, we encountered too many unknown factors and various technical hurdles that would prevent us from making solid conclusions from such experiments, which would require for example subtle quantification of phenotypes at the EM level of SOD1-transfected triple sHSP-depleted cells, in comparison with many other conditions. We think that performing these experiments would therefore take a considerable amount of time with no guarantee of success. We think it would be important that all current data on the SOD1 mutants are kept in the study, but have scanned the manuscript (as specified above) to edit the text so that our text now matches the expectations of this reviewer and

the editor. Moreover, importantly, we added a paragraph in the Discussion to acknowledge the limitations of our study.

Regarding the CMT-mutants, we understand the reviewers' feedback on this subject but would like to point out that in the current manuscript we have already been very careful to avoid making overstatements about the mutants. In the revised manuscript, we, therefore, wrote that *'the mitochondrial phenotypes observed in iPSC-derived neuronal cultures and mouse models for HSPB1 mutants could in part stem from a disturbed mitochondrial function of HSPB1 mutants'* (page 23, lines 583-586). In combination with previously published work on the mutants, showing specific mitochondrial defects (such as; mitochondrial transport defects, changes in mitochondrial morphology, defects in mitochondrial ATP production, etc.), we feel that this statement is balanced and in line with the latest findings on these mutants. As CMT can also be caused by other genes, including mitochondrial genes like mitofusin (MFN2), we think our data on the HSPB1 mutants could provide an explanation for why HSPB1 mutants display mitochondrial defects and provide a link between HSPB1 and other CMT-causing genes. At the same time, we also mentioned that *'our work suggests that mutations in HSPB1 may result in protein quality control defects in both the cytosol and mitochondria'* (page 28, line 718-719), which would not exclude the involvement of other (cytoplasmic) pathways. We would have, however, have no objection to adding the latter more explicitly and have, therefore, as suggested by this reviewer above, amended the abstract to make this point clearer.

Reviewer #3 (Remarks to the Author):

The authors have done a good job addressing all of the reviewer concerns. I particularly enjoyed the experiment in which they explored SOD1-G93A binding (Fig 5D) and the re-distribution upon ClpB knockdown (Fig 5G). This is a well done study on an important topic.

We thank the reviewer for this feedback and are grateful to hear that our revision experiments were well received.

Final Decision Letter:

Dear Dr Timmerman,

I am pleased to inform you that your manuscript, "Small heat shock proteins operate as molecular chaperones in the mitochondrial intermembrane space", has now been accepted for publication in Nature Cell Biology. Congratulations on this very nice study!

Please note that *Nature Cell Biology* is a Transformative Journal (TJ). Authors may publish their research with us through the traditional subscription access route or make their paper immediately open access through payment of an article-processing charge (APC). Authors will not be required to make a final decision about access to their article until it has been accepted. Find out more about Transformative Journals

If you have not already done so, we strongly recommend that you upload the step-by-step protocols used in this manuscript to the Protocol Exchange (www.nature.com/protocolexchange), an open online resource established by Nature Protocols that allows researchers to share their detailed experimental know-how. All uploaded protocols are made freely available, assigned DOIs for ease of citation and are fully searchable through nature.com. Protocols and Nature Portfolio journal papers in which they are used can be linked to one another, and this link is clearly and prominently visible in the online versions of both papers. Authors who performed the specific experiments can act as primary authors for the Protocol as they will be best placed to share the methodology details, but the Corresponding Author of the present research paper should be included as one of the authors. By uploading your Protocols to Protocol Exchange, you are enabling researchers to more readily reproduce or adapt the methodology you use, as well as increasing the visibility of your protocols and papers. You can also establish a dedicated page to collect your lab Protocols. Further information can be found at www.nature.com/protocolexchange/about

With kind regards,

Melina

Melina Casadio, PhD
Senior Editor, Nature Cell Biology
ORCID ID: <https://orcid.org/0000-0003-2389-2243>
